# Multimodal cell maps as a foundation for structural and functional genomics

Leah V. Schaffer[1,17], Mengzhou Hu[1,17], Gege Qian[1,2], Kyung-Mee Moon[3], Abantika Pal[4], Neelesh Soni[4], Andrew P. Latham[4], Laura Pontano Vaites[5], Dorothy Tsai[1], Nicole M. Mattson[1], Katherine Licon[1], Robin Bachelder[1], Anthony Cesnik[6], Ishan Gaur[6], Trang Le[6], William Leineweber[6], Aji Palar[4], Ernst Pulido[6], Yue Qin[1,7], Xiaoyu Zhao[1], Christopher Churas[1], Joanna Lenkiewicz[1], Jing Chen[1], Keiichiro Ono[1], Dexter Pratt[1], Peter Zage[8], Ignacia Echeverria[9,10], Andrej Sali[4,10,11], J. Wade Harper[5], Steven P. Gygi[5], Leonard J. Foster[3], Edward L. Huttlin[5✉], Emma Lundberg[6,12,13,14✉] & Trey Ideker[1,15,16✉]

Human cells consist of a complex hierarchy of components, many of which remain unexplored[1,2]. Here we construct a global map of human subcellular architecture through joint measurement of biophysical interactions and immunofluorescence images for over 5,100 proteins in U2OS osteosarcoma cells. Self-supervised multimodal data integration resolves 275 molecular assemblies spanning the range of $10^{-8}$ to $10^{-5}$ m, which we validate systematically using whole-cell size-exclusion chromatography and annotate using large language models[3]. We explore key applications in structural biology, yielding structures for 111 heterodimeric complexes and an expanded Rag–Ragulator assembly. The map assigns unexpected functions to 975 proteins, including roles for C18orf21 in RNA processing and DPP9 in interferon signalling, and identifies assemblies with multiple localizations or cell type specificity. It decodes paediatric cancer genomes[4], identifying 21 recurrently mutated assemblies and implicating 102 validated new cancer proteins. The associated Cell Visualization Portal and Mapping Toolkit provide a reference platform for structural and functional cell biology.

Human cells are organized across a spatial hierarchy of components, ranging from small protein complexes at the scale of nanometres to large condensates, compartments and organelles at the scale of micrometres[5,6]. One of the ultimate goals of the biological sciences is to understand this multiscale subcellular organization and its relationship to biological function and human disease. As much of cell structure still remains uncharted, there has been long-standing interest in strategies to map this architecture systematically[7–9].

A variety of complementary technologies have been implemented for systematically determining subcellular organization across scales. In particular, methods such as whole-cell electron microscopy have led to maps of subcellular organelles and their placement within cells[10,11]. Protein immunofluorescence (IF) staining[12] and endogenous fluorescent tagging[13], coupled to confocal microscopy imaging, have begun to reveal the subcellular locations of proteins. Biochemical proteomics approaches, such as affinity purification–mass spectrometry (AP–MS)[14], cross-linking MS[15], size-exclusion chromatography–MS (SEC–MS)[16,17], proximity labelling[18] and isotope tagging[19,20] have revealed patterns of protein–protein interaction and subcellular localization that inform the makeup of protein complexes and organelles. Although these cell mapping technologies have typically been applied separately, integration of multiple complementary data modalities provides the opportunity to incorporate biological structure robustly across physical scales. Towards this aim, we recently demonstrated proof-of-concept for how two modalities—protein IF and AP–MS profiles—can be computationally fused to systematically map subcellular assemblies, with the initial version covering 661 human proteins[21].

Here we substantially scale the cell mapping datasets and pipeline, yielding protein biophysical interactions and protein IF images for a matched set of more than 5,100 proteins in U2OS cells (Fig. 1). Integrating these data produces a global cell biology reference map with extensive coverage of human subcellular components, including 275 distinct protein assemblies. We systematically annotate this map, assisted by recent advances in large language models (LLMs), then systematically validate its assemblies by generating a third distinct data modality—proteome-wide SEC–MS—in the same U2OS cellular

[1]Department of Medicine, University of California San Diego, La Jolla, CA, USA. [2]Bioinformatics and Systems Biology Program, University of California San Diego, La Jolla, CA, USA. [3]Department of Biochemistry & Molecular Biology, Michael Smith Laboratories, University of British Columbia, Vancouver, British Columbia, Canada. [4]Department of Bioengineering and Therapeutic Sciences, University of California San Francisco, San Francisco, CA, USA. [5]Department of Cell Biology, Harvard Medical School, Boston, MA, USA. [6]Department of Bioengineering, Stanford University, Palo Alto, CA, USA. [7]Broad Institute of MIT and Harvard, Boston, MA, USA. [8]Department of Pediatrics, Division of Hematology-Oncology, University of California San Diego, La Jolla, CA, USA. [9]Department of Cellular and Molecular Pharmacology, University of California San Francisco, San Francisco, CA, USA. [10]Quantitative Biosciences Institute, University of California San Francisco, San Francisco, CA, USA. [11]Department of Pharmaceutical Chemistry, University of California San Francisco, San Francisco, CA, USA. [12]Department of Pathology, Stanford University, Palo Alto, CA, USA. [13]Science for Life Laboratory, School of Engineering Sciences in Chemistry, Biotechnology and Health, KTH Royal Institute of Technology, Stockholm, Sweden. [14]Chan Zuckerberg Biohub, San Francisco, CA, USA. [15]Department of Computer Science and Engineering, University of California San Diego, La Jolla, CA, USA. [16]Department of Bioengineering, University of California San Diego, La Jolla, CA, USA. [17]These authors contributed equally: Leah V. Schaffer, Mengzhou Hu. ✉e-mail: edward_huttlin@hms.harvard.edu; emmalu@stanford.edu; tideker@ucsd.edu

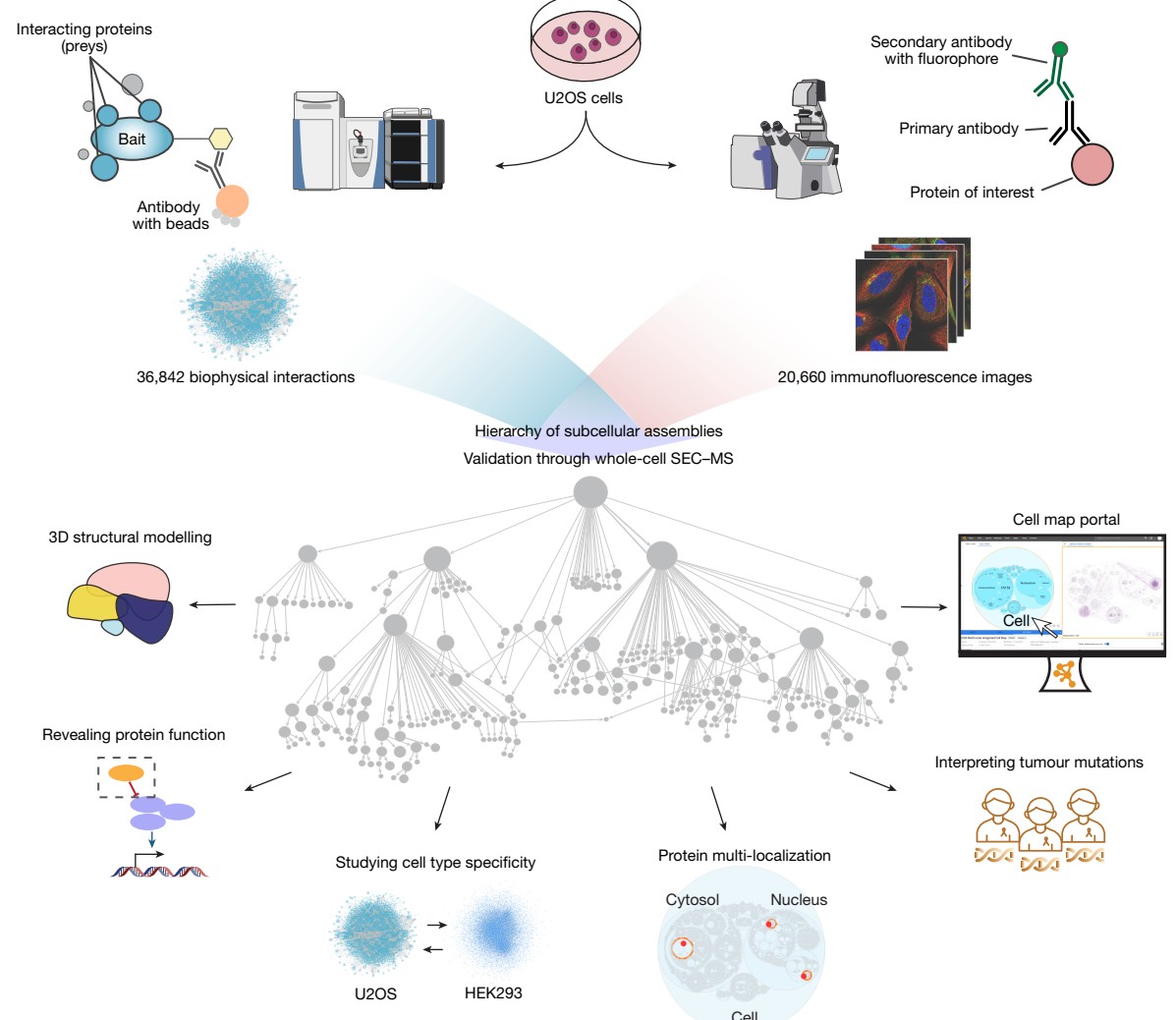

**Fig. 1 | Study overview.** Proteins are purified from whole-cell biochemical extracts and their biophysical interactions are determined using AP–MS. In parallel, proteins are illuminated by IF and their subcellular distributions are determined using high-resolution confocal imaging. These IF imaging and biophysical interaction data are integrated into a multimodal cell map, which is explored across five biological use cases and in an interactive visualization portal. MS and confocal microscopy illustrations are from the NIAID NIH BIOART Source (https://bioart.niaid.nih.gov/bioart/286; https://bioart.niaid.nih.gov/bioart/86).

context. Finally, we examine how such proteome-wide cell maps can be used to guide diverse biological studies including structural biology, protein functional annotation, analyses of cell-type specificity and multi-localization, and interpretation of the cancer genome.

## Multimodal proteomics data acquisition

We systematically tagged proteins in U2OS osteosarcoma cells through lentiviral expression of C-terminal Flag–HA-tagged baits available in the human ORFeome library[14]. A total of 2,174 proteins were successfully tagged and isolated from U2OS whole-proteome extracts using affinity purification, and interacting partners were identified by tandem MS (AP–MS) to yield a total of 36,842 interactions among 7,543 proteins (Methods and Extended Data Fig. 1a,b). Data were required to pass a panel of quality-control measures implemented as previously described[14,22]; these measures included sequence validation of lentiviral clones, detection of tagged bait proteins in each AP–MS run, and monitoring for sufficient numbers of protein and peptide identifications (Methods). Additional quality-control metrics included recovery of known complexes (Extended Data Fig. 1c,d), for which the new interactions showed coverage comparable to previous AP–MS datasets.

To match these protein interactions with parallel information on protein subcellular locations, we amassed a large collection of confocal images of U2OS cells stained with IF antibodies against each of 10,348 proteins (20,660 images total; Methods). Each sample was simultaneously co-stained with reference markers for nucleus, endoplasmic reticulum and microtubules, providing a reference set of subcellular landmarks common to all images. Of these data, 17,368 images were collected in a previous publication[12], and the remaining 3,292 images were more recently generated and validated according to the Human Protein Atlas (HPA) standard procedures for image and antibody quality control.

Combining across the interaction and imaging data, a total of 5,147 proteins was well represented in both modalities. These proteins captured approximately half of the detectable U2OS proteome[12] and provided representative coverage over the full catalogue of human protein functions, other than under-representation of transmembrane and immunoglobulin proteins (Extended Data Fig. 1e). We found that the protein pairs measured as most similar by one modality were enriched for pairs similar in the other, showing that the biophysical interaction and imaging data share information (Extended Data Fig. 1f,g).

## Construction of a global cell map

We devised a self-supervised machine learning approach for fusing protein confocal imaging and biophysical interaction data to create a global map of protein subcellular organization (Methods). First, the two data streams were processed separately to generate protein features for each modality; this information was subsequently fused to create a unified multimodal embedding for each protein. Achieving a quality embedding—a low-dimensional representation extracted from complex high-dimensional data—has been a major focus of machine learning research in recent years[23,24]. Here we adopted a self-supervised embedding approach (Extended Data Fig. 2a), in which proteins were positioned such that the original imaging and AP–MS features could each be reconstructed with minimal loss of information (reconstruction loss) while capturing the relative similarities and differences of each protein to others in both data modalities (contrastive loss). This multimodal embedding exhibited good performance in recovering known subcellular organization (Fig. 2a and Extended Data Fig. 2b–d), performing as well as, or better than, alternative supervised and unsupervised approaches (Methods and Extended Data Fig. 2e).

Once the multimodal embedding had been learned, all pairwise protein–protein distances were computed and analysed using the multiscale community detection technique (Methods). Using this procedure, protein assemblies were resolved as modular communities of proteins in close proximity to one another, with such detection performed at multiple resolutions to identify protein assemblies at increasing diameters.

Application of this analytical pipeline to the data generated in U2OS osteosarcoma cells identified a hierarchy of 275 discrete protein assemblies (Fig. 2b and Supplementary Table 1). By calibrating the map using 13 well-known subcellular components with characterized physical sizes (for example, nucleus, mitochondria and proteasome; Supplementary Table 2), we found that we could translate the size of an assembly (number of proteins) to an estimate of its physical diameter (in nanometres, $R^2 = 0.90$) along with a prediction interval on this estimate (Methods). Estimated assembly diameters spanned the relevant scales of cell biology from $10^1$ nm to $10^4$ nm (Fig. 2c), with assemblies robustly identified at each of these scales (Methods and Extended Data Fig. 3a). By contrast, we found that maps constructed from only the imaging data tended to recover large assemblies but miss small ones, while maps constructed from only the AP–MS data recovered small assemblies but tended to miss large ones (Extended Data Fig. 3b,c). Overall, the integrated map identified the largest number of assemblies, including 104 that were not resolved by either individual modality (Extended Data Fig. 3d and Supplementary Table 1).

## Annotation of the U2OS cell map

To study and annotate the U2OS cell map, we held a series of in-person Annotation Jamborees, during which approximately a dozen individuals worked in pairs to assign names and putative functional roles to assemblies on the basis of expert knowledge and literature curation. First we examined the correspondence of assemblies to known subcellular components documented in the Comprehensive Resource of Mammalian protein complexes (CORUM)[25], Gene Ontology (GO)[26] or HPA[12] (Methods; Jaccard index ≥ 10%). We found that 41 assemblies closely reconstructed a known component (Jaccard index ≥ 50%) while 90 had moderate agreement, with some unexpected differences (20% ≤ Jaccard index < 50%).

The remaining 144 assemblies were designated as not previously documented assemblies. In these cases, team members worked collaboratively to consider the current biological literature relevant to the assembly's protein subunits and their potential functions. This process was greatly informed by suggestions from OpenAI's pre-trained transformer (GPT-4)[27], a generative LLM that we recently showed is capable of providing insightful names and functional interpretations for gene sets identified in omics data[3]. As in this previous study, we used an engineered prompt and pipeline (Methods and Extended Data Fig. 4a,b) to guide the LLM to generate descriptive names for gene sets indicative of their biological roles, along with a fully referenced analysis essay providing its rationale (Extended Data Fig. 4c) and a self-assessment of confidence in the suggested name. When applied to the U2OS cell map, we found that the LLM assigned names to known assemblies with very high confidence (median of 0.92 for both high overlap and substantial variation; Fig. 2d) and to the previously undocumented assemblies with moderately high confidence (median, 0.85), contrasting starkly with its confidence for sets of proteins drawn randomly without any correspondence to biological structure (median, 0.0). For 104 out of the 144 not previously documented assemblies, the literature about the various proteins was sufficiently coherent for GPT-4 to propose a confident assembly name (confidence ≥ 0.85), each of which was subsequently passed to the human curation team for final naming determination (Supplementary Table 1).

We noted that the highest level of organization in the cell map covers previously documented organelles and large subcellular compartments of >100 proteins, including the nucleus with 102 nuclear subassemblies, the mitochondrion with 16 mitochondrial subassemblies, 127 assemblies inside the cytosol and 3 assemblies related to microtubules (Fig. 2b). Organized within the nucleus are subcomponents such as nucleoli and the nucleoplasm, which itself hierarchically resolves 67 components including the Mediator and RNA polymerase complexes and an array of other transcriptional machines. Notably, components of the plasma membrane and cytosolic periphery, such as G-protein and clathrin-coated-pit complexes, are tightly associated with numerous other cytosolic proteins under a single large compartment, which we simply labelled 'cytosol' (Fig. 2b). Major expected components of the cytosolic compartment, such as the endoplasmic reticulum and Golgi apparatus, are also resolved. We found 48 assemblies that are potential biomolecular condensates[28] on the basis of their enrichment for proteins with intrinsically disordered regions, proteins predicted to phase separate or proteins recorded in the CD-Code condensate database (Methods and Supplementary Table 3). Of these, 39 had a significant overlap with a recent complementary effort to predict protein condensates through integration of diverse biochemical protein features[29] (hypergeometric test FDR < 5%), while the remaining nine putative condensates had not been previously identified (Supplementary Table 3).

## Systematic validation by SEC–MS

We next sought to systematically validate the cell map components using whole-cell SEC–MS as an orthogonal approach. Using this technique, cellular extracts from a cell population of interest are separated by SEC, followed by identification of proteins in each size fraction by tandem MS (Fig. 3a). Here we subjected triplicate cultures of U2OS cells to SEC–MS of 40 separate chromatography fractions, yielding quantitative fractionation profiles for 5,509 proteins in at least two replicates, of which 3,020 were present in the cell map. Quality assessment of the SEC–MS dataset showed that elution profiles were largely reproducible across replicate biological measurements (Extended Data Fig. 5a,b), with protein peaks present across the full range of fractions (Extended Data Fig. 5c).

Integration of these measurements with the multiscale cell map revealed significant agreement, with proteins in the same assembly (as identified earlier by AP–MS and imaging) having a strong tendency to co-elute in the same chromatography size fractions (Fig. 3b,c). Overall, SEC data validated 89 assemblies (5% false-discovery rate (FDR)), corresponding to 43% of assemblies (76 out of 175) with more than 5 proteins and 61% of assemblies (59 out of 96) with more than 15 proteins (Fig. 3d, Methods and Supplementary Table 4).

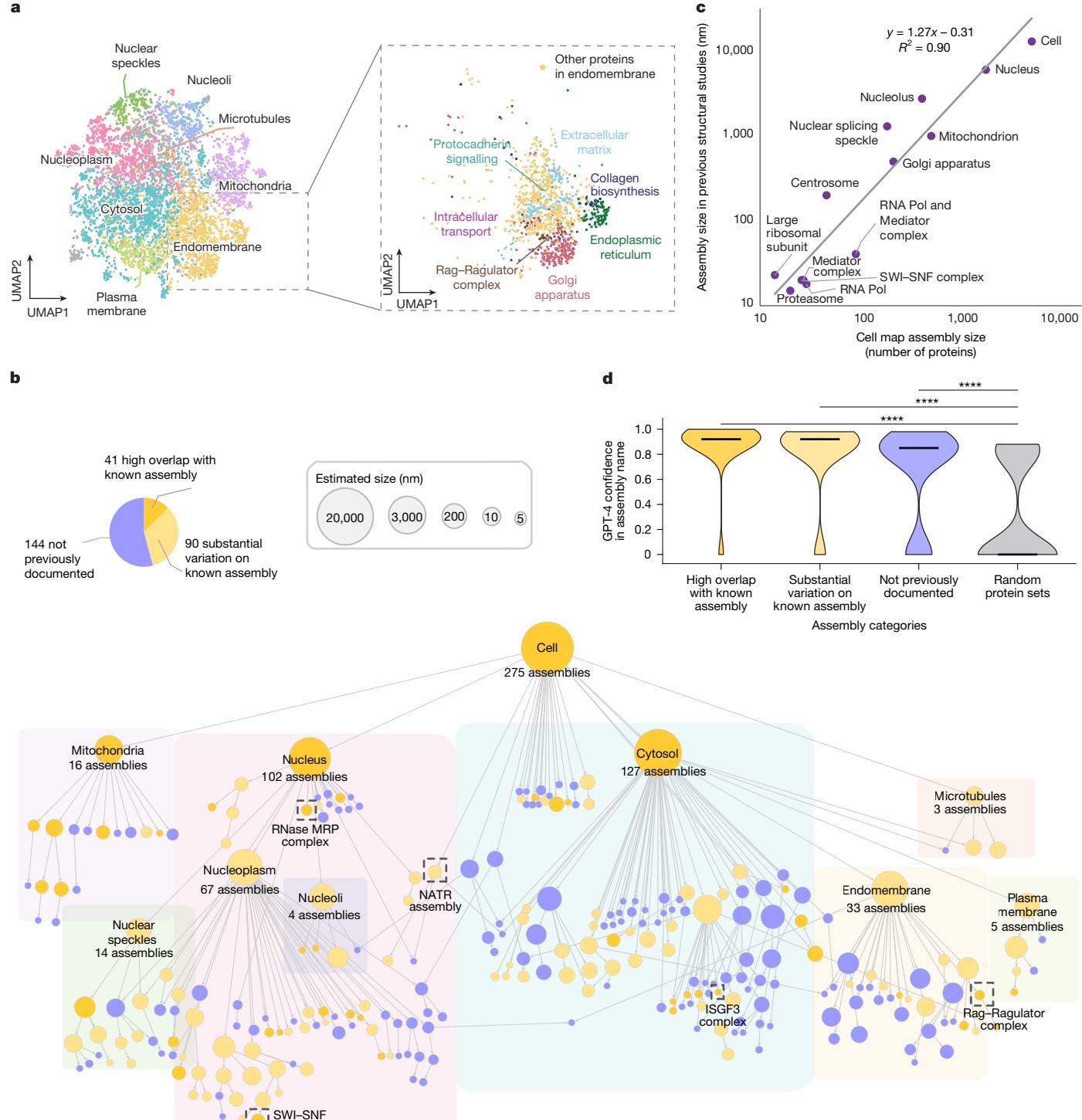

**Fig. 2 | Multiscale integrated map of a U2OS cell. a**, Multimodal embedding of proteins based on integration of AP–MS and imaging data, reduced to two dimensions using the UMAP method[56] (left). The points are proteins that are coloured and annotated on the basis of the top-level protein communities that can be resolved. Right, enlargement of the embedding, centred on the endomembrane community and its substructure. **b**, A multiscale hierarchical view of subcellular assemblies resolved in the U2OS cell map. The nodes represent assemblies, and the edges represent containment of a smaller assembly (lower) by a larger one (upper). The node size is proportional to the estimated size in nanometres. The node colour is based on three categories of overlap with known subcellular components (defined in pie chart). The dashed boxes denote assemblies described in the text and figures. **c**, Calibrating the sizes of assemblies in the cell map (number of proteins) to the physical diameters of known structures (nanometres). **d**, GPT-4 self-confidence in generating informative names for assemblies in the cell map, shown for the categories of assemblies denoted in **b** and random assemblies (grey). The distributions of confidence scores are shown as violin plots, with the thick black lines representing the median confidence in each category. The significance of differences between distributions was calculated using one-sided Mann–Whitney $U$-tests; ****$P < 0.0001$.

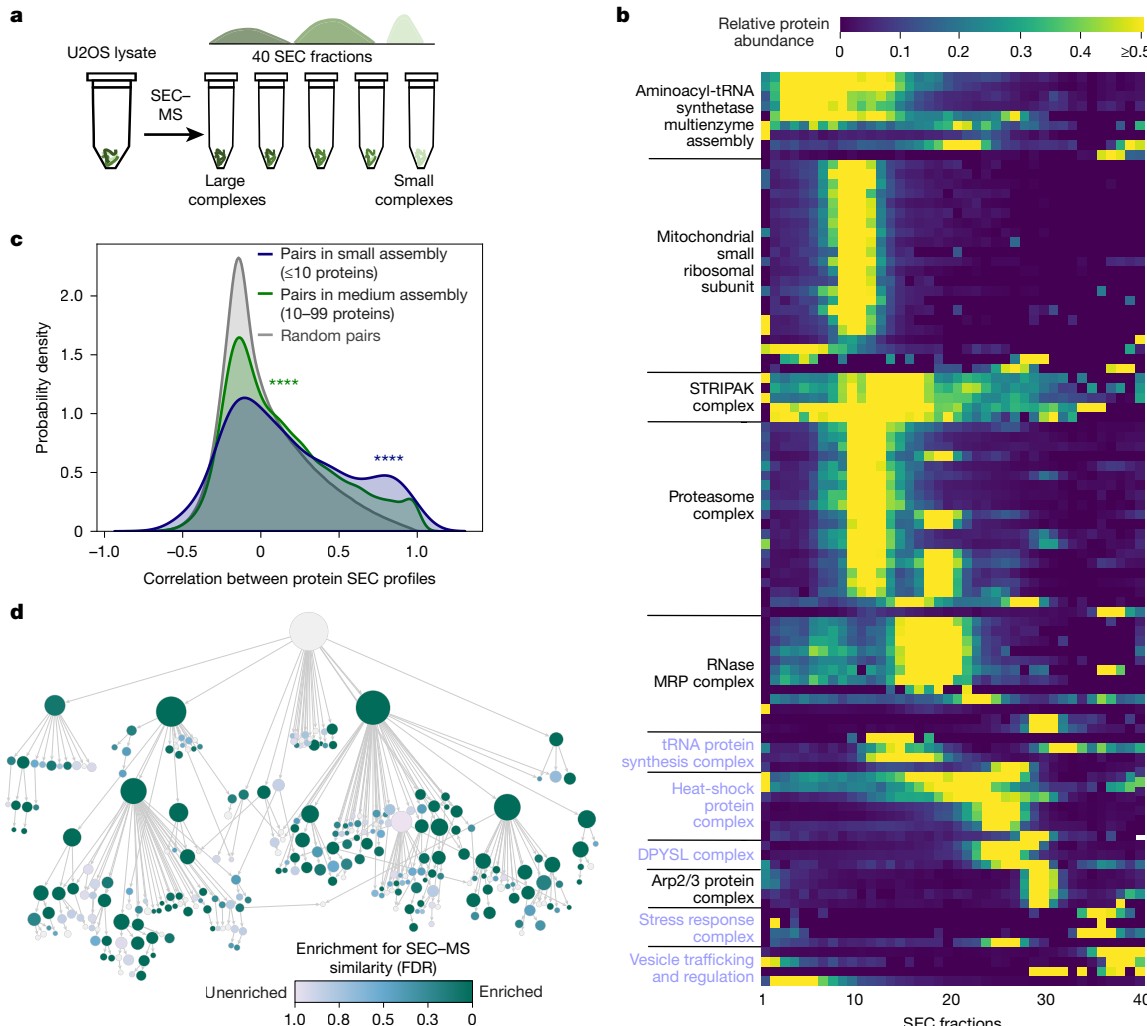

**Fig. 3 | Global analysis of assemblies with SEC–MS. a**, Overview of the SEC–MS experiment. **b**, The elution fractions (columns) for proteins (rows) in representative cell map assemblies. The intensity (colour) represents the relative protein abundance in each fraction, scaled in the range [0–1] for each protein across all fractions. Previously undocumented assemblies are indicated in purple font. **c**, The distribution of SEC–MS Pearson correlations among pairs of proteins in assemblies, shown separately for small assemblies, medium assemblies and random pairs. Significant differences from random pairs are indicated, determined using one-sided Wilcoxon rank-sum tests; ****$P < 0.0001$. **d**, Cell map with assemblies coloured on the basis of the significance of validation by SEC. The assembly colour indicates the FDR as determined using one-sided Wilcoxon rank-sum tests comparing SEC co-elution profiles for protein pairs in that assembly versus protein pairs that do not co-occur in any assembly under the root.

Among small-to-medium size assemblies of <50 proteins, we found 39 for which the SEC data had specifically corroborated the inclusion of unexpected members (Methods and Supplementary Table 4), with functions related to heat shock, stress response and vesicle trafficking.

At this stage of the study, we had interrogated U2OS cells with multimodal proteomics data; integrated these data to resolve subcellular components at multiple scales; annotated these components; and lent support to many using an independent whole-cell profiling technique. We next turned our attention from map construction to use, exploring key impacts in structural and functional biology (use cases 1–5: three-dimensional (3D) structural modelling; revealing protein function; studying cell type specificity; protein multi-localization; and interpreting tumour mutations).

## 3D structural modelling

We first explored the cell map as a platform to guide 3D structural modelling projects, interfacing with the recent advances in structure prediction enabled by artificial intelligence (AI)[30]. We used AlphaFold-Multimer[31] to predict structural models for every pair of proteins arising in the same focal protein assembly (142 assemblies of <10 proteins, 1,666 protein pairs in total; Supplementary Table 5). We noted that the estimated accuracies of these structures (AlphaFold pTM and ipTM scores; Methods) were significantly higher than expected at random, supporting that these protein pairs have direct biophysical interaction interfaces (one-sided Mann–Whitney $U$-test, $P = 2.7 \times 10^{-12}$). Particularly high structural accuracy was indicated for 161 pairs, which also received highly confident per-residue scores at the protein–protein interaction interface (Fig. 4a and Methods).

Of these high-confidence structures, 111 had not been previously documented in the Protein Data Bank (PDB). An example was a biophysical assembly identified among DPYSL2, DPYSL3 and DPYSL4, a family of phosphoproteins important for nervous system development[32]. Their initial association was validated by SEC–MS co-elution profiling (Fig. 4b), after which AlphaFold-Multimer yielded high-confidence structures for all pairwise interactions of these proteins (Fig. 4c). Additional complexes that were validated first by SEC–MS, then resolved structurally by AlphaFold-Multimer, included an interaction between TARS3, a threonyl-tRNA synthetase, and EPRS1, a member of the aminoacyl-tRNA synthetase multienzyme subsystem[33]

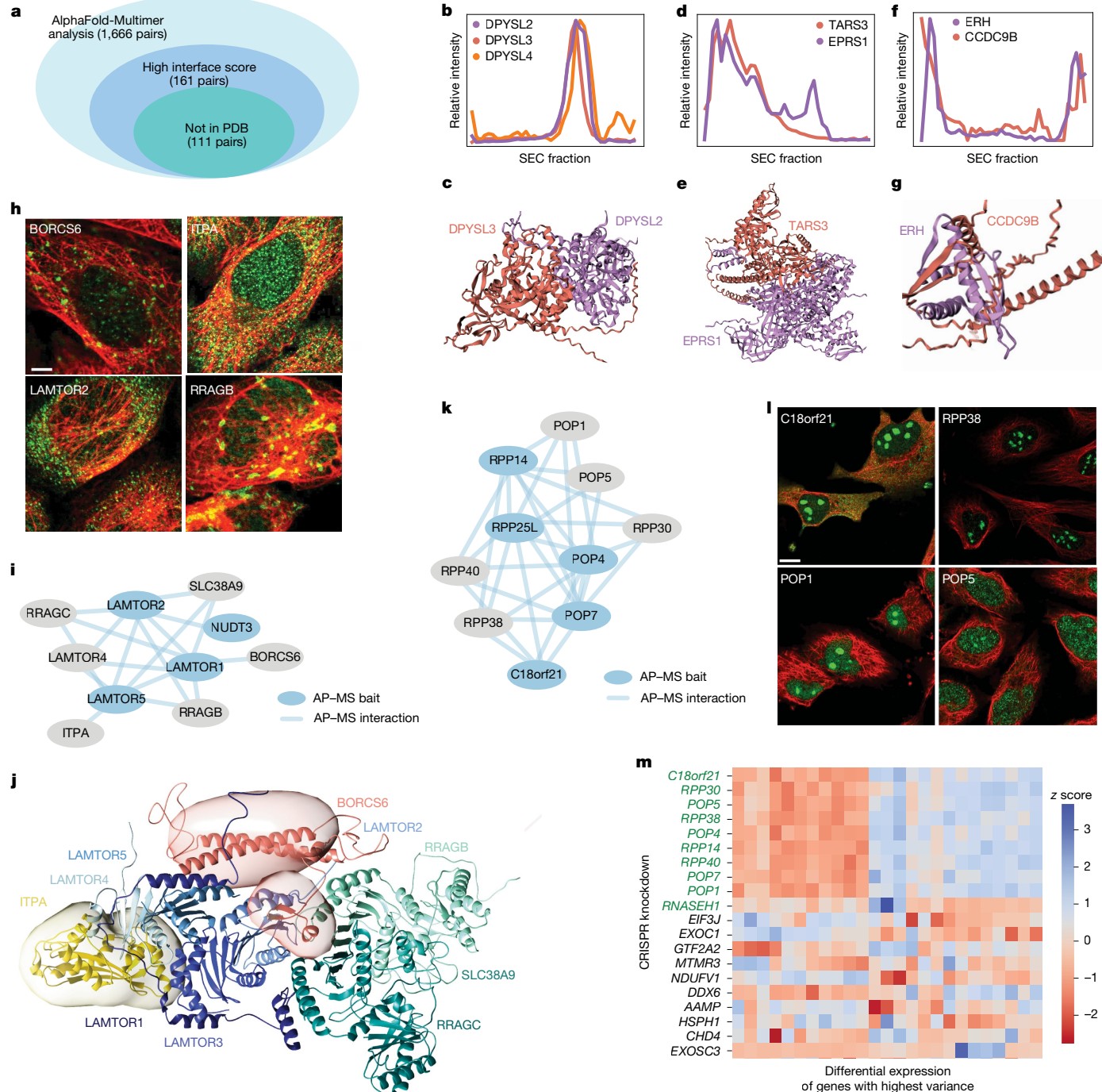

**Fig. 4 | Use of the cell map in driving studies of subcellular structure and function. a**, The results from AlphaFold-Multimer folding of heterodimeric protein complexes in the U2OS cell map. **b,c**, SEC–MS plot (**b**) and the corresponding structure (**c**) for the DPYSL2–DYSL3 heterodimer. **d,e**, SEC–MS plot (**d**) and the structure (**e**) for TARS3 and EPRS1. **f,g**, SEC–MS plot (**f**) and the structure (**g**) for ERH and CCDC9B, excluding disordered regions. **h**, IF images for representative members of the Rag–Ragulator complex. Members are immunostained (green) with cytoskeleton counterstain (red). Scale bar, 2 μm. **i**, Biophysical interaction data for the Rag–Ragulator complex. **j**, Integrative structure model of the Rag–Ragulator complex. The structural ensembles of ITPA and BORCS6 are presented as 3D localization probability densities, with surfaces transparent for visual clarity. **k**, Biophysical interaction data for representative RNase MRP complex members. **l**, IF images for four RNase MRP proteins, immunostained (green) and with cytoskeleton counterstain (red). Scale bar, 5 μm. **m**, Differential expression (z score, colour bar) after CRISPR knockdown of genes encoding the RNase MRP complex (top rows, green) versus a random sampling of other proteins. The rows represent CRISPR knockdowns, and the columns represent genes with the 20 most variable differential expression patterns across the full dataset.

(Fig. 4d,e); another example was a structure involving ERH and CCDC9B (Fig. 4f,g).

We also examined how AI predictions can be integrated with experimental structural data to create a 3D model of a large protein assembly.

We selected the Rag–Ragulator complex, which is located on the lysosomal membrane where it regulates growth signalling through the activation of the mammalian target of rapamycin complex 1 (mTORC1)[34]. The assembly that we had resolved in the cell map (Fig. 4h,i) included

members of the recombination-activating genes (RAG) and Ragulator protein families (LAMTOR1–5, RRAGA, RRAGC, SLC38A9) as well as two unexpected proteins, BORCS6 and ITPA. We built an integrative structural model[35] of this Rag–Ragulator assembly (Methods), incorporating and expanding on the base cryo-EM structure[36] (PDB: 6WJ2), AlphaFold single structure predictions of BORCS6 and ITPA, as well as pairwise AlphaFold-Multimer predictions of BORCS6 or ITPA interactions with each of the other members of the Rag–Ragulator complex. The integrated structure (Fig. 4j) indicated that BORCS6 interacts with LAMTOR2 and is proximal to LAMTOR1, LAMTOR3 and LAMTOR5. Similarly, the model supported the interaction of ITPA with LAMTOR1, LAMTOR3 and LAMTOR4. These examples illustrate how a data-driven compendium of subcellular components can identify new target protein components for downstream 3D structural studies.

## Revealing protein function

Notably, 138 proteins of previously unknown function[37] were present in the cell map, of which 24 fell in small-to-medium size assemblies of fewer than 25 proteins. Most of these assemblies had been assigned robust biological names during map curation (see above), enabling us to propose functions for their uncharacterized proteins through guilt by association (Supplementary Table 6). One such functional assignment was for C18orf21, which our cell mapping data placed robustly in the RNase mitochondrial RNA processing (MRP) complex (Fig. 4k,l). Corroborating this assignment, we observed that knockdown of *C18orf21* induces a distinct transcriptional cell state very similar to knockdowns of other MRP genes (Fig. 4m).

Expanding to proteins with some previous functional annotation, we found 951 cases in which a protein was assigned to an unexpected assembly of fewer than 25 proteins, suggesting new functional roles (Supplementary Table 6). For example, the interferon-stimulated gene factor 3 (ISGF3) complex[38], previously defined as consisting of STAT1, STAT2 and IRF9, also included dipeptidyl peptidase 9 (DPP9), a serine protease previously associated with inflammation[39]. Our AP–MS data implicated DPP9 as a potential member of this complex based on the STAT2 pull-down (Extended Data Fig. 6a) and this association was reinforced by the confocal images, which indicated similar cytosolic patterns of localization with ISGF3 proteins (Extended Data Fig. 6b). We observed that inhibition of DPP9 by 1G244 (a selective DPP9 inhibitor[40]) upregulated the canonical ISG targets of STAT transcription factors, including IFNβ1, IFNγ1 and IFNγ2, while a non-ISG control was unaffected (Methods and Extended Data Fig. 6c), suggesting that DPP9 acts to suppress the IFN response (Extended Data Fig. 6d). These examples illustrate how a data-derived reference cell map provides a substantial aid in completing the functional annotation of the human proteome.

## Studying cell type specificity

Defining a global map of a given cell type confers the potential to distinguish subcellular components that are specific to that type from those that are more widely conserved. As an initial proof of concept towards this aim, we examined each protein assembly in the U2OS cell map for evidence of shared versus distinct biophysical interaction patterns in comparison to HEK293 human embryonic kidney cells (previously characterized by AP–MS in the BioPlex 3.0 resource[22]; Methods and Extended Data Fig. 7a). Of the 258 assemblies with AP–MS data coverage in both cell types, we identified 103 that were conserved across cell types (Extended Data Fig. 7b and Supplementary Table 7). These included large assemblies, including the nucleus and cytosol, as well as small assemblies such as the spliceosome, the 9–1–1 RAD–RFC complex (Extended Data Fig. 7c,d) and components of the SNARE complex. The remaining 155 assemblies showed biophysical interaction patterns that were significantly different between HEK293 and U2OS cell types. For example, a cytosolic component named the energy

metabolism regulation complex was robustly identified in the U2OS AP–MS data, but none of the corresponding interactions were detected in HEK293 cells (Extended Data Fig. 7e). These examples illustrate how a data-driven cell map can elucidate protein assemblies that are specific or shared between cell types, providing a basis to explain different cell phenotypes and identify cell-type-specific drug targets.

## Protein multi-localization

A substantial fraction of proteins have been postulated to multi-localize, that is, to have a role in multiple subcellular assemblies or compartments[12,41]. To this point, we noted that approximately 30% of proteins in the cell map (1,520 out of 5,147) are present in more than one distinct assembly (Extended Data Fig. 8a and Supplementary Table 8). For example, XAB2, a known factor of the spliceosome and transcription-coupled repair[42], localized not only to nuclear assemblies as expected, but also to the endomembrane (Extended Data Fig. 8b). Evidence for such localizations was present in the fluorescence images as well as in the AP–MS interaction network, in which XAB2 showed strong interactions with both nuclear spliceosomal and membrane-associated stress factors (Extended Data Fig. 8c).

Moving beyond single proteins, we also investigated whether there was evidence of multiple localizations for entire protein assemblies, noting 23 that were indeed documented to multi-localize according to the U2OS cell map (Extended Data Fig. 8d,e). For example, the amyloid precursor protein (APP) complex (APP, APBA2, APBA3, APLP2, TJAP1) was clearly resolved in both the cytosol and endomembrane compartments (Extended Data Fig. 8d) on the basis of evidence from both the protein imaging and biophysical interaction modalities (Extended Data Fig. 8f,g). This finding aligns with previous studies showing that APP and its homologue, APLP2, have a role in subcellular trafficking from the endoplasmic reticulum to the cell surface[43] (with vesicular and endoplasmic reticulum localizations captured in our U2OS imaging data; Extended Data Fig. 8g). APBA2 and APBA3 are members of the X11 adaptor protein family, which is known to regulate the translocation of APP[44]. These examples illustrate how a multimodal cell map can reveal both single proteins and whole assemblies that localize to multiple subcellular compartments, suggesting pleiotropic functions.

## Interpreting tumour mutations

Determining how diverse genetic alterations disrupt common molecular machines is critical to understanding the complexity of diseases such as cancer. Towards this aim, we obtained genome-wide somatic mutation profiles for a compendium of 772 paediatric primary tumours encompassing 18 tumour types[4] (Supplementary Table 9). We then analysed these mutational profiles using the U2OS cell map, looking for mutational selection on the set of genes of an assembly as a whole (Methods). Each assembly was tested for mutation within each tumour type separately and across the entire pan-cancer cohort. While individual gene mutations are rare in paediatric cancer, with only 6 genes altered in >2% of tumours (Fig. 5a), we identified a total of 11 recurrently mutated assemblies at this same 2% threshold (Fig. 5b). For example, the *SMARCA4* SWI–SNF transcriptional activator is a well-known cancer driver that is genetically altered in 2.5% of paediatric tumours[45] (Fig. 5a), but this frequency increases to 6.0% when including coding alterations across all 13 proteins in SWI–SNF complexes (Fig. 5b). Some recurrently mutated assemblies were highly specific to certain cancer types, as was the case for an unexpected finding of frequent mutations of cell junctions in B cell lymphoblastic lymphoma (Fig. 5c). Other assemblies appeared to be under mutational selection more generally across tumours, as in the case of the nuclear pore (Fig. 5c). Cumulative across subtypes, this analysis identified a total of 21 assemblies that were recurrently mutated, suggesting positive selective pressure during tumour evolution (Fig. 5d,e and Supplementary Table 10). Mutated

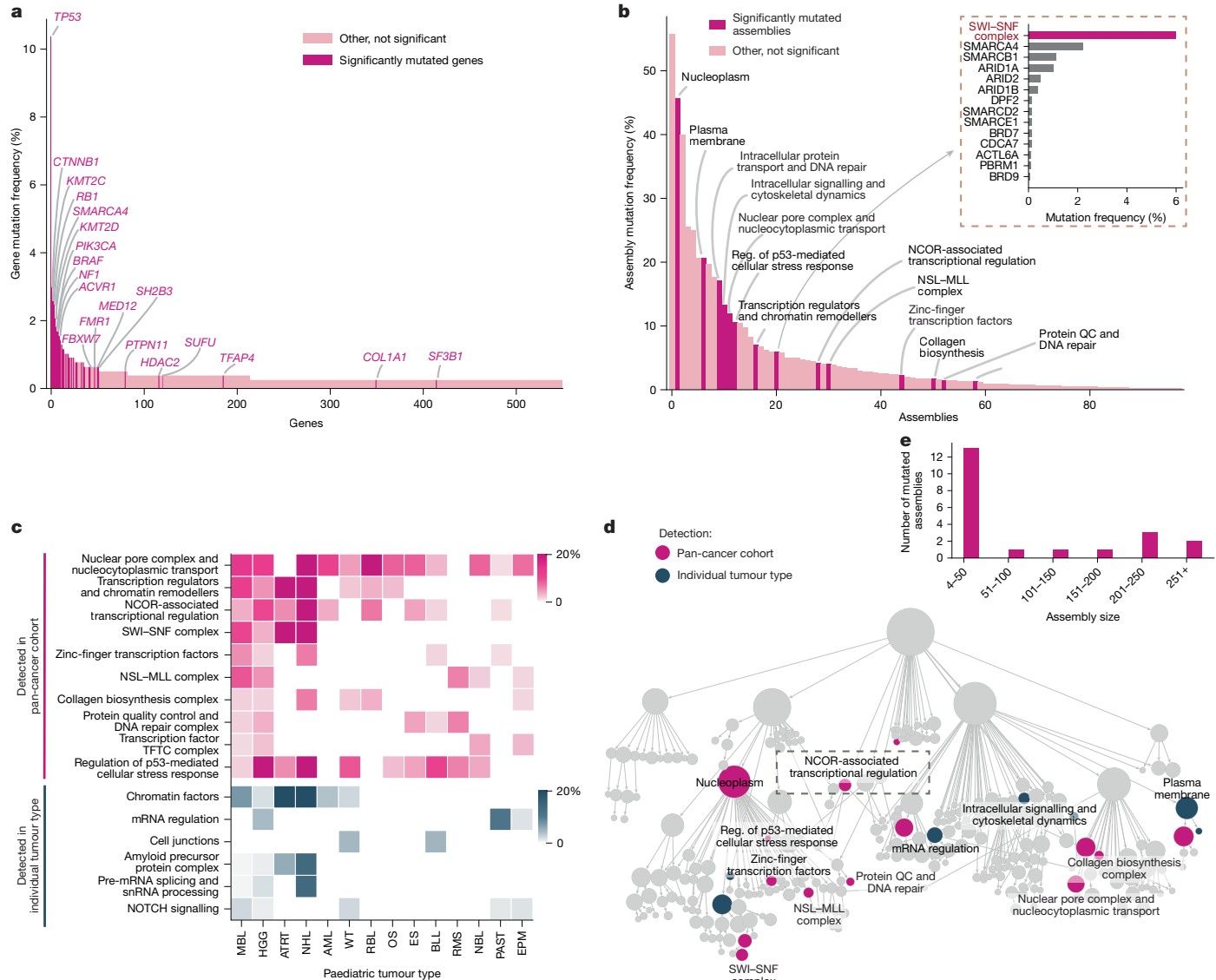

**Fig. 5 | Protein assemblies as convergence points for paediatric cancer mutations. a**, The mutation frequencies of the top 550 proteins (*x* axis), quantified in the pan-paediatric cancer cohort (*y* axis, *n* = 772 tumours). Non-silent point mutations or insertion/deletions are included. Proteins with magenta bars were previously reported as being under significant mutational pressure[4]. **b**, The mutation frequencies of 98 cancer protein assemblies (*x* axis), quantified in the same pan-paediatric cancer cohort (*y* axis). The magenta bars highlight assemblies under significant mutational pressure (FDR ≤ 0.4, Methods). Inset (top right): expansion of one of these assemblies (SWI–SNF complex) by the protein-level mutation frequencies of its members (grey bars). QC, quality control; reg., regulation. **c**, The mutation frequencies (colour gradient) of assemblies (rows) within paediatric tumour types (columns). Pink gradient is

used for recurrently mutated assemblies detected in the pan-cancer analysis. Navy gradient is used for recurrently mutated assemblies detected in individual tumour cohorts. MBL, medulloblastoma; HGG, high-grade glioma; ATRT, atypical teratoid/rhabdoid tumour; NHL, non-Hodgkin lymphoma; AML, acute myeloid leukaemias; WT, Wilms' tumours; RBL, retinoblastoma; OS, osteosarcoma; ES, Ewing's sarcoma; BLL, B cell lymphoblastic leukaemia/lymphoma; RMS, rhabdomyosarcoma; NBL, neuroblastoma; PAST, pilocytic astrocytoma; EPM, ependymoma. **d**, Cell map indicating assemblies that are under mutational pressure across the pan-paediatric patient cohort (magenta, *n* = 14) or in individual tumour cohorts (navy, *n* = 7). The assembly indicated by a dashed rectangle is further discussed in Extended Data Fig. 9. **e**, The distribution of sizes for the recurrently mutated assemblies.

assemblies were identified at all size scales but had a clear preference for small complexes of fewer than 50 proteins (Fig. 5e).

Within these assemblies, we focused on 250 putative cancer proteins, defined as proteins that are not only present in recurrently mutated assemblies but are also themselves mutated in multiple tumour samples (Methods). To further investigate a role for these proteins in cancer, we performed a large meta-analysis of transposon-based mutagenesis screens in mouse tumour models[46] (Methods and Extended Data Fig. 9a). The putative cancer proteins showed a very high degree of enrichment for genes in which transposon mutagenesis leads to tumour development (Extended Data Fig. 9b), with specific validation support

for 102 proteins (FDR < 0.3). The majority of these proteins had not been implicated in previous gene-level mutational analysis of either adult or paediatric cancer (Extended Data Fig. 9c,d and Supplementary Table 10). For example, the significantly mutated NCOR-associated transcriptional regulation assembly (Extended Data Fig. 9e) contained a total of 28 proteins, of which 16 were impacted by paediatric cancer mutations (Supplementary Table 10). Two proteins in this complex, NCOR1 and TBL1XR1, had been previously reported as cancer driver genes and shown to regulate key signalling pathways in modulating tumour growth[47,48]. Of others in this complex, we found that three validate as cancer drivers through mouse transposon mutagenesis

(GTFIRD1, NRIP1, NCOR2). We also noted that proteins in this complex show a high proclivity to phase separate (22 out of 28; Methods and Supplementary Table 3) with distinct punctae in the IF images, suggestive of nuclear condensate formation (Extended Data Fig. 9f). These findings demonstrate how knowledge of cancer protein assemblies can focus a genome analysis to increase the sensitivity of detecting cancer mutational events.

## Cell map toolkit and portal

To enable interactive exploration of the human cell map, we developed the companion Multiscale Integrated Cell visualization portal (available at http://musicmaps.ai/u2os-cellmap/), which combines a high-performance graphical web interface with the general analysis functionality of the widely used Cytoscape application[49]. The map is browsable as a tree view (that is, the hierarchy in Fig. 2b) or a cell view, in which hierarchical assembly relationships are represented as nested circles (Extended Data Fig. 10). Tables provide key information such as the proteins comprising each assembly, estimated assembly sizes in nanometres and links to confocal images. Each assembly can be selected to display its supporting subnetwork of evidence, including biophysical interactions (denoting proteins with high subcellular proximity as revealed by AP–MS pull-downs) and imaging interactions (denoting proteins with high subcellular proximity as revealed by the confocal images). Built-in search functionality is used to select and highlight assemblies that contain proteins of interest, and the platform also integrates LLM functional interpretation (Extended Data Fig. 4) to allow assemblies to be explored for insightful names and functional interpretations[3]. To facilitate continued map improvement, incorporation of new datasets, and construction of new cell maps across subtypes and disease states, we also developed the Cell Mapping Toolkit (https://github.com/idekerlab/cellmaps_pipeline), which implements the end-to-end pipeline described here as a series of Python packages complete with full user documentation. This toolkit provides a flexible and generalizable framework for cell map construction, enabling researchers to integrate and construct cell maps via multiple input modalities.

## Discussion

Although the basic sequence of the human genome has been known for over two decades[50], knowledge of how its proteins are organized within cells is still very much evolving. To advance this cause, we have developed a reference human cell map with extensive coverage of subcellular assemblies spanning four orders of magnitude (around $10^{-8}$ to $10^{-5}$ m). Achieving coverage across proteins and scales relied on at least two advances: interrogating the cell with matched proteome-wide datasets tuned to complementary types of information, and integrating these views systematically through a multimodal deep learning workflow. These advances provide a blueprint for mapping subcellular architecture that can be readily applied across human cell types and disease states. They also pave the way to expanded cell maps incorporating new modalities, such as proximity labelling, subcellular fractionation or cryo-electron tomography, as well as time-dependent measurements, such as monitoring of subcellular dynamics over a progression of cell cycle phases.

With such generality in mind, we surveyed a series of use cases representing common areas of investigation in which a global data-driven cell map can powerfully drive biological discovery. First, we examined how protein assemblies provide the starting material for 3D structural modelling, leading to the generation of high-confidence heterodimeric structures using AlphaFold (Fig. 4a and Supplementary Table 5) and a large integrative model of the Rag–Ragulator complex combining computational predictions with experimental 3D coordinates (Fig. 4h–j). A second key impact was in the study of individual proteins, in which

the cell map suggests unexpected roles for numerous proteins (Supplementary Table 6). As a proof of concept, we further investigated a role for C18orf21 in the RNase MRP complex (Fig. 4k–m) and for DPP9 in the ISGF3 complex (Extended Data Fig. 6). Other key applications were in the study of cell type specificity (Supplementary Table 7 and Extended Data Fig. 7), molecular condensates (Supplementary Table 3) and multi-localizing proteins and protein assemblies (Supplementary Table 8 and Extended Data Fig. 8). A final, critical demonstration was in decoding human genetics. By identifying patterns of genetic mutations that converge on protein assemblies (Supplementary Table 10), numerous proteins were implicated that had not been previously reported as paediatric cancer drivers (Extended Data Fig. 9c,d).

Through multimodal analysis, the human cell map presented here unifies and extends multiple ongoing efforts that have thus far progressed independently. In this respect, we found that the integration of multiple modes of data substantially broadens the sensitivity and robustness with which subcellular components can be resolved across scales (Extended Data Fig. 3). These benefits translate to real impacts in biological discovery as exhibited in the use cases. Approximately half of AlphaFold structures (47 out of 111; Supplementary Table 5) and 40% of new protein functional annotations (Supplementary Table 6) were driven by assemblies that were robustly identified only by integrating both AP–MS and imaging datasets.

A separate distinct benefit of a multimodal analysis is that, by design, it provides multiple lines of evidence for new biological findings. In a typical omics study, a single modality of data is presented and analysed with many putative findings, only a few of which can be validated or pursued at any depth. By contrast, each new finding of the U2OS cell map is derived from two complementary experimental platforms by default (AP–MS biochemical pull-downs and spatial proteomics imaging), and the systematic lines of evidence deepen further in the use cases through support from SEC–MS, AlphaFold predictions, perturb-seq and/or transposon mutagenesis. For example, the assembly of multifunctional protein ERH with RNA-binding protein CCDC9B was supported by an AP–MS interaction, image subcellular annotations, SEC–MS elution profiles (Fig. 4f) and a high-confidence AlphaFold 3D model (Fig. 4g). Such confluence of data, also seen in other recent multi-omic studies[51,52], increases the confidence in each result and provides substantial additional structural, functional and/or spatial information. This aspect pushes towards a new mode of end-to-end cell biology whereby multiple datasets are generated, integrated and simultaneously corroborated, informing a unified and foundational representation of the cell[9,53–55].

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

## Methods

### AP–MS data collection

U2OS cell cultures were processed for protein–protein physical interaction mapping by AP–MS, according to a previously described protocol developed as part of the BioPlex project[14]. U2OS cells were obtained from American Type Culture Collection (ATCC) and tested for *Mycoplasma* contamination. C-terminal HA-Flag-tagged DNA constructs targeting each of 2,174 bait proteins were constructed using clones from the human ORFeome library[57] and introduced into U2OS cells by lentiviral transfection. Baits were selected based on success in previous pull-down experiments and to ensure broad sampling of the interactome as observed previously[22]. Immobilized and pre-washed mouse monoclonal anti-HA agarose resin was incubated with cell lysates to extract protein baits and their associated protein complexes. Subsequently, these were eluted with HA peptide then reduced and digested with trypsin. Approximately 1 μg of peptide was loaded for reversed-phase liquid chromatography with a C18 microcapillary column followed by tandem MS (Thermo Fisher Q-Exactive HFX) using data-dependent acquisition selecting the top 20 precursors for MS2 analysis. Proteins were identified from the MS2 spectra using Sequest[58], filtered to 1% protein-level FDR with additional entropy-based filtering[14]. The CompPASS algorithm[59,60] was used to select high-confidence (top 2%) protein–protein interactions on the basis of the abundance of proteins in each immunoprecipitation compared with their average levels across all other immunoprecipitations. Interactions were further filtered with CompPASS-Plus at a 1% FDR[14,61]. Steps for quality control were as follows. Clones were sequence-validated as described previously[57]. AP–MS analyses required the bait protein to be detected in the Sequest results; moreover, bait proteins were required to have a higher abundance (based on spectral counting) in their own pull-down compared with the other pull-downs on the same 96-well plate. To remove under-loaded samples, we required LC–MS runs to contain a minimum of around 5,000 PSMs and about 700 proteins. Enrichment of interactions within CORUM complexes (Extended Data Fig. 1c,d; CORUM v.4.1) was computed using a one-sided binomial test, assuming background probability of interaction equal to the network's interaction density, with Benjamini–Hochberg (BH) FDR correction. CORUM complexes for each case were limited to those with at least three proteins and at least one AP–MS bait in the network. Randomized networks were constructed preserving the overall number of interactions per bait (node degrees).

### Matched protein IF imaging data

U2OS cell cultures were analysed using IF confocal imaging as part of the Human Protein Atlas project (HPA) using a previously described protocol[12]. U2OS cells were obtained from ATCC and were authenticated according to the manufacturer using morphology, karyotyping and PCR-based approaches to confirm the identity and to exclude intraspecies and interspecies contaminations. U2OS cells were seeded in 96-well glass-bottom plates and grown to a confluence of 60 to 70% at 37 °C in McCoy 5A medium, supplemented with 10% fetal bovine serum (FBS) and 5% $CO_2$ for propagation. Cells were then fixed in 4% paraformaldehyde followed by permeabilization with Triton X-100 detergent and incubated with the HPA primary antibody for the target protein, overnight at 4 °C. HPA antibodies were diluted to 2–4 μg ml⁻¹ in blocking buffer with 1 μg ml⁻¹ mouse anti-tubulin and 1 μg ml⁻¹ chicken anti-calreticulin. The next day, cells were incubated at 90 min at room temperature with secondary antibodies (goat anti-rabbit AlexaFluor 488; goat anti-mouse and goat anti-chicken AlexaFluor 647; or goat anti-rat AlexaFluor 647) diluted to 1 μg ml⁻¹ and counterstained with 4′,6-diamidino-2-phenylindole (DAPI). IF images were acquired using a Leica SP5 confocal microscope equipped with a ×63 HCX PL APO 1.40 oil CS objective. Each IF image contains four colour channels, one for the protein of interest and the other three channels for reference markers corresponding to nucleus (DAPI), microtubule (anti-tubulin antibody) and endoplasmic reticulum (anti-calreticulin antibody). Antibody quality was scored according to a standard HPA protocol (https://www.proteinatlas.org/about/antibody+validation); the highest scoring antibody per protein was selected with up to two technical replicate images.

### SEC–MS data collection

We collected a proteomic SEC–MS dataset in the U2OS cell line according to a previously described procedure[62]. U2OS cells were tested for *Mycoplasma* contamination. Three 15 cm dishes of confluent U2OS cells for each replicate ($n$ = 3) were washed and collected in ice-cold SEC buffer (50 mM KCl, 50 mM NaCH₃COO, 50 mM Tris, pH 7.2, containing 1× EDTA-free HALT protease and Thermo Fisher Scientific phosphatase inhibitor cocktail). These samples were subjected to a fractionation protocol described previously[63], with modifications. In brief, cells were lysed using a Dounce homogenizer with a tight pestle for 3.5 min on ice. Lysates were ultracentrifuged at 100,000 rcf for 15 min at 4 °C, and the supernatants were concentrated over 100 kDa molecular mass cut-off spin columns (Sartorius). A standard Bradford assay was performed to inject 600 μg of protein for each replicate into a single 300 × 7.8 mm BioSep-4000 column (Phenomenex) using SEC buffer without protease inhibitors. The samples were then separated into 40 fractions at 15 s per fraction using the 1290 Series semi-preparative HPLC (Agilent Technologies) system at a flow rate of 0.6 ml min⁻¹ at 6 °C. The collection end point was predetermined by measuring the end of the BSA standard peak, discarding anything smaller than a single BSA protein size. The resulting fraction volumes of protein were denatured by adding to a final concentration 20% (v/v) 2,2,2-trifluoroethanol (Sigma-Aldrich), reduced and alkylated[64]. Subsequently, we added an equal volume of 50 mM ammonium bicarbonate for overnight digestion with trypsin (New England Biolabs) at 37 °C. The resulting peptides were cleaned with C-18 STop And Go Extraction (STAGE) tips[65] using 40% (v/v) acetonitrile and 0.1% (v/v) formic acid in water as the elution buffer. Peptide concentrations were measured on a NanoDrop One instrument (Thermo Fisher Scientific, 205 nm, Scopes method), after which we loaded approximately 50 ng of peptides onto the TimsTOF Pro2 (Bruker Daltonics) system with CaptiveSpray source coupled to a nanoElute UHPLC (Bruker Daltonics) device using an Aurora Series Gen2 analytical column (25 cm × 75 μm, 1.6 μm FSC C18; Ion Opticks). The instrument was set to acquire in DIA-PASEF mode as previously outlined[66]. The sample batch was randomized before injection. Acquired SEC–MS data were searched on DIA-NN (v.1.8.1.0)[67] against the UniProt human sequences (UP000005640, downloaded 2 June 2023) and common contaminant sequences (229 sequences). Library-free search was enabled, using trypsin/P protease specificity and 1 missed cleavages. Other search parameters included 1 maximum number of variable modifications, N-terminal M excision, carbamidomethylation of C and oxidation of M. Peptide length ranged from 7 to 30, precursor charge ranged from 1–4, precursor $m/z$ ranged from 300 to 1,800, and fragment ion $m/z$ ranged from 200 to 1,800. Precursor FDR was set to 1%, with 0 for settings 'mass accuracy', 'MS1 accuracy' and 'scan window'. The settings 'heuristic protein inference', 'use isotopologues', 'match between run (MBR)' and 'no shared spectra' were all enabled. 'Protein name from FASTA' was chosen for the protein inference parameter along with 'double-pass mode' for neural network classifier. Robust LC (high precision) was used for the quantification strategy, RT-dependent mode for cross-run normalization, and smart profiling mode for library generation. Analyses of SEC–MS data used the protein elution profiles, defined as the protein-level quantification values reported by DIA-NN across all fractions. The similarity was calculated between the elution profiles for every pair of proteins, taking the mean Pearson correlation across the three replicates. For assessment of reproducibility across biological measurements (Extended Data Fig. 5b), we first selected

the set of proteins present in all three replicates ($n = 5{,}018$). For each replicate, we determined each protein's elution pattern, defined as the set of Pearson correlations between that protein and every other of the 5,018 proteins. We then calculated the Pearson correlation of protein elution patterns across replicates for the same protein or, alternatively, between random pairs of proteins.

### AP–MS and IF data preprocessing

Proteins were first pre-processed within the AP–MS and IF modalities separately. For the AP–MS data, the node2vec[68] Python3 implementation (https://github.com/eliorc/node2vec) was used to represent each protein $i$ as a 1,024-dimension feature vector ($\mathbf{x}_i$) based on its protein–protein interaction neighbourhood ($p = 2$, $q = 1$, walk length = 80, number of walks = 10). For the IF data, we applied DenseNet-121, a convolutional neural network pre-trained for object recognition in protein IF confocal images[69]. DenseNet-121 was used to represent each protein as a 1,024-dimension feature vector ($\mathbf{y}_i$) from the four channels of the colour image.

### Multimodal embedding overview

We developed a self-supervised multimodal machine learning model to integrate (co-embed) the AP–MS and IF protein representations into a single low-dimensional (128-dimension) embedding space (Extended Data Fig. 2a). Our model is based on the autoencoder architecture known as multimodal structured embedding[70] with modifications. Parameters of the autoencoder are trained using a two-component loss function that combines reconstruction loss and triplet (contrastive) loss. Details are provided in the 'Encoder/decoder architecture', 'Loss functions' and 'Model training' sections below.

### Encoder/decoder architecture

The separate AP–MS and IF vector inputs ($\mathbf{x}_i$ and $\mathbf{y}_i$ for each protein $i$, see above) are compressed by modality-specific encoders ($f_x$ and $f_y$) yielding 128-dimension vectors $\mathbf{a}$ and $\mathbf{b}$:

$$\mathbf{a}_i = f_x(\mathbf{x}_i)$$
$$= \text{Tanh}(\text{BatchNorm}(\text{Linear}(\text{Dropout}(\text{ELU}(\text{BatchNorm}(\text{Linear}(\text{Dropout}(\mathbf{x}_i))))))))$$

$$\mathbf{b}_i = f_y(\mathbf{y}_i)$$
$$= \text{Tanh}(\text{BatchNorm}(\text{Linear}(\text{Dropout}(\text{ELU}(\text{BatchNorm}(\text{Linear}(\text{Dropout}(\mathbf{y}_i))))))))$$

where Dropout indicates dropout layers[71]; Linear indicates linear transformation layers; BatchNorm indicates batch normalization[72]; Tanh indicates a hyperbolic tangent function; and ELU indicates an exponential linear unit function. The $\mathbf{a}$ and $\mathbf{b}$ vectors are then input to a joint encoder $f_z$ that learns the L2-normalized 128-dimension latent representation $\mathbf{z}_i$:

$$\mathbf{z}_i = f_z[\text{concat}(\mathbf{a}_i, \mathbf{b}_i)]$$
$$= \text{L2Norm}(\text{BatchNorm}(\text{Linear}(\text{Dropout}(\text{concat}(\mathbf{a}_i, \mathbf{b}_i)))))$$

Values of $\mathbf{z}_i$ constitute the self-supervised multimodal embedding used for subsequent cell map evaluation (see the 'Evaluation of embedding approaches' section below) and construction (see the 'Pan-resolution community detection' section below). For the decoder step, $\mathbf{z}$ is reverse-transformed to extract 128-dimension modality-specific features through weight matrices $w_x$ and $w_y$:

$$\mathbf{c}_i = w_x \mathbf{z}_i$$

$$\mathbf{d}_i = w_y \mathbf{z}_i$$

Finally, these features are passed to modality-specific decoders ($g_x$ and $g_y$), yielding the 1,024-dimension reconstructed inputs ($\hat{\mathbf{x}}_i, \hat{\mathbf{y}}_i$):

$$\hat{\mathbf{x}}_i = g_x(\mathbf{c}_i) = \text{Linear}(\text{Tanh}(\text{Linear}(\text{ELU}(\text{Linear}(\mathbf{c}_i)))))$$

$$\hat{\mathbf{y}}_i = g_y(\mathbf{d}_i) = \text{Linear}(\text{Tanh}(\text{Linear}(\text{ELU}(\text{Linear}(\mathbf{d}_i)))))$$

### Loss functions

To compute the reconstruction loss $R$, the ($\hat{\mathbf{x}}_i, \hat{\mathbf{y}}_i$) outputs of the autoencoder are compared to the original input values ($\mathbf{x}_i, \mathbf{y}_i$) for each modality:

$$R_x = \frac{1}{n} \sum_{i=1}^{n} \|\mathbf{x}_i - \hat{\mathbf{x}}_i\|_2$$

$$R_y = \frac{1}{n} \sum_{i=1}^{n} \|\mathbf{y}_i - \hat{\mathbf{y}}_i\|_2$$

where $n$ is the total number of proteins. The overall reconstruction loss is the sum of modality-specific reconstruction losses and a regularization term, where $\lambda_{\text{regularization}}$ is the regularization weight and $\|w\|_F$ is the $F$-norm of the matrix:

$$R = R_x + R_y + \lambda_{\text{regularization}}(\|w_x\|_F + \|w_y\|_F)$$

To compute triplet loss $T$, clustering using the Louvain algorithm[73] is performed on the ($\mathbf{a}, \mathbf{b}$) vectors of each modality (during early training clusters are defined using input ($\mathbf{x}, \mathbf{y}$) values instead; see the 'Model training' section below). This clustering defines selection functions $S_x$ and $S_y$ for each modality, with $S(i,j) = 1$ for proteins $i, j$ in the same cluster, else 0. This information is used to compute $T$ for each modality:

$$T_x = \frac{1}{m} \sum_{i \varepsilon N} \sum_{j \varepsilon N, j \neq i} \sum_{k \varepsilon N, k \neq i,j} S_x(i,j)(1 - S_x(i,k))$$
$$\times \max(D(\mathbf{z}_i, \mathbf{z}_j) - D(\mathbf{z}_i, \mathbf{z}_k) + \varepsilon, 0)$$

$$T_y = \frac{1}{m} \sum_{i \varepsilon N} \sum_{j \varepsilon N, j \neq i} \sum_{k \varepsilon N, k \neq i,j} S_y(i,j)(1 - S_y(i,k))$$
$$\times \max(D(\mathbf{z}_i, \mathbf{z}_j) - D(\mathbf{z}_i, \mathbf{z}_k) + \varepsilon, 0)$$

where $N$ is the set of all proteins, $D$ denotes the cosine distance (1 − cosine similarity), and $m$ is the total number of terms inside the summation that are greater than 0. The full loss function $L$ is a weighted sum of the reconstruction and triplet losses:

$$L = R + \lambda_{\text{triplet}}(T_x + T_y)$$

### Model training

Model parameters were trained with standard neural network learning procedures provided by Pytorch[74] v.2.0.1, based on backpropagation using the Adam stochastic gradient descent method[75]. Training occurred in three phases: (1) Over the first 200 epochs, only the reconstruction loss $R$ was used for backpropagation. (2) Over an additional 200 epochs, the full loss function $L$ was used for backpropagation, with $S_x$ and $S_y$ defined using input $\mathbf{x}, \mathbf{y}$ vectors. (3) Over a final 500 epochs of training, the full loss function $L$ was used for backpropagation, with $S_x$ and $S_y$ defined using $\mathbf{a}, \mathbf{b}$ vectors (updated every 200 epochs). Values of hyperparameters were set based on previous work[70] without fine-tuning: batch size = 64, $\lambda_{\text{regularization}} = 5$, $\lambda_{\text{triplet}} = 5$, Adam optimization learning rate = 0.0001. Triplet loss margin and dropout percentages ($\varepsilon = 0.10$, dropout = 0.25) were set based on commonly recommended values[76,77].

## Evaluation of embedding approaches

The above self-supervised embedding model was evaluated in comparison to two alternative multimodal embedding approaches: (1) simple unsupervised concatenation of the separate AP–MS and IF inputs (**x**,**y**); and (2) a random forest regression model supervised to use (**x**,**y**) to predict protein–protein semantic similarities from the Gene Ontology (June 2023 release), trained as previously described[21] (Python Scikit-learn package, fivefold cross-validation, n_estimators=1000, max_depth=30). These embedding models were each scored for their recovery of interacting protein pairs documented in three complementary reference databases: (1) high-confidence protein–protein interactions in STRING[78,79] (v.12, NDEx uuid 0b04e9eb-8e60-11ee-8a13-005056ae23aa; Extended Data Fig. 2b,e); (2) protein pairs assigned to the same CORUM[25] complex (v.4.1, NDEx uuid 764f7471-9b79-11ed-9a1f-005056ae23aa; Extended Data Fig. 2c,e); or (3) protein pairs with high functional similarity in a genome-wide CRISPR-perturbation/mRNA sequencing screen (perturb-seq[80]; Extended Data Fig. 2d,e). Here, high functional similarity was defined as the top 1% of protein pairs by Pearson correlation between the profiles of mRNA transcriptional changes induced by CRISPR disruptions of the two proteins (see the 'Analysis of perturb-seq data' section below).

## Pan-resolution community detection

The cosine similaritiy between the multimodal embeddings for each pair of proteins was used to generate a series of protein–protein proximity networks in which edges were defined from the most similar 0.2, 0.3, 0.4, 0.5, 1.0, 2.0, 3.0, 4.0, 5.0 or 10.0% pairs, respectively, yielding 10 networks in total. Pan-resolution community detection was performed in each of these networks using the Hierarchical community Decoding Framework (HiDeF; https://github.com/fanzheng10/HiDeF)[81], with a persistence threshold ($k$) of 10 and a maximum resolution (maxres) of 80, with other parameters kept at the default settings. HiDeF identifies protein communities at different resolutions and represents their hierarchical relationships as a directed acyclic graph (DAG). In this DAG, the nodes represent communities and the directed edges ($a \rightarrow b$) represent that community $a$ contains community $b$. The DAG was refined by assigning parent–child containment relationships between assemblies with containment index ≥ 75% and removing redundant systems with Jaccard index ≥ 90% with parent systems. This final DAG defines the cell map referenced in Fig. 2b.

## Estimation of assembly diameter

A subset of 13 protein assemblies was selected from the cell map corresponding to assemblies with a known physical diameter documented in the literature (Supplementary Table 2). Linear regression was used to fit the $\log_{10}$-transformed diameter (nm, $y$) against the $\log_{10}$-transformed size of the assembly (number of proteins, $x$): $y = 1.27x - 0.31$. This linear equation was then used to estimate a diameter $\hat{y}$ for each assembly in the map. A 95% prediction interval (PI) was estimated on the basis of the standard error as follows:

$$\log_{10}\text{PI} = \hat{y} \pm (t_{(1-\alpha/2, n-2)} \times \text{s.e.}(\hat{y}))$$

with $t$ determined by the Student's $t$-distribution ($t = 2.2$ with d.f. = $n - 2$, $n = 13$ components). The s.e. is the standard error between predicted and measured sizes, calculated as follows:

$$\text{s.e.}(\hat{y}) = s_e\sqrt{1 + \frac{1}{n} + \frac{(x - \bar{x})^2}{\sum_{i=1}^{n}(x_i - \bar{x})^2}}$$

where, $s_e = \sqrt{\frac{\sum_{i=1}^{n}(y_i - \hat{y})^2}{n-2}}$. Relevant to Fig. 2c.

## Evaluation of assembly robustness

The robustness of protein assemblies was evaluated using a statistical jackknifing approach, as described previously[21]. A random set of 10% of proteins was removed before multimodal embedding (see the 'Multimodal embedding overview' section above); integration and community detection were then performed using the same parameters described in the 'Model training' and 'Pan-resolution community detection' sections. This randomization procedure was repeated 300 times to create a set of jackknifed hierarchies. The robustness of each assembly from the original hierarchy was then calculated as the fraction of all jackknifed hierarchies that contained at least one matching assembly, defined as substantial and significant overlap between the protein sets representing the target and the match (Jaccard index ≥ 40% and hypergeometric statistic FDR < 0.001). To assess the dependence of each assembly on the protein imaging data, we created a dataset with AP–MS features randomized (1,024-dimension random vectors sampled from a normal distribution) before the statistical jackknifing procedure, and the robustness of each assembly was computed as described above. For assessing the dependence of each assembly in the map on the AP–MS data, a reciprocal procedure was performed in which image embeddings were randomized. Relevant to Extended Data Fig. 3.

## Annotation of cell map assemblies

The cell map was annotated by first aligning assemblies with the GO cellular component branch (June 2023 release), CORUM (4.1 human complexes) or HPA (v.23) resources. Each of these cell biology resources defines a list of protein sets (GO terms, CORUM complex, HPA subcellular localizations), referred to here as components. Hypergeometric tests were performed for each assembly versus each component in the resource, and the FDR was determined using BH correction. The results were tabulated for all assembly–component pairs with Jaccard index ≥ 10% and hypergeometric statistic FDR < 0.01 (Supplementary Table 1). Assemblies in the map were labelled as high overlap with known assembly (Jaccard index ≥ 50% for at least one of the three resources); substantial variation on known assembly (Jaccard index < 50% for all three resources and 20% ≤ Jaccard index < 50% for at least one of the resources); or not previously documented assembly (Jaccard index < 20% for all three resources) based on this enrichment analysis. We also used our recently developed Gene Set AI (GSAI) pipeline[3] to guide the GPT-4 model[27] (v.gpt-4-1106-preview) to annotate assemblies with <1,000 proteins (Extended Data Fig. 4a). This approach uses a well-engineered prompt that follows the chain-of-thought[82] and one-shot[83] strategies to query GPT-4 for a descriptive name, a confidence score and a detailed reasoning assay of the protein members from each assembly. One example is shown in Extended Data Fig. 4c, and the full result for each assembly is available in Supplementary Table 1. Literature references are provided by a separate GPT-4 based citation module developed in the previous study[3] (Extended Data Fig. 4b) to aid in interpretability. The citation model extracts gene symbols and functional keywords from each paragraph of the LLM-generated analysis text; these are used to construct and execute PubMed queries that search titles and abstracts. The returned publications are prioritized based on relevance and the number of matching genes in their abstracts. Finally, a separate GPT-4 instance is asked to evaluate whether the top three publication titles and abstracts provide supporting evidence for factual statements in the original analysis paragraph, selecting those that satisfy this requirement as references. To evaluate the reproducibility of GPT-4 naming (Extended Data Fig. 4d), we performed the GSAI pipeline for five additional replicate runs of GPT-4 and calculated the semantic similarity between the assembly names generated in each of these runs versus the original run. Similarity was computed using the SapBERT model[84] from huggingface (cambridgeltl/SapBERT-from-PubMedBERT-fulltext) using the

transformers package[85] (v.4.29.2). Assemblies that were not named by the original run were eliminated from the reproducibility test.

## Biological condensate analysis

To analyse the cell map for biological condensates, we used three resources: IUPred3.0[86], a sequence-based predictor of protein disorder; FuzDrop[87], a sequence-based predictor for the ability of a protein to drive condensate formation; and CD-Code[88], a database containing proteins known to participate in biological condensates. IUPred3.0 predicts the probability of each amino acid in a sequence as being disordered. Proteins containing a contiguous sequence of amino acids >30 residues, where each amino acid has a >50% chance of being disordered, were annotated as likely disordered. FuzDrop assigns a probability of a sequence driving phase separation, which we thresholded at >60% to annotate a protein as 'likely phase-separated'. Finally, we searched for each gene's UniProtID in CD-Code (accessed 31 May 2023) under '*Homo sapiens*', enabling us to annotate a protein as 'associated with known condensates'. We used a hypergeometric test to assign statistical significance ($P < 0.01$) to each protein assembly that was enriched in proteins that were likely disordered, likely phase-separated, or associated with known condensates. Assemblies that were significant in one of these three analyses were considered possible biological condensates (Supplementary Table 3).

## Validation of protein assemblies and subunits by SEC–MS data

For the set of proteins in each assembly, we determined the Pearson correlation in SEC–MS elution profiles for all pairs of these proteins (see the 'SEC–MS data collection' section). This similarity distribution was then compared to a null distribution (all pairs of proteins not in any common U2OS assembly, that is, assigned to root node only) using a one-sided Wilcoxon rank-sum test with BH correction (Fig. 3d and Supplementary Table 4). Assemblies with FDR < 5% were considered validated. A similar analysis was performed using PrinCE[89] (https://github.com/fosterlab/PrInCE) scores to rank protein pairs rather than Pearson correlations, with PrinCE run using the default parameters. We found that 90 assemblies were validated at 5% FDR in the complementary analysis using PrInCE, including 70 assemblies validated by both Pearson correlation and PrinCE similarity measures (Supplementary Table 4). For validation of unexpected protein subunits within assemblies, for each assembly <50 proteins, 'unexpected proteins' were defined as those not included in the best matching cellular component from any of three cell biology resources (GO, CORUM, HPA; see the 'Annotation of cell map assemblies' section above). For each unexpected member, its SEC–MS elution profile was compared against all other proteins in the assembly using Pearson correlation; this similarity distribution was compared to the null distribution as described above to compute an FDR. Unexpected proteins with FDR < 5% were considered validated (Supplementary Table 4).

## AlphaFold-Multimer analysis

All pairs of proteins in small assemblies (<10 proteins) were selected for AlphaFold-Multimer analysis. AlphaFold-Multimer was run on each pair using localcolabfold (https://github.com/YoshitakaMo/localcolabfold) with the default settings[90]. Sequences were acquired from the complete human protein UniProt FASTA file (UP000005640, reviewed sequences, downloaded 11 September 2023). For each predicted heterodimeric structure, we calculated a weighted average between the predicted template modelling score (PTM, an estimate of the similarity between the predicted and ground truth structures) and the ipTM score (the pTM score modified to score the interfaces across different proteins)[31]:

$$\text{model score} = 0.8 \times \text{ipTM} + 0.2 \times \text{pTM}$$

We calculated the median score out of five independent models generated per protein pair. A null score distribution was generated by repeating this score computation for pairs of proteins drawn randomly from those pairs that were not part of the same small assembly (<10 proteins as above). This null distribution was used to calculate an FDR for actual protein pair scores, selecting a cut-off of 30% corresponding to a weighted PTM score of 0.39. Pairs were further evaluated for the presence of a confident interface residue (within 10 Å of the other protein and plDDT score > 80). Relevant to Fig. 4a.

## Integrative structure modelling of the Rag–Ragulator complex

A structural model of the Rag–Ragulator community was computed by using an integrative modelling approach[35,91-93], proceeding through the standard four stages[35,91,94] as follows. (1) Gathering input information: the Rag–Ragulator model in the cell map included LAMTOR1 through LAMTOR5, RRAGA, RRAGC, SLC38A9, BORCS6, NUDT3 and ITPA. An integrative model was computed based on the SLC38A9–RagA–RagC–Ragulator comparative model (PDB: 6WJ2 template)[36], AlphaFold[30] predictions for BORCS6 and ITPA, and pairwise AlphaFold-Multimer predictions[31] for BORCS6 or ITPA versus all other members of the Rag–Ragulator complex. One-hundred AlphaFold-Multimer models were generated for each pair and evaluated using FoldDock[95]. The model excluded NUDT3 because AlphaFold-Multimer did not produce high-confidence models of NUDT3 and other Rag–Ragulator components according to FoldDock. (2) Representing subunits and translating data into spatial restraints: the components of the Rag–Ragulator community were represented as rigid bodies. Alternative models were ranked through a scoring function corresponding to a sum of terms, each one of which restrains some aspect of the model based on a subset of input information. The spatial restraints included a binary binding mode restraint on the position and orientation of pairs of proteins as derived from ensembles of AlphaFold-Multimer predictions, connectivity restraints between consecutive pairs of beads in a subunit and excluded volume restraints between non-bonded pairs of beads. (3) Configurational sampling to produce an ensemble of structures that satisfies the restraints: the initial positions and orientations of rigid bodies and flexible beads were randomized. The generation of structural models was performed using replica exchange Gibbs sampling, based on the Metropolis Monte Carlo algorithm[96]. Each Monte Carlo step consisted of a series of random translations of flexible beads and random translations and rotations of rigid bodies. (4) Analysing and validating the data and ensemble structures: model validation[93,97] included selection of the models for validation; estimation of sampling precision; estimation of model precision; and quantification of the degree to which a model satisfies the information used to compute it. The above four-step modelling protocol was scripted using the Python Modelling Interface (PMI) package, a library for modelling macromolecular complexes based on the open-source Integrative Modelling Platform (IMP) package v.2.18 (https://integrativemodeling.org)[91]. The configuration of the rigid Rag–Ragulator complex, ITPA protein and the two BORCS6 domains was computed by minimizing the violations of the spatial restraints implied by the input information, using IMP[91]. Relevant to Fig. 4j.

## Analysis of perturb-seq data

The K562 day-8 perturb-seq dataset[80] was acquired at https://gwps.wi.mit.edu (BioProject: PRJNA831566). This dataset provides single-cell transcriptional profiles for 9,867 distinct gene knockouts, which underwent filtering based on the following criteria: (1) gene knockout corresponds to a protein in our U2OS cell map; (2) gene knockout has efficient on-target mRNA reduction of >30%; (3) gene knockout induces a strong transcriptional phenotype defined by ≥20 differentially expressed genes at a significance of $P < 0.05$ on the basis of the Anderson–Darling test followed by BH correction. This filtering process resulted in a list of 1,289 gene knockouts. The functional cell states due to each of these perturbations were represented using the mean-normalized differential expression profile. Relevant to Fig. 4m and Extended Data Fig. 2d,e.

### Analysis of DPP9 inhibition
U2OS cells were seeded in triplicate at 300,000 cells per well in a six-well plate (two biological replicates). The next day, cells were treated with 1G244, a DPP9 inhibitor (HY-116304, MedChem Express) at the indicated concentrations for a total of 6 h. After treatment, The medium was aspirated and washed once with ice-cold PBS. Cells were collected in 500 μl of cold TRIzol reagent (15596026, Invitrogen) using a cell scraper. 100 μl of chloroform was added to the TRIzol lysate and vortexed for 20 s followed by a 3 min incubation at room temperature. The homogenate was centrifuged at 10,000$g$ for 18 min at 4 °C. A total of 200 μl of aqueous phase was removed with a pipette and transferred to a new Eppendorf tube. An equal volume of 100% ethanol was slowly added to the aqueous phase and mixed by gentle pipetting. The entire sample was transferred to an RNeasy Mini spin column placed in a 2 ml collection tube (74104, Qiagen). The rest of the extraction was carried out according to the Qiagen RNeasy protocol. 2 μg of RNA per sample was reverse-transcribed according to the iScript cDNA Synthesis Kit protocol (1708890, Bio-Rad, interferon beta 1: Hs01077958_s1; interferon gamma 1, Hs00194264_m1; interferon gamma 2, Hs00988304_m1; non-ISG–18S, 4333760T; and *GAPDH*, Hs0275889q_g1). qPCR was carried out in triplicates in a 96-well plate according to the TaqMan Fast Advanced Master Mix protocol (4444557, Thermo Fisher Scientific) on a CFX96 Touch Real-Time PCR Detection System from Bio-Rad. The expression levels were compared against a housekeeping gene (*GAPDH*), and the relative expression levels were compared against the DMSO control. Relevant to Extended Data Fig. 6.

### Analysis of conservation of U2OS assemblies in a second cell type
We downloaded the AP–MS BioPlex v3 network from NDEx (uuid 6b995fc9-2379-11ea-bb65-0ac135e8bacf), which provides high coverage of human protein interactions in a second cell type, HEK293 cells (14,033 proteins, 127,732 protein–protein interactions). Node2vec was used to represent the interaction pattern of each protein in this HEK293 network (see the 'AP–MS and IF data preprocessing' section). The cosine similarity in interaction patterns was then computed for all protein pairs (separately for HEK293 and U2OS). For the set of proteins included in each U2OS assembly, the distribution of pairwise protein similarities in HEK293 were compared to those in U2OS cells using the two-sided Mann–Whitney $U$-test. This test was translated to an effect size using Cliff's delta[98]; assemblies with Cliff's delta ≥ 0.5 were considered to be increasingly U2OS-specific whereas those with Cliff's delta < 0.5 were considered to be increasingly conserved. Relevant to Extended Data Fig. 7; in Extended Data Fig. 7b, Cliff's delta scores of <0 are set to 0.

### Multi-localization analysis
For each protein, we identified its terminal locations in the cell map hierarchy, defined as assemblies (hierarchy nodes) where the protein appeared but was absent in all subassemblies (child nodes). We then counted the number of unique paths from these terminal locations to the root of the hierarchy (root node). Proteins with multiple distinct paths to the root were classified as multi-localized, indicating their presence in different branches of the cell map. Multi-localized assemblies were identified as assemblies with more than one parent node in the hierarchy. Relevant to Extended Data Fig. 8.

### Pre-processing of paediatric cancer mutational profiles
Data were obtained from a pan-paediatric cancer study[4] of 914 individual patients with cancer aged under 25 years (study ID: pediatric_dkfz_2017, downloaded from cBioPortal[99,100]). We selected the following types of non-silent somatic mutation events: 'Frame_Shift_Del', 'Frame_Shift_Ins', 'In_Frame_Del', 'In_Frame_Ins', 'Missense_Mutation', 'Nonsense_Mutation', 'Nonstop_Mutation', 'RNA', 'Splice_Region', 'Splice_Site' and 'Translation_Start_Site'. A total of 772 primary tumour samples, spanning 18 cancer types, were in the resulting list (Supplementary Table 9). We recorded the number of tumours in the pan-paediatric cohort, as well as each individual tumour cohort, in which each gene was observed to have at least one somatic mutation event ($N_{(g,\mathrm{obs})}$). Moreover, we calculated the expected number of mutations for each gene in the pan-paediatric cohort ($N_{(g,\mathrm{exp})}$) using the default setting of MutSigCV v.1.4, as described in a previous study[101]. For expected mutation counts for individual cancer cohorts, we down-scaled the pan-paediatric cancer $N_{(g,\mathrm{exp})}$ based on the proportion of patients (for example, 44 patients with Wilms' tumours (WT) account for 5.7% of the pan-paediatric cohort, so $N_{g,\mathrm{exp,WT}} = 0.057 \times N_{g,\mathrm{exp,pan\text{-}paediatric}}$). Finally, the corrected log mutation count of each gene ($M_g$) for each cohort was calculated as:

$$M_g = \log_2(\max(N_{(g,\mathrm{obs})} - N_{(g,\mathrm{exp})}, 0) + 1)$$

### Statistical identification of recurrently mutated assemblies
We applied a previously described statistical model, HiSig[101] (https://github.com/fanzheng10/HiSig), to calculate the mutation selection pressure on assemblies with the default parameter settings. HiSig implements linear regression (with L1 lasso regularization) of the mutation count against the organization of proteins in assemblies. We calculated an empirical $P$ value by comparing the mutational selection on assemblies against 10,000 randomly permuted assignments of proteins to assemblies. The FDR was calculated by BH correction. Recurrently mutated assemblies were selected on the basis of FDR ≤ 0.4. Assembly-level mutation frequencies were calculated from the number of distinct patients who carried at least one mutated protein in the assembly. Tumour types with fewer than 15 patients were excluded from analysis, as were mutated assemblies with >50 mutated proteins.

### Validation of cancer driver genes
Genes mutated in more than one patient with cancer and located in the significantly recurrent mutated assemblies (see above) were defined as putative cancer proteins. We obtained a large collection of transposon-based mutagenesis screens in mice from the Candidate Cancer Gene Database (CCGD)[46] (http://ccgd-starrlab.oit.umn.edu/index.html, downloaded on 26 March 2024). This database consists of a total of 72 studies with mouse transposon insertion mutagenesis screens across 13 tumour categories (Extended Data Fig. 9a). We determined the number of studies in which a gene was disrupted by transposon insertion in mice tumours. Mutated genes in cancer assemblies were designated positives (genes expected to have high study counts because they are mutated), and all other genes were designated negatives (genes not expected to have high study counts). We calculated the kernel density estimation (KDE) for the mutated genes in cancer assemblies and other genes in the cell map using the stat.gaussian_kde function from the Python package scipy (v1.7.3). The area under the KDE curves was integrated using the trapz function from Python package numpy (v.1.21.6). The FDR was then computed as the ratio of the area under the curve for false positives (Area$_{\mathrm{FP}}$) to the total area under the KDE curve representing both false positives and true positives (Area$_{\mathrm{TP}}$), mathematically shown as: $\mathrm{FDR} = \frac{\mathrm{Area}_{\mathrm{FP}}}{\mathrm{Area}_{\mathrm{FP}} + \mathrm{Area}_{\mathrm{TP}}}$. We specified the minimum number of screens reporting a gene at 4 ($x \geq 4$), corresponding to FDR = 0.28, as the threshold cut-off for validated cancer drivers (Extended Data Fig. 9b). Adult cancer driver genes were collected from the TCGA Pan-Cancer Atlas[102]; significantly mutated genes in the pan-paediatric cancer cohort were collected from refs. 4,103. These genes were defined as known cancer genes in Extended Data Fig. 9c,d.

### Running the cell mapping toolkit
The Cell Mapping Toolkit (https://github.com/idekerlab/cellmaps_pipeline) implements a series of Python packages to execute the end-to-end pipeline described herein. Specific packages include steps

for processing the protein imaging and biophysical interaction datasets (cellmaps_imagedownloader, cellmaps_ppidownloader), embedding the input modalities (cellmaps_image_embedding, cellmaps_ppi_embedding), integrating the modalities (cellmaps_coembedding), constructing the hierarchical cell map (cellmaps_generate_hierarchy) and annotating the cell map with known resources such as GO (cellmaps_hierarchyeval). Each package is pip-installable and is linked to complete user documentation hosted at ReadTheDocs (https://cellmaps-pipeline.readthedocs.io/). A step-by-step guide is provided at the GitHub repository.

## Statistics and reproducibility

Statistical tests were performed using SciPy[104] with BH multiple-testing correction where appropriate. Statistics involving comparison between two data distributions were calculated using Mann–Whitney U-tests or Wilcoxon rank-sum tests (Figs. 2d, 3c,d and Extended Data Figs. 2b–d, 7b, 9b). Statistics for assessing the enrichment of proteins or protein pairs were calculated using hypergeometric tests (Fig. 2b and Extended Data Fig. 3) unless stated otherwise. The SEC–MS data were reproduced in three biological replicates. The IF stainings were reproduced in at least two different cell lines in HPA (Fig. 4h,l and Extended Data Figs. 6b, 7d, 8c,g, 9f). The qPCR experiment for DPP9 was repeated for two biological replicates and three technical replicates each (Extended Data Fig. 6c).

## Reporting summary

Further information on research design is available in the Nature Portfolio Reporting Summary linked to this article.

## Data availability

The Multiscale Integrated Cell web portal (musicmaps.ai/u2os-cellmap) provides links to all major data and derived resources associated with this study, including AP–MS protein interactions, protein IF images, SEC data and the online interactive U2OS cell map. The U2OS cell map is available at https://ndexbio.org under uuid f693137a-d2d7-11ef-8e41-005056ae3c32. Protein assemblies in the cell map are also available at the European Bioinformatics Institute (EBI) Protein Complex Portal (https://www.ebi.ac.uk/complexportal) with the query CLO:0009454. The AP–MS protein interaction data are available at https://ndexbio.org under uuid 95bc75d5-d1d1-11ee-8a40-005056ae23aa. In addition to its release here, the U2OS protein interaction network will be included as part of the upcoming BioPlex[105] v.4.0 database release (E.L.H. et al., manuscript in preparation). AP–MS raw MS files are available at MassIVE under the identifier MSV000097168. The entire image dataset is included in the Human Protein Atlas v23 release. SEC–MS raw MS files and search results are available at the Proteome Xchange under the identifier PXD052362. All structural models are available at the ModelArchive Database (https://modelarchive.org) with the identifiers ma-idk-u2osmap and ma-m5og4. Other public databases and resources used in this study include Gene Ontology (June 2023 release; https://geneontology.org), CORUM (v.4.1 release, https://mips.helmholtz-muenchen.de/corum/), UniProt *Homo sapiens* proteome (accessed 2 June and 11 September 2023; https://uniprot.org), STRING interactome (v.12; NDEx uuid: 0b04e9eb-8e60-11ee-8a13-005056ae23aa), OpenCell interactions (https://opencell.czbiohub.org/download), CD-CODE condensate database (accessed 31 May 2023; https://cd-code.org), FuzDrop (dataset S7 in ref. 87), Protein Condensate Atlas (supplementary dataset 8 in ref. 29), K562 day-8 perturb-seq dataset (https://gwps.wi.mit.edu), HEK-293 BioPlex v.3.0 (NDEx uuid: 6b995fc9-2379-11ea-bb65-0ac135e8bacf), paediatric cancer mutation data (https://www.cbioportal.org/study/summary?id=pediatric_dkfz_2017) and transposon-based mutagenesis screens from the Candidate Cancer Gene Database (http://ccgd-starrlab.oit.umn.edu/index.html; downloaded 26 March 2024).

## Code availability

Open-source software for cell map construction (Cell Mapping Toolkit) has been released as a series of Python PyPI packages at GitHub (https://github.com/idekerlab/cellmaps_pipeline) and is also linked through the Multiscale Integrated Cell web portal (https://musicmaps.ai/u2os-cellmap).

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

**Acknowledgements** We acknowledge funding from Schmidt Futures (T.I., E.L.), the Bridge2AI Program (NIH Common Fund; OT2 OD032742; T.I., E.L., A.S.), the Cancer Cell Map Initiative (NCI Center for Cancer Systems Biology; U54 CA274502; T.I., E.L., A.S.), the Cytoscape (5U24HG012107; T.I.) and the Network Data Exchange programs (5U24CA269436; T.I., D.P.), the Canada Foundation for Innovation and Genome BC (374PRO, L.J.F.), Knut and Alice Wallenberg Foundation (2021.0346 E.L.), Göran Gustafsson Foundation (E.L), Stanford Institute for Human-Centered AI and Param Hansa Philanthropies, and NIH grants R01GM083960 and P41GM109824 (A.S.). We acknowledge funding for the BioPlex project from NHGRI U24 HG006673 (S.P.G., J.W.H., E.L.H.), Third Rock Ventures (S.P.G., J.W.H., E.L.H.), Google Ventures (S.P.G., J.W.H., E.L.H.), Interline Therapeutics (S.P.G., E.L.H.) and Xaira (S.P.G., E.L.H.).

**Author contributions** L.V.S., M.H., E.L. and T.I. designed the study. L.V.S., M.H., G.Q., X.Z., T.L., A. Pal, A.P.L., Y.Q., P.Z., E.L., and T.I. developed ideas for data analyses. L.V.S. and M.H. implemented computational methods and analyses. R.B., A.C., I.G., T.L., W.L., A. Palar, E.P., L.V.S., M.H., D.P., I.E., A.S., E.L. and T.I. annotated the hierarchy. A. Pal, N.S., I.E. and A.S. designed and performed structural modelling. E.L.H., L.P.V., J.W.H. and S.P.G. generated and analysed the AP–MS data. K.-M.M. and L.J.F. generated and analysed the SEC–MS data. D.T., K.L. and N.M.M. generated qPCR data. L.V.S., C.C., J.L. and D.P. designed and implemented the cell map toolkit. K.O., D.P. and J.C. designed and implemented Cytoscape Web. L.V.S., M.H., E.L. and T.I. wrote the manuscript with input from all of the authors.

**Competing interests** T.I. is a co-founder, advisor and holder of equity for Data4Cure and Serinus Biosciences, and he is an advisor and shareholder for Ideaya BioSciences. The terms of these arrangements have been reviewed and approved by UC San Diego in accordance with its conflict of interest policies. E.L. is an advisor for and has equity interest in Cartography Biosciences, Element Biosciences, Santa Ana Bio, Pixelgen Technologies and Moleculent. The terms of these arrangements have been reviewed and approved by Stanford University in accordance with its conflict of interest policies. The BioPlex project has been supported by Interline Therapeutics and Xaira Therapeutics (S.P.G. and E.L.H.). J.W.H. is a co-founder for Caraway Therapeutics (a subsidiary of Merck) and is a scientific advisory board member for Lyterian Therapeutics. S.P.G. is on the scientific advisory board for Thermo Fisher Scientific, Cell Signaling Technology and Frontier Medicine. E.L.H. is a consultant for Matchpoint Therapeutics, Flagship Ventures and Calico Life Sciences. The other authors declare no competing interests.

**Additional information**
**Correspondence and requests for materials** should be addressed to Edward L. Huttlin, Emma Lundberg or Trey Ideker.

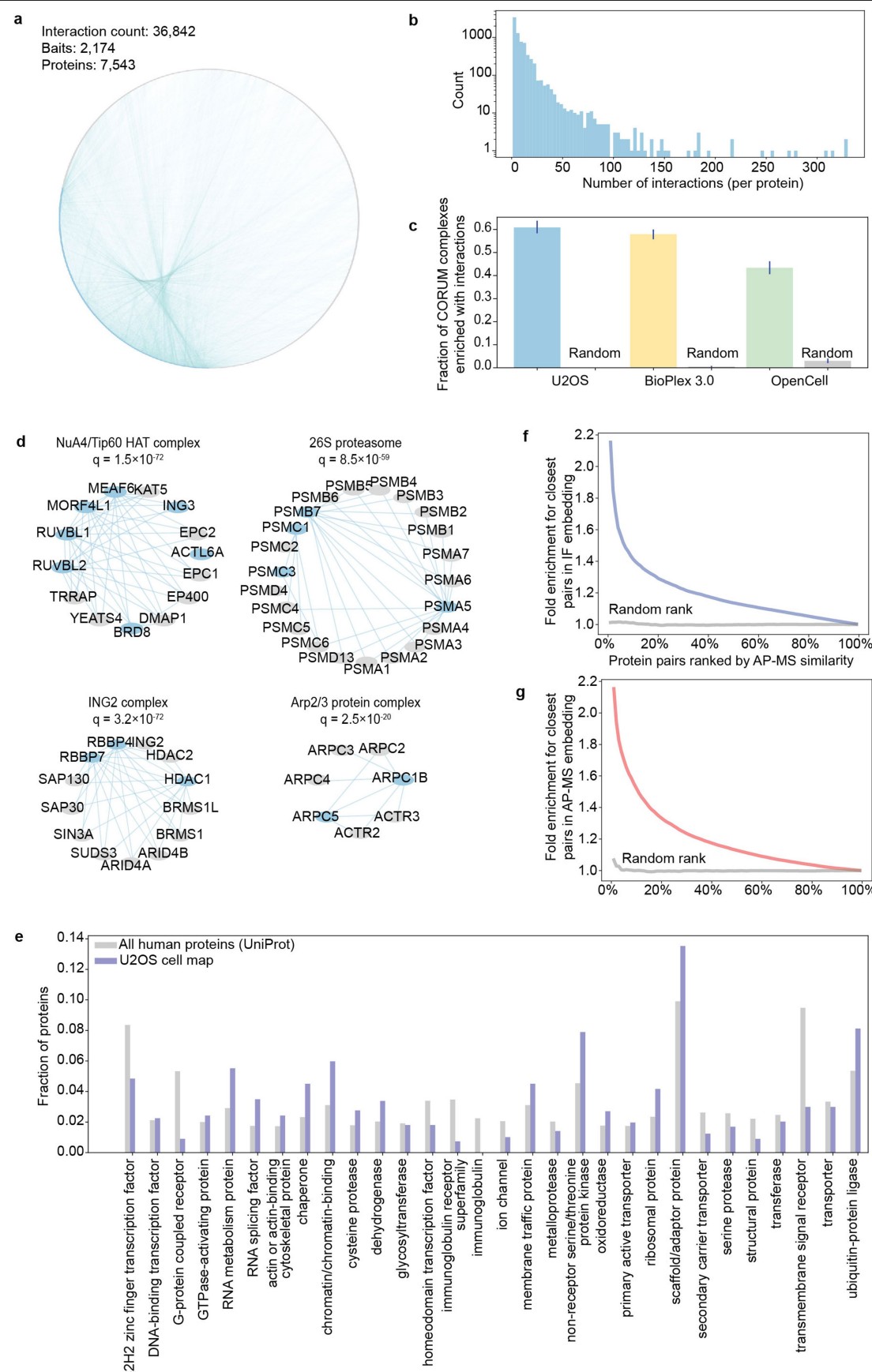

**Extended Data Fig. 1 |** See next page for caption.

**Extended Data Fig. 1 | Data quality assessment. a)** Complete network of U2OS protein-protein interactions as measured by AP-MS. **b)** Histogram of number of interactions per protein. **c)** Fraction of CORUM complexes significantly enriched (1% FDR, **Methods**) for AP-MS interactions measured in USOS (left) alongside interactions ascertained in two previously published AP-MS networks for other cell lines[22,51] (middle and right). Error bars denote 95% confidence intervals. **d)** Interaction networks for select CORUM complexes, with FDR q-values as per panel (**c**). Blue nodes denote bait proteins and grey nodes denote prey proteins. **e)** PANTHER[106] classifications of protein function (top 30 largest classes by number of proteins), shown for proteins covered by U2OS cell map in comparison to the entire human proteome (UniProt, downloaded September 11, 2023). **f)** Protein pairs ranked by cosine similarity in AP-MS features enrich for the most similar protein pairs (top 1% in the immunofluorescent protein image features and **g)** vice versa.

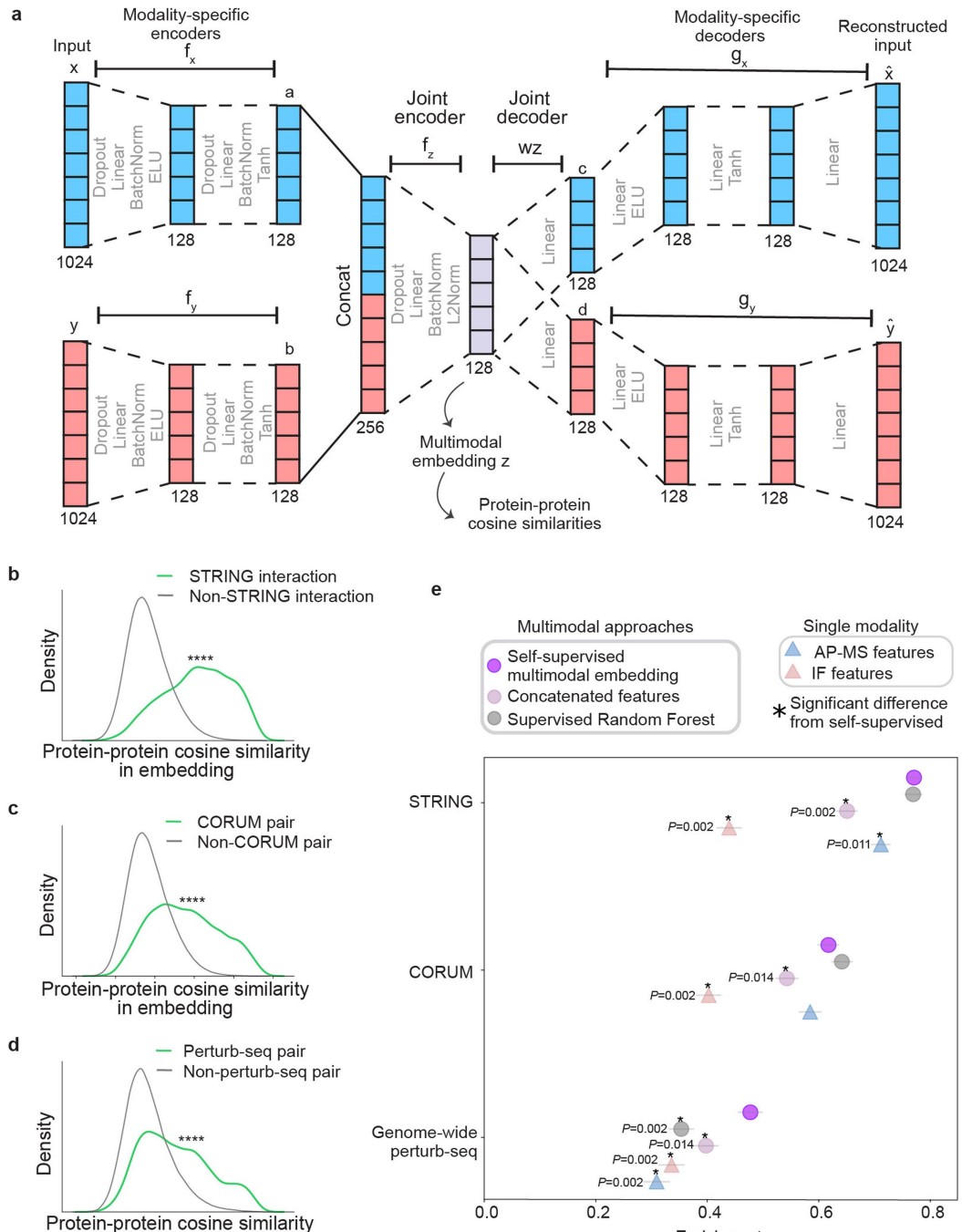

**Extended Data Fig. 2 | Self-supervised embedding of multiple data modalities.**
**a)** Architecture of self-supervised multimodal embedding model. Columns of squares represent feature vectors, with the dimensionality written just below each column. Regions enclosed by dotted lines represent neural networks with layers described. Protein coordinates in the joint multimodal embedding (z) are used for computing pairwise protein-protein similarities in subsequent panels (cosine similarity function). **b)** Distribution of similarities shown for protein pairs with a 'high-confidence interaction' denoted in the STRING database (green) in comparison to all other protein pairs (grey). **c)** Similar to (b) but for protein pairs in the same CORUM complex. **d)** Similar to (b) but for protein pairs that yield highly similar transcriptional profiles (top 1% pairs) when genetically disrupted by CRISPR, drawn from a recent perturb-seq

functional genomics study[80]. **** denotes significant difference, $p < 0.0001$ by one-sided Wilcoxon rank-sum test. **e)** Different protein embedding approaches (coloured points, **Methods**) are evaluated by their degree of enrichment (x-axis) across orthogonal functional and physical interaction resources (y-axis, resources from panels b-d above). Supervised Random Forest trained using the Gene Ontology (**Methods**). Enrichment computed using Cliff's Delta (1,000 samplings of 1,000 protein pairs with replacement, **Methods**) yielding values in range [–1,1], with positive values indicating enrichment above random expectation. Error bars denote standard deviations across 1000 bootstrap resamplings with the center at the mean. * denotes significant difference in comparison with self-supervised multimodal embedding results (two-tailed $p < 0.05$ across bootstrap resamplings).

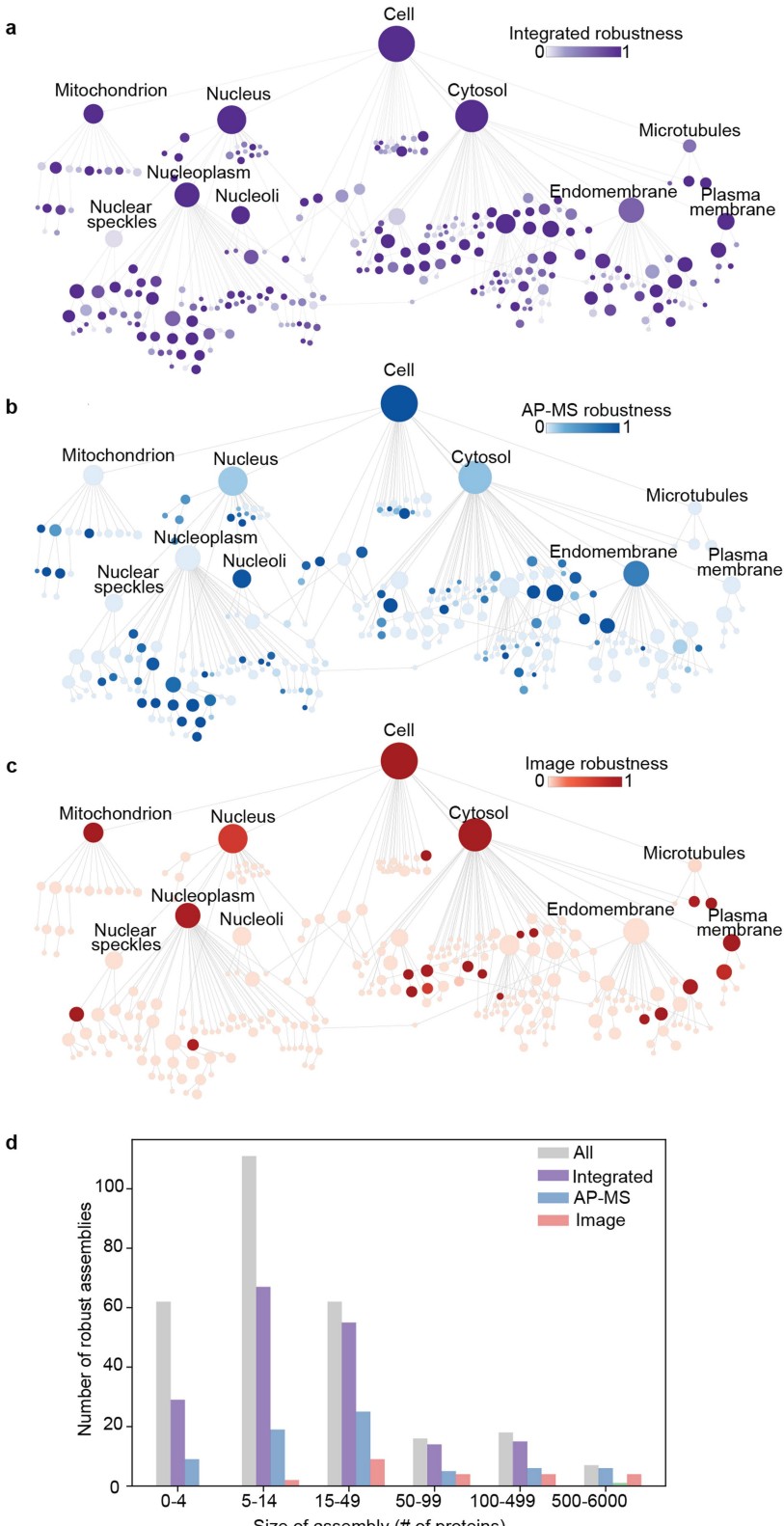

**Extended Data Fig. 3 | Robustness of cell map assemblies. (a-c)** Cell maps are coloured by assembly robustness, measured as the fraction of 300 jackknife resamplings where an assembly was recovered (**Methods**). Three panels show maps built using **a)** both imaging and AP-MS data, **b)** AP-MS data only, or **c)** imaging data only. **d)** Number of robust assemblies (recovered in >50% jackknife resamplings) versus size of assembly in number of proteins. Grey bars denote total number of assemblies in each size category.

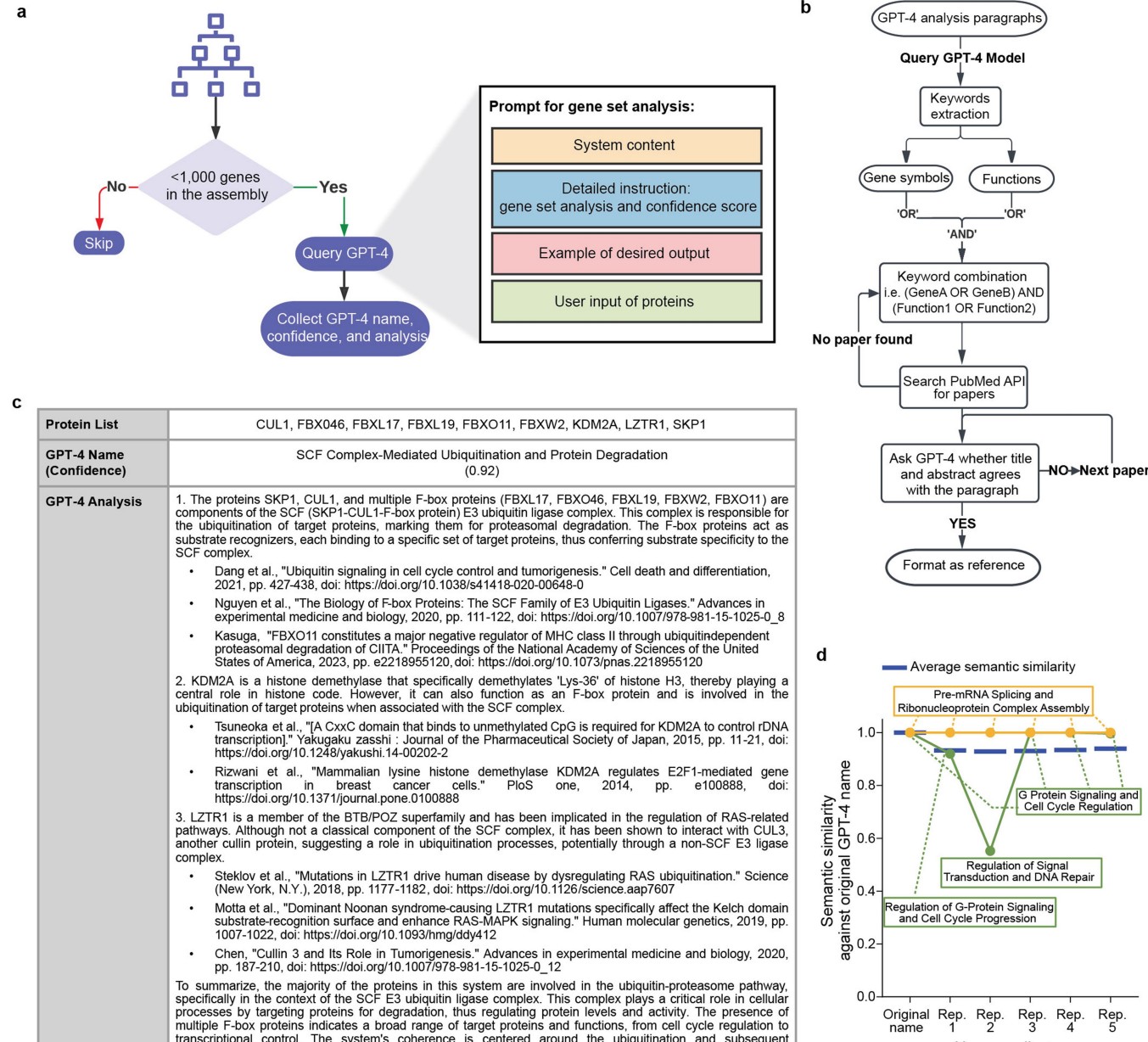

**Extended Data Fig. 4 | Annotation of protein assemblies with GPT-4. a)** Left: Annotation workflow extended from Hu et al.[3], in which the cell map is used to query GPT-4 for a descriptive name, a confidence score and a supporting rationale. Right: Composition of prompt used for GPT-4 query. **b)** Schematic of GPT-4 assisted citation module. GPT-4 is asked to provide gene symbol keywords and functional keywords separately. Multiple gene keywords and functions are combined and used to search PubMed for relevant paper titles and abstracts in the scientific literature. **c)** Example assembly with GPT-4 name and supporting analysis paragraphs with citations generated from citation module (see panel b). **d)** Semantic similarity of the original GPT-4 name given an assembly vs. the name assigned in each of five replicate GPT-4 runs. Results for two example assemblies are shown (yellow and green points), one of which is named identically across replicates (yellow) and one of which shows variation (green). The average performance over all U2OS assemblies (n = 271, excluding assemblies with more than 1000 proteins) is shown in dark blue.

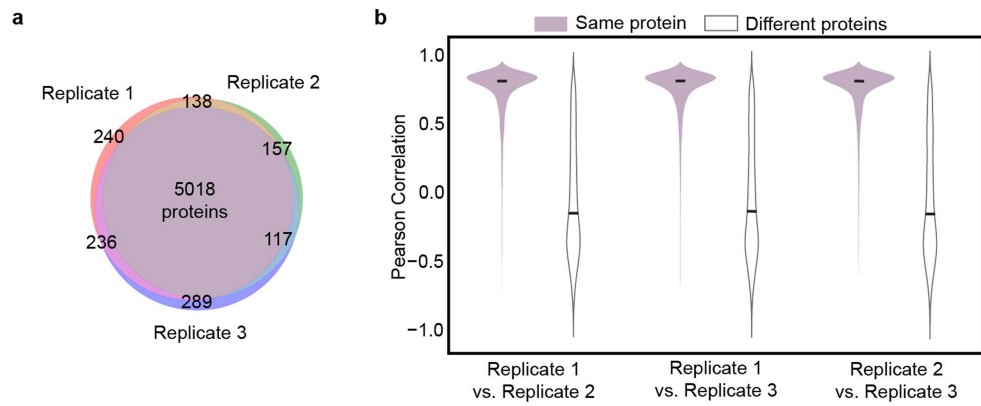

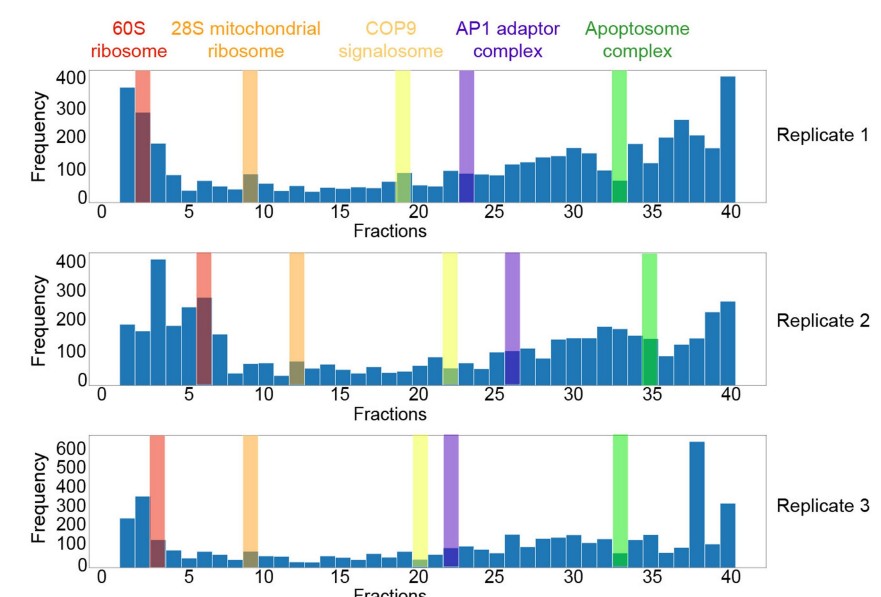

**Extended Data Fig. 5 | Quality assessment of SEC-MS dataset. a)** Overlap of protein identifications across three biological replicates. **b)** Violin plots showing the distribution of the Pearson correlation between replicate measurements of each protein's elution pattern (purple, n = 5018) vs. random pairings of different proteins across replicates (white, n = 5018), with thick black lines representing the median. **c)** Histogram of number of proteins with maximum intensity in each elution fraction for replicate 1 (top), replicate 2 (middle), and replicate 3 (bottom). Select CORUM complexes are highlighted at the median maximum intensity of proteins in the complex.

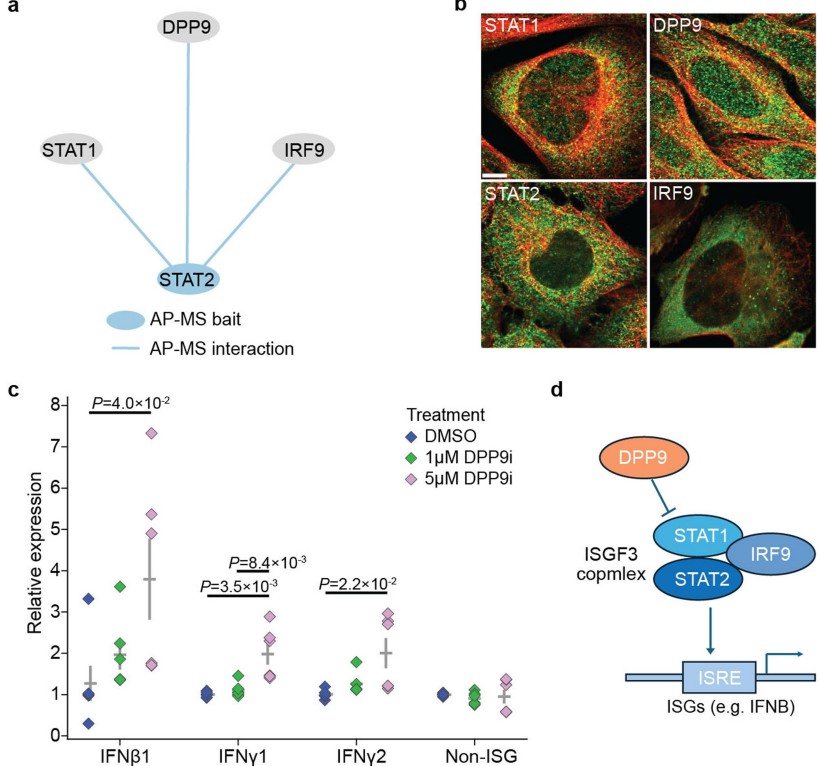

**Extended Data Fig. 6 | DPP9 association with STAT interferon signalling.**
**a)** Interaction data for the ISGF3 complex. **b)** Immunofluorescence images for the ISGF3 complex. Members immunostained (green) with cytoskeleton counterstain (red). Scale bar, 3 µm. **c)** Relative mRNA expression level of IFNβ1, IFNγ1, IFNγ2, and negative control (Non-ISG 18S) upon DPP9 inhibition, separate samples. Expression levels normalized to DMSO control. Points (n = 6, 2 biological replicates and 3 technical replicates each) denote replicate measurements. Light grey whiskers represent mean ± SE. Significance (p-values) are determined by a two-sided student's t-test. **d)** Canonical function of ISGF3 complex with putative upstream activity of DPP9.

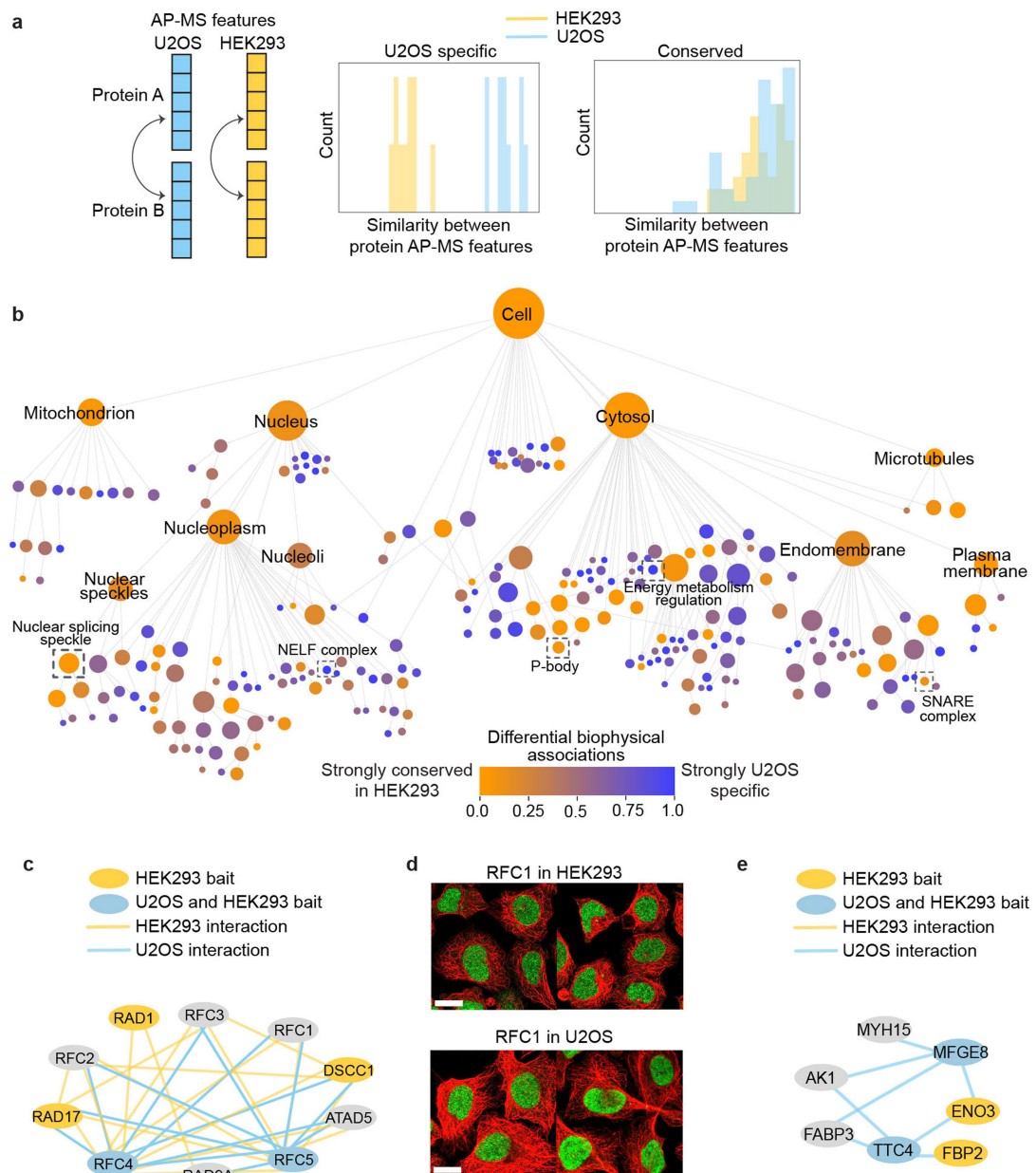

**Extended Data Fig. 7 | Comparison of protein assemblies across U2OS and HEK293 cell lines. a)** Schematic of the approach for determining conserved vs. U2OS-specific protein assemblies in the cell map. For each assembly, the cosine similarities are determined between protein AP-MS features for U2OS and HEK293, separately. The U2OS similarities are then compared to the HEK293 similarities with a two-sided Mann-Whitney U-test. **b)** U2OS cell map (see Fig. 2b), where assembly colour indicates effect size (Cliff's Delta). 18 assemblies that did not have sufficient data in both cell lines were removed. Dashed boxes denote examples of strongly conserved and strongly U2OS-specific assemblies. **c)** Biophysical interaction data for 9-1-1 RAD-RFC complex; edge colour signifies presence in HEK293 (orange) or U2OS (blue) interaction networks. **d)** Immunofluorescence images for RFC1 (orange) in HEK293 cells (top) or U2OS cells (bottom), with cytoskeleton counterstain (red). Scale bar, 5 μm. **e)** Biophysical interaction data for the Energy metabolism regulation complex; edge colour signifies presence in HEK293 (orange) or U2OS (blue) interaction networks.

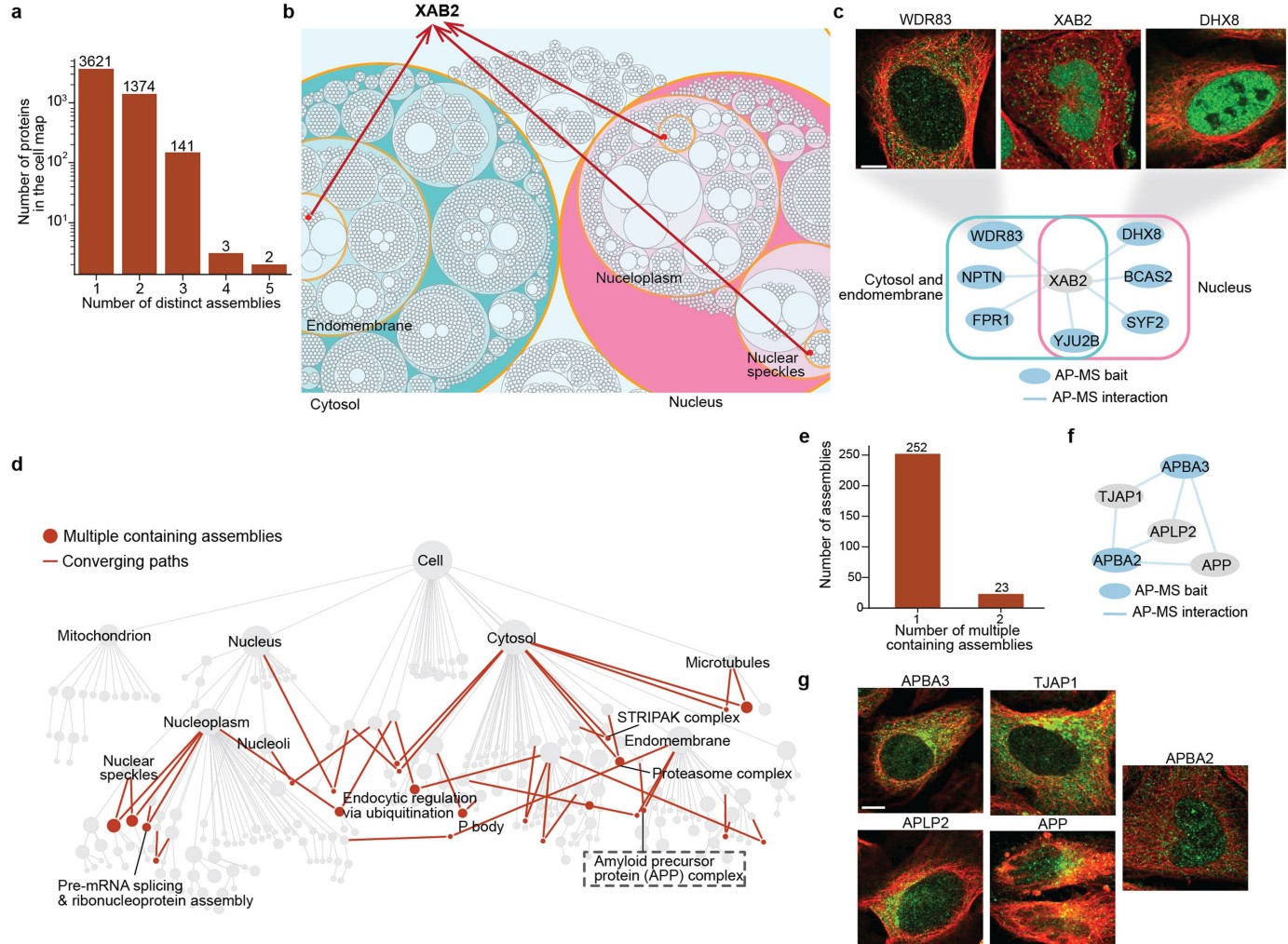

**Extended Data Fig. 8 | Analysis of multi-localized proteins and assemblies.**
**a)** Number of distinct assemblies per protein, defined as the number of distinct paths to the root of the U2OS cell map hierarchy (shown in Fig. 2b). **b)** Tripartite localization of XAB2 in the cell map visualized in circle-packing mode. XAB2 is highlighted by red circle and the cell map assemblies in which it participates are highlighted with orange borders. Cytosol and nucleus are filled as blue and pink respectively. **c)** Top: Immunofluorescence images of XAB2 (middle) and representative interacting partners in cytosol (WDR83, left) or nucleus (DHX8, right). These proteins are immunostained (green) with cytoskeleton

counterstain (red). Scale bar, 2.5 µm. Bottom: Biophysical interaction partners of XAB2 with cell map localizations in cytosol and membrane (turquoise box) or nucleus (pink box). **d)** Cell map coloured to indicate multi-localized assemblies (red nodes). The multiple containing assemblies are indicated in each case (red edges). Dashed box denotes the assembly detailed in text and in panel f–g. **e)** Number of assemblies with single (left) versus multiple (right) localizations. **f)** Biophysical interaction data for Amyloid Precursor Protein (APP) complex. **g)** Immunofluorescence images for proteins in APP complex, immunostained (green) with cytoskeleton counterstain (red). Scale bar, 2.5 µm.

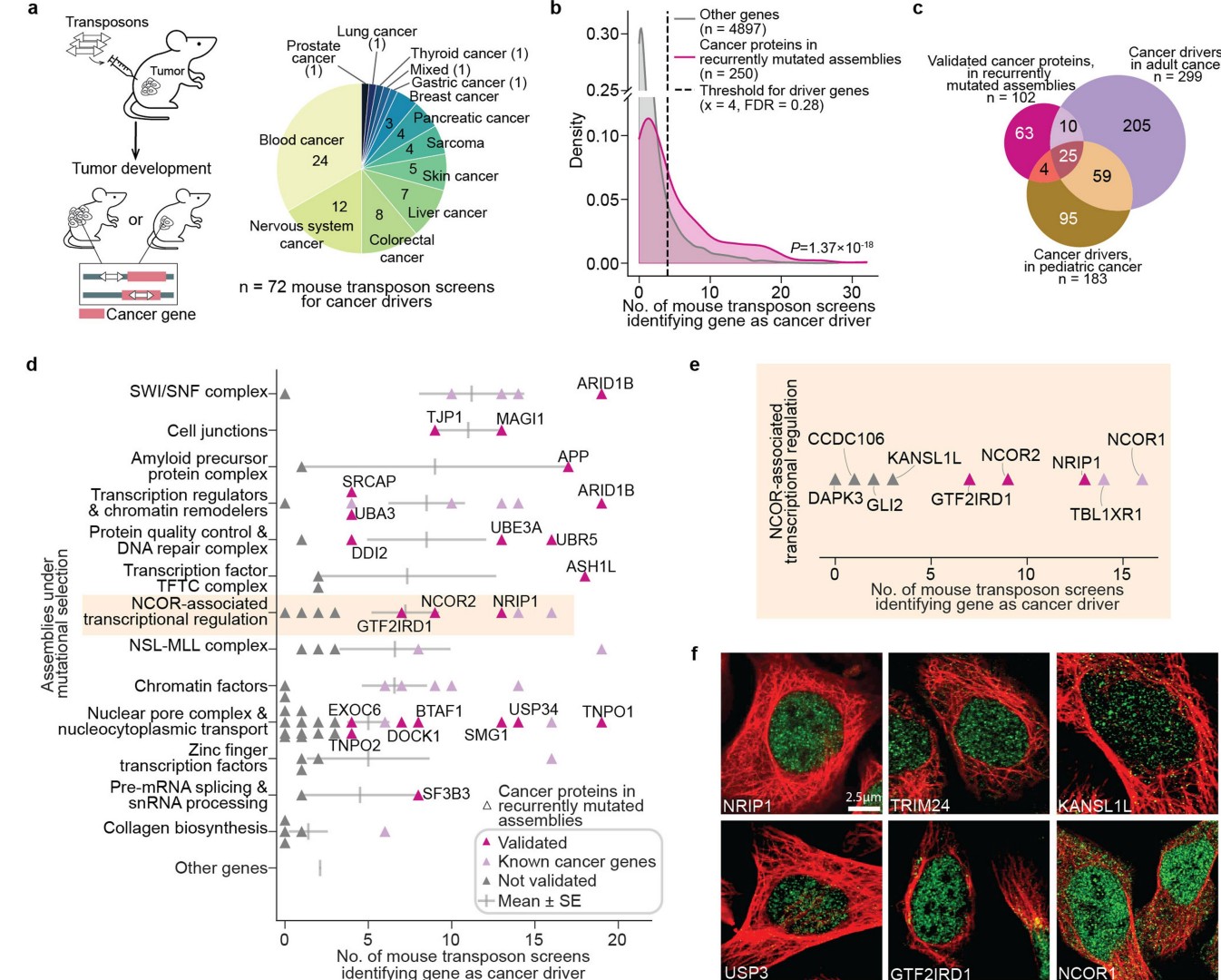

**Extended Data Fig. 9 | Association of protein assemblies with tumour growth via transposon mutagenesis. a)** LEFT: Schematic overview of transposon-based genetic screens in mouse tumour models[46]. RIGHT: Number of screens by cancer type. **b)** Distribution of the number of transposon screens identifying each human gene as a cancer driver. Separate curves show cancer proteins in recurrently mutated assemblies (magenta curve) versus all other proteins in the cell map (grey curve). *P* value between the two distributions are determined by one-sided Mann Whitney U test. Black dashed line represents the threshold number of screens used to call cancer drivers (threshold = 4,

FDR < 0.3). **c)** Number of proteins validated as cancer drivers in recurrently mutated assemblies (magenta), versus cancer drivers previously identified in pediatric (gold) or adult cancer studies (lavender). **d)** Number of transposon screening studies identifying a protein as a cancer driver (total n = 72), shown for select cancer assemblies (rows). Light grey whiskers represent mean ± SE. **e)** As for panel d, focusing on proteins in NATR assembly. **f)** Immunofluorescence images for six representative proteins in the NATR assembly found to be mutated in certain pediatric tumours. Members immunostained (green) with cytoskeleton counterstain (red). Scale bar, 2.5 μm.

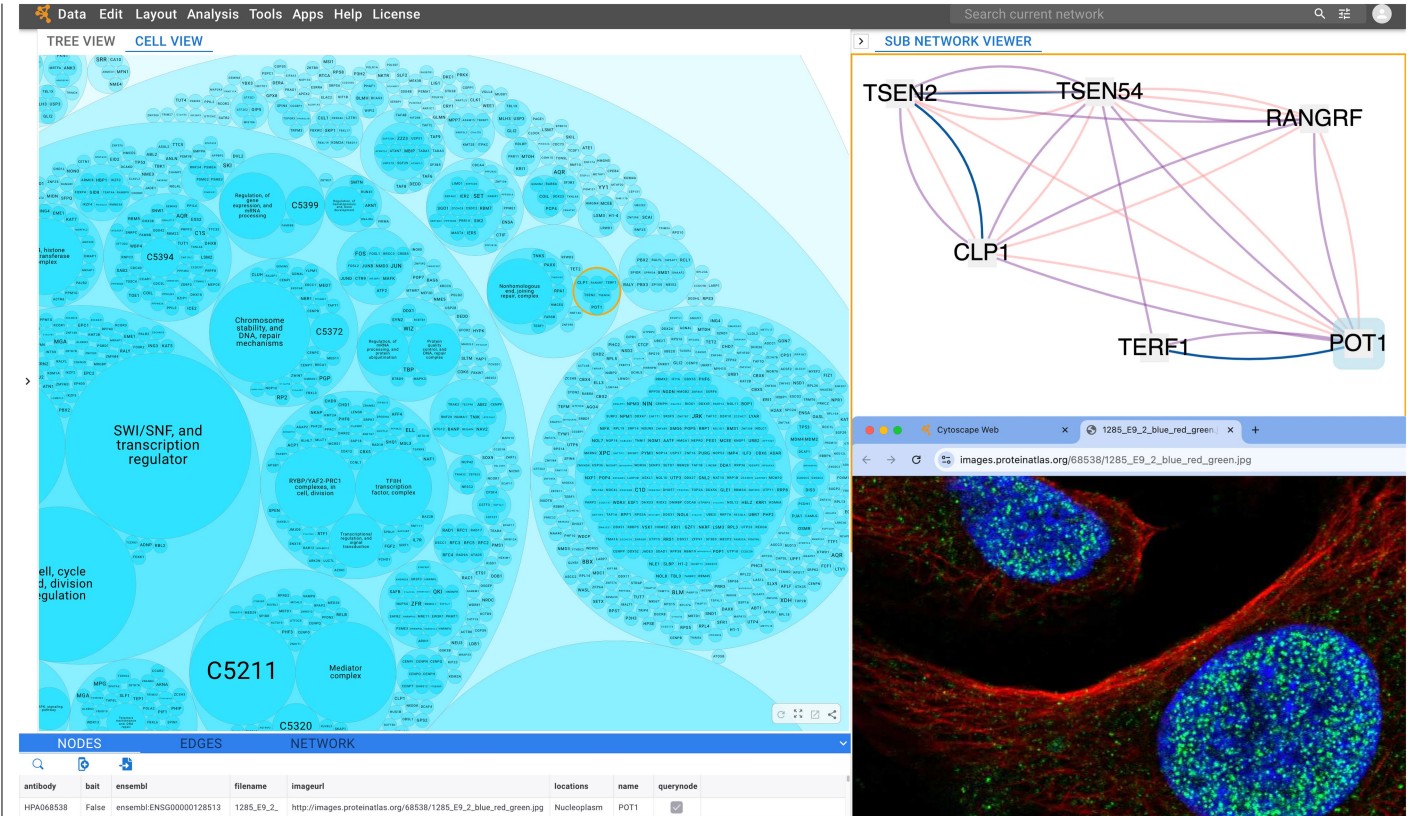

**Extended Data Fig. 10 | Navigating the multiscale cell map.** Portal for visualization of the U2OS cell map and associated data, available at http://musicmaps.ai/u2os-cellmap/. In the main display, subcellular components (protein assemblies) are displayed either as a hierarchy (as in Fig. 2b) or as a circle-packing diagram (shown here, left). In circle-packing mode, nesting of one circle inside another indicates containment of an assembly inside another; proteins are represented by the innermost circles which do not contain others; the outermost circle represents the whole cell. An endonuclease assembly has been selected in the main display (circle with yellow border), with its supporting multimodal data shown in detail in a supplemental window display (upper right). Edge colours represent evidence types: dark blue: AP-MS edges; pink: similar images; purple: similar multimodal embeddings. Further selection of nodes in the subnetwork displays protein-level information with direct links to raw data, including immunofluorescence images (lower right).

# Reporting Summary

## Statistics

For all statistical analyses, confirm that the following items are present in the figure legend, table legend, main text, or Methods section.

| n/a | Confirmed | |
|-----|-----------|---|
| ☐ | ☒ | The exact sample size (*n*) for each experimental group/condition, given as a discrete number and unit of measurement |
| ☐ | ☒ | A statement on whether measurements were taken from distinct samples or whether the same sample was measured repeatedly |
| ☐ | ☒ | The statistical test(s) used AND whether they are one- or two-sided *Only common tests should be described solely by name; describe more complex techniques in the Methods section.* |
| ☒ | ☐ | A description of all covariates tested |
| ☐ | ☒ | A description of any assumptions or corrections, such as tests of normality and adjustment for multiple comparisons |
| ☐ | ☒ | A full description of the statistical parameters including central tendency (e.g. means) or other basic estimates (e.g. regression coefficient) AND variation (e.g. standard deviation) or associated estimates of uncertainty (e.g. confidence intervals) |
| ☐ | ☒ | For null hypothesis testing, the test statistic (e.g. *F*, *t*, *r*) with confidence intervals, effect sizes, degrees of freedom and *P* value noted *Give P values as exact values whenever suitable.* |
| ☒ | ☐ | For Bayesian analysis, information on the choice of priors and Markov chain Monte Carlo settings |
| ☐ | ☒ | For hierarchical and complex designs, identification of the appropriate level for tests and full reporting of outcomes |
| ☐ | ☒ | Estimates of effect sizes (e.g. Cohen's *d*, Pearson's *r*), indicating how they were calculated |

*Our web collection on statistics for biologists contains articles on many of the points above.*

## Software and code

Policy information about availability of computer code

| | |
|---|---|
| Data collection | Immunofluorescence confocal images were collected using 63x oil immersion with Numerical Aperture 1.4. AP-MS data were acquired on first-generation Q-Exactive mass spectrometers (Thermo Fisher Scientific) equipped with Famos autosamplers (LC Packings) and Accela600 liquid chromatography (LC) pumps (Thermo Fisher Scientific). SEC-MS data was acquired on a TimsTOF Pro2 (Bruker Daltonics) with CaptiveSpray source coupled to nanoElute UHPLC (Bruker Daltonics). |
| Data analysis | All data analyses have been described in detail in the relevant Methods section with links to publicly available GitHub repositories. The cell map construction pipeline is available at https://github.com/idekerlab/cellmaps_pipeline. The required Python packages, along with specific versions, were documented in GitHub.

The following versions were used to construct the cell map:
Python== 3.8.16
cellmaps_imagedownloader==0.1.0a7
cellmaps_ppidownloader==0.1.0a6
cellmaps_image_embedding==0.1.0a7
cellmaps_ppi_embedding==0.2.0a6
cellmaps_coembedding==0.1.0a6
cellmaps_generate_hierarchy==0.1.0a13
cellmaps_hierarchyeval==0.1.0a5

The following versions were used to analyze the cell map (Use Cases):
Python==3.7.9 |

```
scipy==1.7.3

The following versions were used for the Random Forest analysis:
Python==3.9.18
scikit-learn==1.3.0

Other softwares used in this study are documented in the Methods section and listed below:
CompPASS    https://github.com/dnusinow/cRomppass
CompPASS-Plus    https://github.com/HMSBioPlex/ CompPASS-Plus-CLI
Cytoscape v3.10.1
DIA-NN  1.8.1.0 https://github.com/vdemichev/DiaNN
HiSig https://github.com/fanzheng10/HiSig
IUPred3.0  https://iupred3.elte.hu/download_new
PrInCE https://github.com/fosterlab/PrInCE
localcolabfold https://github.com/YoshitakaMo/localcolabfold
Integrative Modeling Platform (IMP) package v2.18 https://integrativemodeling.org
MutSigCV v1.4 (available in https://github.com/fanzheng10/HiSig/)
```

For manuscripts utilizing custom algorithms or software that are central to the research but not yet described in published literature, software must be made available to editors and reviewers. We strongly encourage code deposition in a community repository (e.g. GitHub). See the Nature Portfolio guidelines for submitting code & software for further information.

# Data

Policy information about availability of data

All manuscripts must include a data availability statement. This statement should provide the following information, where applicable:
- Accession codes, unique identifiers, or web links for publicly available datasets
- A description of any restrictions on data availability
- For clinical datasets or third party data, please ensure that the statement adheres to our policy

The Multiscale Integrated Cell web portal (musicmaps.ai/u2os-cellmap) provides links to all major data and derived resources associated with this study, including AP-MS protein interactions, protein immunofluorescence images, size-exclusion chromatography data, and the online interactive U2OS cell map. The U2OS cell map is available on ndexbio.org with the uuid f693137a-d2d7-11ef-8e41-005056ae3c32. Protein assemblies in the cell map are also available at the European Bioinformatics Institute (EBI) Protein Complex Portal (https://www.ebi.ac.uk/complexportal) with the query CLO:0009454. The AP-MS protein interactions are available on ndexbio.org with the uuid 95bc75d5-d1d1-11ee-8a40-005056ae23aa. In addition to its release here, the U2OS protein interaction network will be included as part of the upcoming BioPlex v4.0 database release (Huttlin et al. in preparation). AP-MS raw mass spectrometry files are available on MassIVE with the identifier MSV000097168. The entire image dataset is included in the Human Protein Atlas v23 release. SEC-MS raw mass spectrometry files and search results are available via Proteome Xchange with the identifier PXD052362. All structural models are available in the ModelArchive Database (modelarchive.org) with the identifiers ma-idk-u2osmap and ma-m5og4. Other public databases and resources used in this study include the Gene Ontology (June 2023 release, https:// geneontology.org), CORUM (version 4.1 release, https://mips.helmholtz-muenchen.de/corum/),  UniProt Homo sapiens proteome (accessed June 2 and September 11, 2023, https://uniprot.org), STRING interactome (v12, NDEx uuid 0b04e9eb-8e60-11ee-8a13-005056ae23aa), OpenCell interactions (https:// opencell.czbiohub.org/download), CD-CODE condensate database (accessed May 31, 2023, https://cd-code.org), FuzDrop (Dataset S7 in Hardenberg et al.), Protein Condensate Atlas (Supplementary Dataset 8 in Saar et al.) K562 day-8 perturb-seq dataset (gwps.wi.mit.edu), HEK-293 BioPlex v3.0 (NDEx uuid 6b995fc9-2379-11ea-bb65-0ac135e8bacf), pediatric cancer mutation data (https://www.cbioportal.org/study/summary?id=pediatric_dkfz_2017), and transposon-based mutagenesis screens from the Candidate Cancer Gene Database (http://ccgd-starrlab.oit.umn.edu/index.html, downloaded March 26, 2024).

# Research involving human participants, their data, or biological material

Policy information about studies with human participants or human data. See also policy information about sex, gender (identity/presentation), and sexual orientation and race, ethnicity and racism.

| Reporting on sex and gender | Not applicable |
|---|---|
| Reporting on race, ethnicity, or other socially relevant groupings | Not applicable |
| Population characteristics | Not applicable |
| Recruitment | Not applicable |
| Ethics oversight | Not applicable |

Note that full information on the approval of the study protocol must also be provided in the manuscript.

# Field-specific reporting

Please select the one below that is the best fit for your research. If you are not sure, read the appropriate sections before making your selection.

☒ Life sciences  ☐ Behavioural & social sciences  ☐ Ecological, evolutionary & environmental sciences

For a reference copy of the document with all sections, see nature.com/documents/nr-reporting-summary-flat.pdf

# Life sciences study design

All studies must disclose on these points even when the disclosure is negative.

| | |
|---|---|
| Sample size | The number of proteins analyzed in this study (n=5147) was determined based on amount of matched data available when overlapping immunofluorescence image and AP-MS interactions in the U-2 OS cell line. For follow-up experiments in this study, no statistical methods were used to pre-determine sample sizes, which were chosen to reliably observe experimental phenotypes. |
| Data exclusions | No data were excluded from analyses |
| Replication | All the data collected in this study consisted of technical or biological replicates. The number of replicates, as well as the type of replicates (i.e. technical or biological), are labeled in the relevant figures or method sections. |
| Randomization | AP-MS baits were arrayed on 96-well plates in random order, and plates were run in random order during LC-MS analysis. For other experiments in this study, randomization was used whenever possible to determine experimental order. |
| Blinding | All IF, AP-MS, and SEC-MS data were generated and processed with investigators blinded to the hypothesis. For DPP9 RT-qPCR measurement (Extended Data Fig. 6c), blinding was not applied during analysis, which we followed established procedure from previous studies. |

# Reporting for specific materials, systems and methods

We require information from authors about some types of materials, experimental systems and methods used in many studies. Here, indicate whether each material, system or method listed is relevant to your study. If you are not sure if a list item applies to your research, read the appropriate section before selecting a response.

## Materials & experimental systems

| n/a | Involved in the study |
|---|---|
| ☐ | ☒ Antibodies |
| ☐ | ☒ Eukaryotic cell lines |
| ☒ | ☐ Palaeontology and archaeology |
| ☒ | ☐ Animals and other organisms |
| ☒ | ☐ Clinical data |
| ☒ | ☐ Dual use research of concern |
| ☒ | ☐ Plants |

## Methods

| n/a | Involved in the study |
|---|---|
| ☒ | ☐ ChIP-seq |
| ☒ | ☐ Flow cytometry |
| ☒ | ☐ MRI-based neuroimaging |

## Antibodies

| | |
|---|---|
| Antibodies used | The antibodies used in this study are listed as below. The dilutions used for each are specified in the "Matched protein immunofluorescence (IF) imaging data" section in the methods. <br> - Anti-tubulin Abcam, ab7291, RRID:AB_2241126 <br> -Chicken anti-calreticulin, Abcam, ab14234, RRID:AB_2228460 <br> - Rabbit polyclonal HPA antibodies, generated within the Human Atlas Project. The list of HPA antibody IDs used in this study are found at  http://musicmaps.ai/u2os-cellmap. <br> -goat anti-rabbit Alexa488 A11034 , RRID:AB_2576217, RRID:AB_2535845, ThermoFisher, polyclonal <br> -goat anti-mouse Alexa555 A21424,RRID:AB_2535845, ThermoFisher, polyclonal <br> -goat anti-chicken Alexa647 A-21449, RRID:AB_2535866, ThermoFisher, polyclonal <br> -goat anti-rat Alexa647 A21247, ThermoFisher RRID:AB_1056356, polyclonal |
| Validation | All HPA antibodies were validated as described at https://www.proteinatlas.org/about/antibody+validation. The HPA antibodies are quality controlled for sensitivity and lack of cross-reactivity to other proteins using western blot and protein arrays. Antibodies that pass initial quality assessment are labeled as 'approved'. Antibodies that yield a staining pattern supported by independent data in UniProt are labeled as 'supported'.  For 'enhanced' antibody validation, we use the strategies outlined by the International Working Group for Antibody Validation (IWGAV), including genetic validation, recombinant expression validation, independent antibody validation targeting a different epitope, and capture validation by mass spectrometry. |

# Eukaryotic cell lines

Policy information about cell lines and Sex and Gender in Research

| | |
|---|---|
| Cell line source(s) | U-2 OS cells were obtained from the American Type Culture Collection (ATCC). HEK-293 data was published previously (Huttlin et al. Cell 2021) |
| Authentication | The U-2 OS cells used for IF stainings were authenticated according to the manufacturer ATCC using morphology, karyotyping and PCR based approaches to confirm the identity and to exclude intra and interspecies contaminations. These include an assay to detect species specific variants of the cytochrome C oxidase I gene (COI analysis) to rule out interspecies contamination and short tandem repeat (STR) profiling to distinguish between individual human cell lines and rule out intraspecies contamination. These cells were also used or the SEC-MS experiments and DPP9 experiments. The U-2 OS cells used for AP-MS were purchased directly from ATCC and no further authentication was performed. |
| Mycoplasma contamination | All cells used in this study were tested negative for mycoplasma contamination. |
| Commonly misidentified lines (See ICLAC register) | No commonly misidentified cell lines were used in this study. |

# Plants

| | |
|---|---|
| Seed stocks | No seed stocks were used in this study. |
| Novel plant genotypes | No novel plant genotypes were produced in this study. |
| Authentication | Not applicable |

