## [Peer Review File · Nature]

Multimodal cell maps as a foundation for structural and functional genomics

Corresponding Author: Professor Trey Ideker

Version 0:

Reviewer comments:

Referee #1

(Remarks to the Author)

In this manuscript, Schaffer and colleagues describe the generation of a cellular interaction map for U2OS cells by integrating cell imaging data with protein pull down data using a larger dataset and a modified strategy from a previous work of some of the same authors (Qin et al. Nature 2021). For this they have now used an extended set of protein pull-down experiments performed in this cell-line as part of the BioPlex project and immunofluorescence based protein localization data on the same cell line from the Human Protein Atlas project. The data integration is not sufficiently well explained (see below) but involves generating embeddings for both data modalities - very much like what was done previously (Qin et al. Nature 2021) - which are combined using the MUSE auto-encoder neural network (from Bao et al. Nature Biotech 2022). This generated functional/proximity association data for around 5000 proteins which were grouped in a hierarchical clustering scheme into 275 assemblies. These assemblies were, to some extent, validated by comparing with previously known gene-sets (e.g. known complexes and GO terms) as well as by comparison with co-fractionation MS data generated also for U2OS.

The second half of the manuscript is then devoted to a few applications of this “map”, including: 1) the prediction of structural models using AF-multimer, with one example that included fitting to previously published cryo-EM results; 2) the function of poorly characterized genes using a guilt-by-proximity principle; 3) the study of mutational patterns of pediatric cancers. Finally, the authors provide an easy to access database for others to search through the results, as well as links to the relevant code and data.

Studying cellular interaction networks and assemblies at different scales remains an important and challenging goal. Most large functional association networks that are commonly used today, such as the STRING database, represent messy aggregations of data that may not reflect any specific human cell type. It remains critical to develop scalable technologies that generate high confidence but context specific maps. Therefore, I think this current manuscript represents an important addition for research in this field. There are some aspects that the authors need to work on to improve the clarity of the computational methods and it would be potentially interesting to provide some comparisons with their own previous work done in a different cell type.

Major concerns:

1 - Understanding what was done in this manuscript for data integration and clustering required me to go through the methods and supplementary materials of 3 other papers and it was still not clear to me why/how some decisions in the data integration were done. The authors really need to improve the description and benchmarking of the data integration strategy. Specifically:

1a) The overall data integration strategy is similar to what was done in Qin et al. (Nature 2021) with a CNN embedding for images (DenseNet-121) and the node2vec embedding for protein interaction data. The next step diverges in that the previous paper (Qin et al.) used a random forest model trained to predict estimated protein-protein distances using different distant metrics formed from the embeddings. The current manuscript uses the MUSE autoencoder to combine the embeddings in a latent space representation that is used to infer the protein-protein “distances”/associations. The authors should explain in this manuscript exactly how they train the AE. For example - what are the loss functions ? are they also using the self-supervision loss function based on clustering and how is that clustering done ?

1b) Why was the MUSE AE used instead of the previous random forest based model ? More importantly, the authors should show that this produces better results. It is unclear if the latent space embeddings of the AE are actually more informative than any simple combination of the embeddings of the two data modalities.

2 - A major aspect of the paper is the underlying data that is used to derive the map - the pull-down MS and immunofluorescence data. It was unclear to me if these data are really new and published here or if they are public data that have been well described before and are re-used here. In particular, the immunofluorescence data seems similar to Thul et al. Science 2017. The pull-down data seems to be part of the ongoing work of the BioPlex project.

2a) If the data have been previously published then the authors should make this clear.

If these represent new datasets that are published here then there needs to be some minimum quality assessment described. While these data are being generated by labs with recognizable expertise and prior work, any new systematic dataset should come with expected quality controls that give a sense of accuracy and reproducibility. For example: are pull-downs of stable protein complex members generally reproducing the known complexes; what is the reproducibility of biological replicates; are known organelle specific proteins mostly co-localizing with the used markers, etc.

2b) For the co-fractionation data, while less important given that it is not used to build the map, it would also be useful to have some QC description.

Minor concerns

The validation of the assemblies was done at FDR<30% which seems relatively high. I think it would be best to be more cautious and just report fewer fraction of "validated" assemblies by SEC-MS. It was also unexpected to see a simple Pearson correlation metric for the pairwise distance of the SEC-MS profiles given that some of the authors have published approaches to make use of these profiles more efficiently. I don't suggest this should be done but I would be curious if this was tried and simply resulted in worse results as that would be interesting to know.

One very important benefit of doing this work is to be able to define the cellular map of specific cell types or cellular contexts. I realize that the manuscript has many use cases that I am not commenting on but one of the most exciting applications of this work for me would have been to be able to compare it with other cell types and say something about differences in cell biology. At the very least, I would be curious to know which assemblies match well or not their previous map in HEK293 cells. I realize that the previous map had fewer proteins but I think this comparison would have been potentially interesting.

For the AF-multimer, the model-score cut-off used was selected based on the distribution of predicted random pairs, also with a FDR of 30%. An ipTM score of 0.5-0.6 would be a typical expected cut-off but I agree that a permutation based approach also makes sense. The authors should indicate what actual cut-off was used and provide a table of all models attempted, including the random pairs, with the corresponding scores (ipTM, pTM and model score).

Referee #2

(Remarks to the Author)

The manuscript by Schaffer et al describes a strategy to create multimodal cell maps by integrating different types (modalities) of data, such as immunofluorescence and AP-MS data, into a single, coherent and hierarchical map of the cell. The basic concept for this was presented in a 2021 landmark study by the same authors (Qin et al, Nature). However, the present manuscript takes the approach to the next level: protein coverage is expanded by an order of magnitude, a novel self-supervised machine-learning workflow is used to combine the data more effectively, a ChatGPT4-based annotation strategy is deployed, the map predictions are validated experimentally by an orthogonal SEC-MS approach and the data are made available through an interactive web-portal as a resource. Perhaps most importantly, Schaffer et al outline three distinct use cases for these novel cell maps, each of which could lead to many similar applications in a wide range of fields. While the underlying technology is sophisticated, the overall approach is conceptually straightforward. I think the authors present a blueprint for solving a major, recurring challenge of modern biology: how to integrate different data modalities in a coherent way that delivers novel knowledge, which could not have been obtained with individual datasets alone.

The abstract and manuscript are written clearly, and I have no major concerns, but there are a number of minor issues that I think the authors could address:

- Use of the self-supervised, MUSE-based multimodal embedding strategy: Intuitively, it makes a lot of sense to use this sort of approach to combine the individual embeddings. However, I think the authors should explain in more detail their rationale for switching to this approach, and ideally also demonstrate its superiority to alternative options. For example, does this help to preserve the hierarchical nature of the data, in the sense that APMS data are given more weight for lower levels of the cell map, where they are more informative, as opposed to IF data which should be more informative for higher levels of the cell map?

- It was a little unclear to me which experimental data were specifically generated for this project: The U2OS IF data and all of the APMS and SEC-MS data? Are the AP-MS data identical to Bioplex 4 or a subset thereof?

- Page 6, line 134-135: How was this FDR calculated?

- Page 7, line 157, the 48 assemblies speculated to be condensates. Perhaps the authors could cross-reference that list with the condensate atlas recently published by Saar et al (Nat Commun 15, 5418, 2024)

- SEC-MS validation: The authors mention that it is particularly useful for "medium-to-large" assemblies, but the biological interpretation of these types of assemblies was not clear to me. Are we talking mostly about large protein complexes here, where SEC-MS confirmed novel / unexpected subunits predicted by the multimodal cell map? Or was SEC-MS helpful for even larger assemblies like condensates / compartments?

- About the use cases: one aspect which remains a bit unclear is the actual benefit of the multimodality. For example, the characterisation of understudied proteins is a powerful application of the cell maps, but how many of the 131 proteins of unknown function could have been annotated by using the APMS and IF input data separately / individually? I guess one thing to keep in mind here is that the annotation level also matters, e.g. showing that an unknown protein is nuclear is not quite as informative as showing it's part of a specific RNA processing complex.

- Related to this, how useful is the hierarchical nature of the cell map? For example, to predict cancer-associated proteins in case 3, as one goes up the hierarchy in the map, at which point does the information stop being useful? Individual proteins under-detect a disease, while complexes or pathways are more effective, but on the other extreme an entire organelle would presumably also be ineffective (unspecific) as biomarker. In principle, could one use the cell map to identify a "sweet spot" for the size of "biomarker modules"? I'm guessing this would be disease / module specific though.

- Looking at individual "cancer proteins" (page 12, second paragraph). If I understand this correctly, the authors identify 60-70 proteins which tend to be mutated in cancers without this being previously known. Clearly this could be clinically quite important. But since this analysis focusses on individual proteins, what is the benefit of the assemblies (modules)? Do they improve detection of cancer-relevant mutations by narrowing down the search space to proteins within specific modules that are linked to a cancer?

- The use of references is generally appropriate, as far as I can tell, except perhaps that I noticed the absence of fractionation-based spatial cell mapping (Lopit) in the introduction.

Referee #3

(Remarks to the Author)

Review of Nature ms 2024-06-11297 "Multimodal cell maps as a foundation for structural and functional genomics" by Ideker and colleagues

Summary: The authors report the creation of a comprehensive multi-scale cellular map outlining human protein organization. To accomplish this, they combined approximately 20,000 protein immunofluorescence images of more than 10,000 proteins with affinity purification mass spectrometry data obtained for more than 2000 proteins. Integrating these datasets yielded an intersection of 5,147 proteins. By employing a combination of machine learning-based data fusion and their previously described HiDeF algorithm, they predict 275 multiprotein assemblies. These macromolecules were assessed and annotated using a combination of structural modeling, automated LLM/GPT-4 inference, and subjective expertise. The general composition of a substantive fraction of the predicted communities/complexes were independently evaluated using size exclusion co-fractionation.

Perspective: There is potential utility for this cell map model in various biological areas, including structural biology, protein function, and cancer studies. Yet while certainly an interesting read, the paper does not seem as transformational as the title would suggest. I do not see the kind of conceptual, technical or biological advancement expected of a Nature paper, and the novelty overall is borderline. Most of the IF results (over 17,000 images and around 9,000 antibodies) has been previously published (about 3000 images and 1600 antibodies are new to this study). As for the methodology, the computational methods do not appear to be innovative, although the overall pipeline (way they put together different steps) is progressive. The Ideker group has for some time now reported using AI to fuse image and molecular data and this is another example of such work. Their approach takes image data that is run through an AI approach as well as the proteomics data to produce an embedding that is subsequently analyzed by HiDef. The use of ChatGPT4 to annotate gene assemblies is an interesting application of this LLM tool. However, in terms of novelty, they are still largely on conventional ground, wherein they analyze a network from their established computational approach under the assumption that the assemblies are static (don't change in different physiological contexts). In the end, they select 3 use cases, wherein they illustrate the utility of their map. The use of AlphaFold-Multimer to predict the structure of assemblies and also figuring out the likely function of unannotated proteins is expected/intuitive. The same holds for the co-analysis with the cancer genomes. What is missing is a big moment, where the authors show something counterintuitive or groundbreaking that their map allows them to define. As it currently stands, I believe this is an expensive attempt to produce something of limited general use (i.e. relevant to researchers studying U2OS cells). A generalizable workflow and associated bioinformatics scripts that capitalize on existing databases would likely have broader adoption and greater impact.

Other points:

1. The paper identifies many new protein assemblies, but the biological significance and functional implications of these findings are not fully explored. While the study lists novel groupings, it does not always provide a clear delineation of how

these discoveries advance current knowledge or potential therapeutic applications. The authors would do well to discuss examples that better showcase the utility of this resource-intensive workflow. Specifically, in all three examples of use cases that the authors have presented, it is unclear how the authors justify the claim that they leverage the combined multi-layer data (=use case). For example, while they effectively leverage AI tools like AlphaFold-Multimer to predict protein structures, the critical link to the microscopy data remains elusive. In other words, how does the addition of the microscopy data to derive PPIs and communities make this workflow superior, in the context of these specific use cases, to any other method of detecting PPIs and protein complexes? In other words, how would the outputs of AF-multimer (i.e., a structural model between protein A and protein B) change if the PPI were derived from another protein complex profiling approach, e.g., SEC-MS?

2. The authors report an overlap of only 41 assemblies out of 275 identified with a known complex using a Jaccard index of $\geq 50\%$ and an FDR of 20%. Does this level of overlap inspire confidence in the robustness of the analysis pipeline, relative to well-established knowledge? What is the overlap with established knowledge at a more stringent FDR of 1%? This is a critical question, as the model was trained, and often referenced, on this knowledge base. Addressing these concerns is key to evaluate the impact of the study's conclusions.

3. While the methodology described in the study captures the latest buzzwords (AI, GPT4, etc.), it raises fundamental questions about why existing resources such as curated databases (UniProt, HPA, etc.) on protein localizations—which contain experimentally validated knowledge—were not sufficient. Did they encounter specific limitations in these databases that necessitated generating their own dataset? These reasons should be clearly articulated in this manuscript.

4. Approximately 20-30% of proteins (if not more) are known to exhibit multifunctional and multicompartmental characteristics, operating within multiple subcellular locations concurrently. Figure 2, however, appears to oversimplify this complex reality, potentially neglecting a critical dimension of cellular organization. It is disconcerting that such a significant aspect of protein behavior and regulation might have been overlooked in this representation.

5. The use of self-supervised machine learning to fuse IF and AP-MS data relies on the assumption that the features extracted are representative of true biological phenomena. The model's performance might be compromised by systematic biases present in the training data, such as biases introduced during the selection of proteins for tagging or biases inherent in the imaging and MS techniques. Additionally, the reliance on reconstruction and contrastive loss functions assumes that all relevant biological features can be captured in the learned embedding space, which may not be true if the models lack sufficient capacity or if critical biological features are not well-represented in the input data. Without extensive ground truth datasets to validate these features, the risk of overfitting or capturing spurious correlations remains high.

6. The integration of multiple data types (immunofluorescence, AP-MS, SEC-MS) assumes that these methods are complementary and that discrepancies can be resolved through computational means. However, each method has its own limitations and potential for artifacts, which might not be fully accounted for (or not explained properly in the manuscript) in the integration process. This could lead to erroneous conclusions about the existence or nature of certain protein complexes. For example, how did the authors ensure that the C-terminal FLAG-HA tag did not impact the localization or function of the tagged proteins?

7. While the identification of novel protein assemblies is a highlight of the study, the interpretation of their functional significance is often speculative. The use of large language models like GPT-4 for functional annotation introduces a layer of abstraction that, while interesting and useful, it may not capture nuanced biochemical roles. The reliance on LLMs also raises concerns about interpretability and reproducibility of the annotations, as the models' outputs can be sensitive to phrasing of inputs and the context provided.

Version 1:

Reviewer comments:

Referee #1

(Remarks to the Author)

The authors have addressed my comments very thoroughly. In particular, they have showed that the MUSE multimodal data integration can perform as well or better than the previous random forest model and they have clarified what were new experimental datasets, providing the relevant QC metrics for those. I appreciated the addition of the comparison with the HEK293 interactions to emphasize the importance of cell-type specific interactions. Overall, I believe this represents the state-of-the-art for the multi-modal generation of cell-type a specific protein interaction map that can provide a road-map for the development of additional such maps in different cell-types. I have no further concerns.

Referee #2

(Remarks to the Author)

The authors have successfully addressed all my comments. I was surprised that the self-supervised multimodal embedding

works as well as a supervised approach, but on closer inspection of the methods section it appears that the Random Forest was not trained e.g. on a subset of the high-scoring STRING interactions, but on GO annotation similarity. Perhaps that could be mentioned in the figure legend as well. In any case, I think the comparison with the simple input concatenation is more important, and this is clearly outperformed by the self-supervised approach.

I have no more outstanding concerns.

(Remarks on code availability)

I have not reviewed the actual code in detail, because I'm not sufficiently familiar with Python for that. However, I have checked the overall documentation of the code, which is excellent. The code documentation is not limited to a GitHub Readme file, but there is a dedicated website for it. For an earlier and similarly documented version of this software (Qin et al, Nature 2021) a PhD student in my lab had no problems to set up the reported workflow and apply it to a new dataset.

Referee #3

(Remarks to the Author)

The authors have done a commendable job addressing the main concerns, the revised manuscript is improved and now suitable for publication.

Point-by-point response to reviewer comments

Referee #1 (Remarks to the Author):

In this manuscript, Schaffer and colleagues describe the generation of a cellular interaction map for U2OS cells by integrating cell imaging data with protein pull down data using a larger dataset and a modified strategy from a previous work of some of the same authors (Qin et al. Nature 2021). For this they have now used an extended set of protein pull-down experiments performed in this cell-line as part of the BioPlex project and immunofluorescence based protein localization data on the same cell line from the Human Protein Atlas project. The data integration is not sufficiently well explained (see below) but involves generating embeddings for both data modalities - very much like what was done previously (Qin et al. Nature 2021) - which are combined using the MUSE auto-encoder neural network (from Bao et al. Nature Biotech 2022). This generated functional/proximity association data for around 5000 proteins which were grouped in a hierarchical clustering scheme into 275 assemblies. These assemblies were, to some extent, validated by comparing with previously known gene-sets (e.g. known complexes and GO terms) as well as by comparison with co-fractionation MS data generated also for U2OS.

The second half of the manuscript is then devoted to a few applications of this “map”, including: 1) the prediction of structural models using AF-multimer, with one example that included fitting to previously published cryo-EM results; 2) the function of poorly characterized genes using a guilt-by-proximity principle; 3) the study of mutational patterns of pediatric cancers. Finally, the authors provide an easy to access database for others to search through the results, as well as links to the relevant code and data.

Studying cellular interaction networks and assemblies at different scales remains an important and challenging goal. Most large functional association networks that are commonly used today, such as the STRING database, represent messy aggregations of data that may not reflect any specific human cell type. It remains critical to develop scalable technologies that generate high confidence but context specific maps. Therefore, I think this current manuscript represents an important addition for research in this field. There are some aspects that the authors need to work on to improve the clarity of the computational methods and it would be potentially interesting to provide some comparisons with their own previous work done in a different cell type.

We thank the reviewer for their time in carefully evaluating our work and for appreciating the need for scalable maps of human cell types as “an important addition for research in this field.”

Major concerns:

1 - Understanding what was done in this manuscript for data integration and clustering required me to go through the methods and supplementary materials of 3 other papers and it was still not

clear to me why/how some decisions in the data integration were done. The authors really need to improve the description and benchmarking of the data integration strategy. Specifically:

1a) The overall data integration strategy is similar to what was done in Qin et al. (Nature 2021) with a CNN embedding for images (DenseNet-121) and the node2vec embedding for protein interaction data. The next step diverges in that the previous paper (Qin et al.) used a random forest model trained to predict estimated protein-protein distances using different distant metrics formed from the embeddings. The current manuscript uses the MUSE autoencoder to combine the embeddings in a latent space representation that is used to infer the protein-protein “distances”/associations. The authors should explain in this manuscript exactly how they train the AE. For example - what are the loss functions ? are they also using the self-supervision loss function based on clustering and how is that clustering done ?

As the reviewer notes, some of the methods underlying our data processing pipeline have been published previously whereas others have not. We can see how our previous Methods section, which prioritized mainly the novel aspects, may have been confusing. Accordingly, in the updated manuscript we have consolidated all data integration steps into a self-contained series of sections in the Methods, which together give details of all steps in order, whether or not the relevant methods have been previously published. As part of this refactoring, we have added Extended Data Fig. 2a and included new subsections entitled *Multimodal embedding overview*, *Encoder/decoder architecture*, *Loss functions*, and *Model training*, which together cover all of the requested details:

Multimodal embedding overview

We developed a self-supervised multimodal machine learning model to integrate (co-embed) the AP-MS and IF protein representations into a single low-dimensional (128-dimension) embedding space (Extended Data Fig. 2a). Our model is based on the autoencoder architecture known as Multimodal Structured Embedding (Bao et al. 2022) with modifications. Parameters of the autoencoder are trained using a two-component loss function which combines reconstruction loss and triplet loss; the reconstruction loss encourages the model to retain information from the original modalities in the multimodal embedding, while the triplet loss encourages the model to preserve structures from the original modalities in the embedding space. Details are provided in the following sections: “Encoder/decoder architecture”, “Loss functions”, and “Model training”.

Encoder/decoder architecture

The separate AP-MS and IF vector inputs (x_i and y_i for each protein i , see above) are compressed by modality-specific encoders (f_x and f_y), yielding 128-dimension vectors a and b :

$$a_i = f_x(x_i) = \text{Dropout}(\text{Linear}(\text{BatchNorm}(\text{ELU}(\text{Dropout}(\text{Linear}(\text{BatchNorm}(\text{Tahn}(x_i))))))))$$

$$b_i = f_y(y_i) = \text{Dropout}(\text{Linear}(\text{BatchNorm}(\text{ELU}(\text{Dropout}(\text{Linear}(\text{BatchNorm}(\text{Tahn}(y_i))))))))$$

‘Dropout’ indicates dropout layers (Srivastava et al. 2014); ‘Linear’ indicates linear transformation layers; ‘BatchNorm’ indicates batch normalization (Ioffe and Szegedy

2015); ‘tanh’ indicates a hyperbolic tangent function; and ELU indicates an Exponential Linear Unit function. The a and b vectors are then concatenated to a unified 256-dimension vector input to a joint encoder f_z that learns the L2-normalized 128-dimension latent representation z_i :

$$z_i = f_z[\text{concat}(a_i, b_i)] = \text{Dropout}(\text{Linear}(\text{BatchNorm}(\text{L2Norm}(\text{concat}(a_i, b_i))))))$$

Values of z_i constitute the “self-supervised multimodal embedding” used for subsequent cell map evaluation (see below “Evaluation of embedding approaches”) and construction (see below “Pan-resolution community detection”). For the decoder step, z is reverse-transformed to extract 128-dimension modality-specific features via weight matrices w_x and w_y :

$$c_i = w_x z_i$$

$$d_i = w_y z_i$$

Finally, these features are passed to modality-specific decoders (g_x and g_y), yielding the

1024-dimension reconstructed inputs (\hat{x}_i, \hat{y}_i):

$$\hat{x}_i = g_x(c_i) = \text{Linear}(\text{ELU}(\text{Linear}(\text{Tanh}(\text{Linear}(c_i)))))$$

$$\hat{y}_i = g_y(d_i) = \text{Linear}(\text{ELU}(\text{Linear}(\text{Tanh}(\text{Linear}(d_i)))))$$

Loss functions

To compute the reconstruction loss R , the (\hat{x}_i, \hat{y}_i) outputs of the autoencoder are compared to the original input values (x_i, y_i) for each modality:

$$R_x = \frac{1}{n} \sum_{i=1}^n (x_i - \hat{x}_i)^2$$

$$R_y = \frac{1}{n} \sum_{i=1}^n (y_i - \hat{y}_i)^2$$

where n is the total number of proteins. The overall reconstruction loss is the sum of modality-specific reconstruction losses and a regularization term, where $\lambda_{\text{regularization}}$ is the regularization weight and $\|w\|_F$ is the F-norm of the matrix:

$$R = R_x + R_y + \lambda_{\text{regularization}} (\|w_x\|_F + \|w_y\|_F)$$

To compute triplet loss T , clustering using the Louvain algorithm(Blondel et al. 2008) is performed on the (a, b) vectors of each modality (during pre-training clusters are defined using input (x, y) values instead, see “Model training” section below). This clustering defines selection functions S_x and S_y for each modality, with $S(i, j) = 1$ for proteins i, j in the same cluster, else 0. This information is used to compute T for each modality:

$$T_x = \frac{1}{m} \sum_{i \in N} \sum_{j \in N, j \neq i} \sum_{k \in N, k \neq i, j} S_x(i, j)(1 - S_x(i, k)) \times \max \left(\|z_i - z_j\|_2 - \|z_i - z_k\|_2 + \varepsilon, 0 \right)$$

$$T_y = \frac{1}{m} \sum_{i \in N} \sum_{j \in N, j \neq i} \sum_{k \in N, k \neq i, j} S_y(i, j)(1 - S_y(i, k)) \times \max \left(\|z_i - z_j\|_2 - \|z_i - z_k\|_2 + \varepsilon, 0 \right)$$

where N is the set of all proteins and m is the total number of terms inside the summation that are greater than 0. The full loss function L is a weighted sum of the reconstruction and triplet losses:

$$L = R + \lambda_{\text{triplet}} (T_x + T_y)$$

Model training

Model parameters were trained with standard neural network learning procedures provided by Pytorch(Paszke et al. 2019) v2.0.1, based on backpropagation using the Adam stochastic gradient descent method(Kingma and Ba 2014). Training occurred in three phases: 1) Over the first 200 epochs, only the reconstruction loss R was used for backpropagation. 2) Over an additional 200 epochs, the full loss function L was used for backpropagation, with S_x and S_y defined using input x, y values. 3) Over a final 500 epochs of training, the full loss function L was used for backpropagation, with S_x and S_y defined using a, b values (updated every 200 epochs). Values of hyperparameters were set based on previous work(Bao et al. 2022) without fine-tuning: batch size = 64, $\lambda_{\text{regularization}} = 5$, $\lambda_{\text{triplet}} = 5$, Adam optimization learning rate = 0.0001. Triplet loss margin and dropout percentages ($\epsilon = 0.10$, dropout = 0.25) were set based on commonly recommended values(Choubineh et al. 2023; Zhao et al. 2018).

Extended Data Fig. 2 | Self-supervised embedding of multiple data modalities. a) Architecture of self-supervised multimodal embedding model. Columns of squares represent feature vectors, with the dimensionality written just below each column. Regions enclosed by dotted lines represent neural networks with layers described. Protein coordinates in the joint multimodal embedding (z) are used for computing pairwise protein-protein similarities in subsequent panels (cosine similarity function).

In addition, we have formulated the complete integrative cell mapping workflow into a freely available Python toolkit with documentation. To emphasize this aspect of our work, we have added a new section to the paper, entitled “Construction and exploration of human cell maps.”

... To facilitate continued map improvement, incorporation of new datasets, and construction of new cell maps across subtypes and disease states, we also developed the Cell Mapping Toolkit (<http://cellmaps-pipeline.readthedocs.io/>), which implements the end-to-end pipeline described herein as a series of Python packages complete with full user documentation. This toolkit provides a flexible and generalizable framework for cell map construction, allowing researchers to integrate and construct cell maps via multiple input modalities.

1b) Why was the MUSE AE used instead of the previous random forest based model ? More importantly, the authors should show that this produces better results. It is unclear if the latent space embeddings of the AE are actually more informative than any simple combination of the embeddings of the two data modalities.

We thank the reviewer for prompting us to include additional details about the rationale and evidence in favor of introducing a new method. The MUSE multimodal data integration framework was used instead of the previous random forest classifier for two reasons – theoretical and practical. First, it is a self-supervised method based solely on the image/interaction inputs without requiring any training against prior cell structural knowledge. In contrast, our previous random forests classifier was supervised to relate patterns in data to known cellular substructures. Because unsupervised approaches make fewer assumptions, they tend to be less prone to overfitting and better for discovering novel structures that have not been previously defined. Second, we evaluated performance of the MUSE framework in comparison to a panel of alternative methods, including the previous random forests approach. We found that, in practice, MUSE is not significantly worse than the supervised Random Forests approach for recovery of STRING and CORUM gold standards, and it performs significantly better for recovery of a genome-wide perturb-seq dataset; furthermore, MUSE performs significantly better than simple concatenation of the original features for all three gold standards (new Extended Data Fig. 2, reproduced below). In development of our pipeline, we had found these results to be particularly impressive given MUSE was not supervised by any gold-standard database.

We have included new Results text and a new Extended Data Figure 2 to capture these points:

This multimodal embedding exhibited good performance in recovering known subcellular organization (Fig. 2a, Extended Data Fig. 2b-d), performing as well as, or better than, alternative supervised and unsupervised approaches (Methods, Extended Data Fig. 2e).

Extended Data Fig. 2 | Self-supervised embedding of multiple data modalities. b) Distribution of similarities shown for protein pairs with a ‘high-confidence interaction’ denoted in the *STRING* database (green) in comparison to all other protein pairs (gray). **c)** Similar to (b) but for protein pairs in the same *CORUM* complex. **d)** Similar to (b) but for protein pairs that yield highly similar transcriptional profiles (top 1% pairs) when genetically disrupted by *CRISPR*, drawn from a recent perturb-seq functional genomics study (Replogle et al. 2022). **** denotes significant difference, $p < 0.0001$ by one-sided Wilcoxon rank-sum test. **e)** Different protein embedding approaches (colored points, **Methods**) are evaluated by their degree of enrichment (x-axis) across orthogonal functional and physical interaction resources (y-axis, resources from panels b-d above). Enrichment computed using Cliff’s Delta (1,000 samplings of 1,000 protein pairs with replacement, **Methods**) yielding values in range $[-1,1]$, with positive values indicating enrichment above random expectation. Error bars denote standard deviations across 1000 jackknife resamplings. * denotes significant difference in comparison with self-supervised multimodal embedding results ($p < 0.05$ for difference in means across jackknife resamplings, **Methods**).

2 - A major aspect of the paper is the underlying data that is used to derive the map - the pull-down MS and immunofluorescence data. It was unclear to me if these data are really new and published here or if they are public data that have been well described before and are re-used here. In particular, the immunofluorescence data seems similar to Thul et al. Science 2017. The pull-down data seems to be part of the ongoing work of the BioPlex project.

The pull-down MS data are indeed new, see subsequent points below:

2a) If the data have been previously published then the authors should make this clear. If these represent new datasets that are published here then there needs to be some minimum quality assessment described. While these data are being generated by labs with recognizable expertise and prior work, any new systematic dataset should come with expected quality controls that give a sense of accuracy and reproducibility. For example: are pull-downs of stable protein complex members generally reproducing the known complexes; what is the reproducibility of biological replicates; are known organelle specific proteins mostly co-localizing with the used markers, etc.

Our study invokes three major proteome-wide cell mapping efforts for U2OS cells. The first, based on affinity purification mass spectrometry (AP-MS), is unpublished and not previously available. The second, immunofluorescence imaging (IF), was previously released on the Human Protein Atlas data portal, with most but not all of the data included in previous publications. The third, protein co-fractionation based on size-exclusion chromatography mass spectrometry (SEC-MS), is unpublished and not previously available. Thus, of the three major datasets, only the IF data have been presented previously. We have now edited the Results section to better convey this information and to provide additional quality control metrics for the newly generated AP-MS and SEC-MS datasets. For example, for the AP-MS and IF data:

We systematically tagged proteins in U2OS cells via lentiviral expression of C-terminal FLAG-HA-tagged baits available in the human ORFeome library (Huttlin et al. 2015, 2017) (Fig. 1a, Extended Data Fig. 1a,b). A total of 2,287 proteins were successfully tagged and isolated from U2OS whole-proteome extracts via affinity purification; interacting partners were identified by tandem mass spectrometry (AP-MS) to yield a total of 36,842 interactions among 7,543 proteins (Methods). Data were required to pass a panel of quality control measures implemented as previously described (Huttlin et al. 2015, 2017); these included sequence validation of lentiviral clones, detection of tagged bait proteins in each AP-MS run, and monitoring for sufficient numbers of protein and peptide identifications (Methods). Additional quality control metrics included recovery of known complexes (Extended Data Fig. 1c,d), for which the new interactions showed coverage comparable to previous AP-MS datasets (“BioPlex Interactome,” n.d.).

To match these protein interactions with parallel information on protein subcellular locations, we amassed a large collection of confocal images of U2OS cells stained with immunofluorescence antibodies against each of 10,348 proteins (20,660 images total, Fig. 1a). Each sample was simultaneously co-stained with reference markers for nucleus, endoplasmic reticulum, and microtubules, providing a reference set of subcellular landmarks common to all images. Of these data, 17,368 images had been collected in a previous publication (Thul et al. 2017), whereas the remaining 3,292 images we had more recently generated and validated according to Human Protein Atlas (HPA) standard procedures for image and antibody quality control.

We added the following text to the Methods under the section “Affinity purification mass spectrometry (AP-MS) data collection”:

Steps for quality control were as follows. Clones were sequence-validated as described previously (X. Yang et al. 2011). AP-MS analyses for each bait required the bait protein to be detected in the Sequest results; additionally, bait proteins were required to have a higher abundance (based on spectral counting) in their own pulldown than other pulldowns on the same 96-well plate. To remove under-loaded samples, we required LC-MS runs to contain a minimum of ~5000 PSM's and ~700 proteins. Enrichment of interactions within CORUM complexes (**Extended Data Fig. 1c,d**; CORUM version 4.1) was computed using a one-sided binomial test, assuming background probability of interaction equal to the network's interaction density, with Benjamini-Hochberg (BH) FDR correction. CORUM complexes were limited to those with at least 3 proteins and at least one AP-MS bait in the U2OS network. Randomized networks were constructed for each case preserving the overall number of interactions per bait (node degrees).

We have included a new figure for quality control of the AP-MS network (Extended Data Fig. 1), reproduced below:

Extended Data Fig. 1 | Data quality assessment. a) Complete network of U2OS protein-protein interactions as measured by AP-MS. **b)** Histogram of number of interactions per protein. **c)** Fraction of CORUM complexes significantly enriched (1% FDR, Methods) for AP-MS interactions measured in USOS (left) alongside interactions

ascertained in two previously published AP-MS networks for other cell lines (Huttlin et al. 2021; Cho et al. 2022) (middle and right). Error bars denote 95% confidence intervals. **d)** Interaction networks for select CORUM complexes, with FDR q-values as per panel (c). Blue nodes denote bait proteins and grey nodes denote prey proteins. **e)** PANTHER (Thomas et al. 2022) classifications of protein function (top 30 largest classes by number of proteins), shown for proteins covered by U2OS cell map in comparison to the entire human proteome (UniProt, downloaded September 2023). **f)** Protein pairs ranked by cosine similarity in AP-MS features (**Methods**) enrich for the most similar protein pairs (top 1% in the immunofluorescent protein image features and **g)** vice versa.

2b) For the co-fractionation data, while less important given that it is not used to build the map, it would also be useful to have some QC description.

Since the co-fractionation data (SEC-MS) represent a sizable whole-proteome dataset novel to our study, we very much like the suggestion to include detailed QC information. We have thus included the following text in the Results and new Extended Data Fig. 5 with a quality control analysis of the co-fractionation dataset:

*Quality assessment of the SEC-MS dataset showed that elution profiles were largely reproducible across replicate biological measurements (**Extended Data Fig. 5a,b**), with protein peaks present across the full range of fractions (**Extended Data Fig. 5c**).*

Extended Data Fig. 5 | Quality assessment of SEC-MS dataset. a) Overlap of protein identifications across three biological replicates. **b)** Violin plots showing the distribution of the Pearson correlation between replicate measurements of each protein's elution pattern (purple, $n = 5018$) vs. random pairings of different proteins across replicates (white, $n = 5018$), with thick black lines representing the median. **c)** Histogram of number of proteins with maximum intensity in each elution fraction for each replicate replicate 1 (top), replicate 2 (middle), and replicate 3 (bottom). Select CORUM complexes are highlighted at the median maximum intensity of proteins in the complex.

Minor concerns

The validation of the assemblies was done at $FDR < 30\%$ which seems relatively high. I think it would be best to be more cautious and just report fewer fraction of "validated" assemblies by SEC-MS. It was also unexpected to see a simple Pearson correlation metric for the pairwise distance of the SEC-MS profiles given that some of the authors have published approaches to

make use of these profiles more efficiently. I don't suggest this should be done but I would be curious if this was tried and simply resulted in worse results as that would be interesting to know.

We have adjusted the FDR to a more stringent 5% and also now added a complementary analysis of the SEC-MS data using the PrInCE pipeline, as previously developed by some of the authors of the present paper (Foster laboratory) to infer interacting protein pairs from SEC-MS data. We have added the following text to the Methods and included these new results in Supplementary Table 4:

*Assemblies with FDR < 5% were considered validated. A similar analysis was performed using PrinCE scores(Stacey et al. 2017) to rank protein pairs rather than Pearson correlations, with PrinCE run using default parameters. We found that 90 assemblies were validated at 5% FDR in the complementary analysis using PrInCE, including 70 assemblies validated by both Pearson correlation and PrinCE similarity measures (**Supplementary Table 4**).*

One very important benefit of doing this work is to be able to define the cellular map of specific cell types or cellular contexts. I realize that the manuscript has many uses cases that I am not commenting on but one of the most exciting applications of this work for me would have been to be able to compare it with other cell types and say something about differences in cell biology. At the very least, I would be curious to know which assemblies match well or not their previous map in HEK293 cells. I realize that the previous map had fewer proteins but I think this comparison would have been potentially interesting.

We fully agree with the reviewer that analyzing subcellular organization across cell types is a key application. Accordingly, we have been happy to comply with this request and have now included an initial proof-of-concept analysis comparing the new U2OS map to previous work in HEK-293 cells as a new "Use Case 3" with a new Extended Data Figure. The major new Results text is as follows:

Use Case 3: Studying the cell-type specificity of protein complexes

*Defining a global map of a given cell type confers the potential to distinguish subcellular components that are specific to that type from those that are more widely conserved. As an initial proof-of-concept towards this aim, we examined each protein assembly in the U2OS cell map for evidence of shared versus distinct biophysical interaction patterns in comparison to HEK293 human embryonic kidney cells (previously characterized by AP-MS in the BioPlex 3.0 resource(Huttlin et al. 2021)). Of the 258 assemblies with AP-MS data coverage in both cell types, we identified 103 that were conserved across cell types (**Methods, Extended Data Fig. 7a,b, Supplementary Table 7**). These included large assemblies (e.g. nucleus, cytosol) as well as small assemblies such as the spliceosome, the 9-1-1 RAD-RFC complex (**Extended Data Fig. 7c,d**), and components of the SNARE complex. The remaining 155 assemblies showed biophysical interaction patterns that were significantly different between HEK293 and U2OS cell types. For example, a cytosolic complex named "Energy metabolism regulation complex" was robustly identified in the U2OS AP-MS data, but none of the corresponding interactions were detected in HEK293 cells (**Extended Data Fig. 7e**). These examples*

illustrate how a data-driven cell map can elucidate protein assemblies that are specific or shared between cell types, providing a basis to explain different cell phenotypes and identify cell-type specific drug targets.

Extended Data Fig. 7 | Comparison of protein assemblies across U2OS and HEK293 cell lines. **a)** Schematic of the approach for determining conserved vs. U2OS specific protein assemblies in the cell map. For each assembly, the cosine similarities are determined between protein AP-MS features for U2OS and HEK293, separately. The U2OS similarities are then compared to the HEK293 similarities with a Mann-Whitney U-test. **b)** U2OS cell map (see Fig. 2), where assembly color indicates effect size (Cliff's Delta) from Mann-Whitney U-test comparing U2OS AP-MS feature similarities vs. HEK293 AP-MS feature similarities for assembly proteins. 18 assemblies that did not

have sufficient data in both cell lines were removed. Dashed boxes denote examples of strongly conserved and strongly U2OS-specific assemblies. **c)** Biophysical physical interaction data for 9-1-1 RAD-RFC complex; edge color signifies presence in HEK293 (green) or U2OS (blue) interaction networks. **d)** Immunofluorescence images for RFC1 (green) in HEK293 cells (top) or U2OS cells (bottom), with cytoskeleton counterstain (red). Scale bar, 5 μ m. **e)** Biophysical interaction data for the Energy metabolism regulation complex; edge color signifies presence in HEK293 (green) or U2OS (blue) interaction networks.

We have also largely rewritten the Discussion, which now highlights the study of cell types and contexts as a very important benefit of this work:

Achieving coverage across proteins and scales relied on at least two advances: interrogating the cell with matched proteome-wide datasets tuned to complementary types of information, and integrating these views systematically via a multimodal deep learning workflow. These advances provide a blueprint for mapping subcellular architecture that can be readily applied across human cell types and disease states. They also pave the way to expanded cell maps incorporating new modalities – such as proximity labeling, subcellular fractionation, or cryo-electron tomography – as well as time-dependent measurements – such as monitoring of subcellular dynamics over a progression of cell-cycle phases.

For the AF-multimer, the model-score cut-off used was selected based on the distribution of predicted random pairs, also with a FDR of 30%. An ipTM score of 0.5-0.6 would be a typical expected cut-off but I agree that a permutation based approach also makes sense. The authors should indicate what actual cut-off was used and provide a table of all models attempted, including the random pairs, with the corresponding scores (ipTM, pTM and model score).

We have added to the following text in the Methods to indicate the score cutoff at 30% FDR:

This null distribution was used to calculate an FDR for actual protein pair scores, selecting a cutoff of 30% corresponding to weighted PTM score 0.39.

We also included a new Supplementary Table 5 which indicates all scores and FDR for all models attempted, including both target and random pairs.

Referee #2 (Remarks to the Author):

The manuscript by Schaffer et al describes a strategy to create multimodal cell maps by integrating different types (modalities) of data, such as immunofluorescence and AP-MS data, into a single, coherent and hierarchical map of the cell. The basic concept for this was presented in a 2021 landmark study by the same authors (Qin et al, Nature). However, the present manuscript takes the approach to the next level: protein coverage is expanded by an order of magnitude, a novel self-supervised machine-learning workflow is used to combine the data more effectively, a ChatGPT4-based annotation strategy is deployed, the map predictions are validated experimentally by an orthogonal SEC-MS approach and the data are made available through an interactive web-portal as a resource.

We are happy that the intended impact of our work was read loud and clear by this reviewer.

Perhaps most importantly, Schaffer et al outline three distinct use cases for these novel cell maps, each of which could lead to many similar applications in a wide range of fields. While the underlying technology is sophisticated, the overall approach is conceptually straightforward. I think the authors present a blueprint for solving a major, recurring challenge of modern biology: how to integrate different data modalities in a coherent way that delivers novel knowledge, which could not have been obtained with individual datasets alone.

We sincerely thank the reviewer for their time and appreciation of this work. It is clear from this comment that they grasp a key point of this study: *“a blueprint for solving a major, recurring challenge of modern biology: how to integrate different data modalities in a coherent way that delivers novel knowledge, which could not have been obtained with individual datasets alone.”* We thank the reviewer for this point, which we have incorporated into the new Discussion section.

The abstract and manuscript are written clearly, and I have no major concerns, but there are a number of minor issues that I think the authors could address:

We are glad the reviewer has no major concerns with our study. Our responses to the minor issues are as follows.

- Use of the self-supervised, MUSE-based multimodal embedding strategy: Intuitively, it makes a lot of sense to use this sort of approach to combine the individual embeddings. However, I think the authors should explain in more detail their rationale for switching to this approach, and ideally also demonstrate its superiority to alternative options. For example, does this help to preserve the hierarchical nature of the data, in the sense that APMS data are given more weight for lower levels of the cell map, where they are more informative, as opposed to IF data which should be more informative for higher levels of the cell map?

We thank the reviewer for this important suggestion. We have included additional text to the Results and a new Extended Data Figure to demonstrate the rationale and utility of switching to the self-supervised embedding strategy:

This multimodal embedding exhibited good performance in recovering known subcellular organization (Fig. 2a, Extended Data Fig. 2b-d), performing as well as, or better than, alternative supervised and unsupervised approaches (Methods, Extended Data Fig. 2e).

Extended Data Fig. 2 | Self-supervised embedding of multiple data modalities. b) Distribution of similarities shown for protein pairs with a ‘high-confidence interaction’ denoted in the STRING database (green) in comparison to all other protein pairs (gray). **c)** Similar to (b) but for protein pairs in the same CORUM complex. **d)** Similar to (b) but for protein pairs that yield highly similar transcriptional profiles (top 1% pairs) when genetically disrupted by CRISPR, drawn from a recent perturb-seq functional genomics study (Replogle et al. 2022). **** denotes significant difference, $p < 0.0001$ by one-sided Wilcoxon rank-sum test. **e)** Different protein embedding approaches (colored points, Methods) are evaluated by their degree of enrichment (x-axis) across orthogonal functional and physical interaction resources (y-axis, resources from panels b-d above). Enrichment computed using Cliff’s Delta (1,000 samplings of 1,000 protein pairs with replacement, Methods) yielding values in range $[-1, 1]$, with positive values indicating enrichment above random expectation. Error bars denote standard deviations across 1000 jackknife resamplings. * denotes significant difference in comparison with

*self-supervised multimodal embedding results ($p < 0.05$ for difference in means across jackknife resamplings, **Methods**).*

- It was a little unclear to me which experimental data were specifically generated for this project: The U2OS IF data and all of the APMS and SEC-MS data? Are the AP-MS data identical to Bioplex 4 or a subset thereof?

The U2OS affinity purification mass spectrometry (AP-MS) and size exclusion chromatography mass spectrometry (SEC-MS) data are newly generated as of the present study. The U2OS immunofluorescence imaging (IF) data were previously released on the Human Protein Atlas data portal, with most but not all of the data included in previous publications. We edited the Results section to better convey which specific datasets have been newly generated versus published.

We systematically tagged proteins in U2OS cells via lentiviral expression of C-terminus FLAG-HA-tagged baits available in the human ORFeome library (Huttlin et al. 2015, 2017) (Fig. 1a, Extended Data Fig. 1a,b). A total of 2,287 proteins were successfully tagged and isolated from U2OS whole-proteome extracts via affinity purification; interacting partners were identified by tandem mass spectrometry (AP-MS) to yield a total of 36,842 interactions among 7,543 proteins (Methods). Data were required to pass a panel of quality control measures implemented as previously described (Huttlin et al. 2015, 2017); these included sequence validation of lentiviral clones, detection of tagged bait proteins in each AP-MS run, and monitoring for sufficient numbers of protein and peptide identifications (Methods). Additional quality control metrics included recovery of known complexes (Extended Data Fig. 1c,d), for which the new interactions showed coverage comparable to previous AP-MS datasets (“BioPlex Interactome,” n.d.).

To match these protein interactions with parallel information on protein subcellular locations, we amassed a large collection of confocal images of U2OS cells stained with immunofluorescence antibodies against each of 10,348 proteins (20,660 images total, Fig. 1a). Each sample was simultaneously co-stained with reference markers for nucleus, endoplasmic reticulum, and microtubules, providing a reference set of subcellular landmarks common to all images. Of these data, 17,368 images had been collected in a previous publication (Thul et al. 2017), whereas the remaining 3,292 images we had more recently generated and validated according to Human Protein Atlas (HPA) standard procedures for image and antibody quality control.

- Page 6, line 134-135: How was this FDR calculated?

We have added a reference to the Methods section to the text of this section (“Annotation of the U2OS multiscale integrated cell map”), where the analysis is described in detail.

- Page 7, line 157, the 48 assemblies speculated to be condensates. Perhaps the authors could cross-reference that list with the condensate atlas recently published by Saar et al (Nat Commun 15, 5418, 2024)

We thank the reviewer for this interesting suggestion, as this atlas was also performed in U2OS and validated with HPA images. We have now added the suggested analysis along with the following Results text and updated Supplementary Table 3:

We found 48 assemblies that are potential biomolecular condensates (Pappu et al. 2023) based on their enrichment for proteins with intrinsically disordered regions (Erdős, Pajkos, and Dosztányi 2021), proteins predicted to phase separate (Hardenberg et al. 2020), or proteins recorded in the CD-Code condensate database (Rostam et al. 2023) (Supplementary Table 3). Of these, 38 had significant overlap with a recent complementary effort to predict protein condensates through integration of diverse biochemical protein features (Saar et al. 2024) (hypergeometric FDR < 0.05), while the remaining 10 putative condensates had not been previously identified (Supplementary Table 3).

- SEC-MS validation: The authors mention that it is particularly useful for "medium-to-large" assemblies, but the biological interpretation of these types of assemblies was not clear to me. Are we talking mostly about large protein complexes here, where SEC-MS confirmed novel / unexpected subunits predicted by the multimodal cell map? Or was SEC-MS helpful for even larger assemblies like condensates / compartments?

We think the confusion was with the unclear label of "medium-to-large" which we have now removed. In particular, we have edited this text to clarify that the SEC-MS was more successful for the assemblies with >5 proteins:

Overall, SEC data validated 89 assemblies (5% FDR), corresponding to 43% of assemblies (76 / 175) with more than 5 proteins and 61% of assemblies (59 / 96) with more than 15 proteins (Fig. 3d, Supplementary Table 4, Methods).

- About the use cases: one aspect which remains a bit unclear is the actual benefit of the multimodality. For example, the characterisation of understudied proteins is a powerful application of the cell maps, but how many of the 131 proteins of unknown function could have been annotated by using the APMS and IF input data separately / individually? I guess one thing to keep in mind here is that the annotation level also matters, e.g. showing that an unknown protein is nuclear is not quite as informative as showing it's part of a specific RNA processing complex.

We thank the reviewer for this important suggestion, which we now address with additional analyses and text.

First, we note the improvement of the integrated co-embedding features over single modality features in the new **Extended Data Fig. 2**:

Extended Data Fig. 2 | Self-supervised embedding of multiple data modalities. e) Different protein embedding approaches (colored points, **Methods**) are evaluated by their degree of enrichment (x-axis) across orthogonal functional and physical interaction resources (y-axis, resources from panels b-d above). Enrichment computed using Cliff's Delta (1,000 samplings of 1,000 protein pairs with replacement, **Methods**) yielding values in range $[-1,1]$, with positive values indicating enrichment above random expectation. Error bars denote standard deviations across 1000 jackknife resamplings. * denotes significant difference in comparison with self-supervised multimodal embedding results ($p < 0.05$ for difference in means across jackknife resamplings, **Methods**).

Second, we have added text to the Results where we expand on which assemblies in the cell maps were robustly recovered in single-modality cell maps vs. the integrated map:

Estimated assembly diameters spanned the relevant scales of cell biology from 10^1 to 10^4 nanometers (Fig. 2c), with assemblies robustly identified at each of these scales (Extended Data Fig. 3a, Methods). In contrast, we found that maps constructed from only the imaging data tended to recover large assemblies but miss small ones, while maps constructed from only the AP-MS data recovered small assemblies but tended to miss large ones (Extended Data Fig. 3b,c). Overall, the integrated map identified the largest number of assemblies, including 104 that were not resolved by either individual modality (Extended Data Fig. 3d, Supplementary Table 1).

Extended Data Fig. 3 | Robustness of cell map assemblies. (a-c) Cell maps are colored by assembly robustness, measured as the percentages over 300 jackknife resamplings where an assembly was recovered. Three panels show maps built using a) both imaging and AP-MS data, b) AP-MS data only, or c) imaging data only. d) Number of robust assemblies (recovered in >50% jackknife resamplings) versus size of assembly in number of proteins. Gray bars denote total number of assemblies in each size category.

Finally, we added the following text to the Discussion, where we highlight the benefits of multimodal analysis:

Through multimodal analysis, the human cell map presented here unifies and extends multiple ongoing efforts which have thus far progressed independently. These include the separate unimodal studies of protein interaction networks(Huttlin et al. 2017; Go et al. 2021; Luck et al. 2020), protein confocal images(Thul et al. 2017), human genomes(Gröbner et al. 2018), or genome-wide genetic screens(Replogle et al. 2022; Abbott et al. 2015). In this respect, we found that the integration of multiple modes of data substantially broadens the sensitivity and robustness with which subcellular components can be resolved across scales (**Extended Data Fig. 3**). These benefits translate to real impacts in biological discovery as exhibited in the Use Cases. Approximately half of AlphaFold structures (47/111, **Supplementary Table 5**) and 44% of new protein functional annotations (**Supplementary Table 6**) were driven by assemblies that were robustly identified only by integrating both AP-MS and imaging datasets.

A separate distinct benefit of a multimodal analysis is that, by design, it provides multiple lines of evidence for new biological findings. In a typical omics study, a single modality of data is presented and analyzed with many putative findings, only a few of which can be validated or pursued at any depth. In contrast, each new finding of the U2OS cell map is derived from two complementary experimental platforms by default (AP-MS biochemical pulldowns and spatial proteomics imaging), and the systematic lines of evidence deepen further in the Use Cases via support from SEC-MS, AlphaFold predictions, perturb-seq, and/or transposon mutagenesis. For example, the assembly of multifunctional protein ERH with RNA-binding protein CCDC9B was supported by an AP-MS interaction, image subcellular annotations, SEC-MS elution profiles (**Fig. 4f**), and a high-confidence AlphaFold 3D model (**Fig. 4g**). Such confluence of data, also seen in other recent multi-omic studies(Cho et al. 2022; Cao and Gao 2022; K. D. Yang et al. 2021; Vickovic et al. 2022), not only increases confidence in each result but also provides substantial additional structural, functional, and/or spatial information. This aspect pushes towards a new mode of end-to-end cell biology whereby multiple datasets are generated, integrated, and simultaneously corroborated, informing a unified and foundational representation of the cell(Johnson et al. 2023; Schaff et al. 1997; Karr et al. 2012; Bunne et al. 2024).

- Related to this, how useful is the hierarchical nature of the cell map? For example, to predict cancer-associated proteins in case 3, as one goes up the hierarchy in the map, at which point does the information stop being useful? Individual proteins under-detect a disease, while complexes or pathways are more effective, but on the other extreme an entire organelle would presumably also be ineffective (unspecific) as biomarker. In principle, could one use the cell map to identify a "sweet spot" for the size of "biomarker modules"? I'm guessing this would be disease / module specific though.

We thank the reviewer for this important question; we agree that a compelling application of cell maps is to determine the scales at which human genetic variants and mutations converge to cause a disease phenotype. As an initial investigation towards this point, we measured the distribution of sizes for significantly mutated assemblies, which observed that mutations mainly converge in assemblies of size smaller than 50 proteins. This analysis is presented in the new Fig. 5e and accompanying text in the Results:

Mutated assemblies were identified at all size scales but had a clear preference for small complexes of <50 proteins (Fig. 5e).

Fig. 5 | e) Distribution of sizes for the recurrently mutated assemblies.

- Looking at individual "cancer proteins" (page 12, second paragraph). If I understand this correctly, the authors identify 60-70 proteins which tend to be mutated in cancers without this being previously known. Clearly this could be clinically quite important. But since this analysis focusses on individual proteins, what is the benefit of the assemblies (modules)? Do they improve detection of cancer-relevant mutations by narrowing down the search space to proteins within specific modules that are linked to a cancer?

Yes, the reviewer's intuition is correct. The significantly mutated assemblies help to narrow down (and organize) the search space for mutated proteins. We added the following text to clarify this important point:

Within these assemblies we focused on 250 "putative cancer proteins", defined as proteins that are not only present in recurrently mutated assemblies but are themselves mutated in multiple tumor samples (Methods).

We also added the following text to clarify the benefit of the assembly-level analysis in the Discussion:

A final, critical proof-of-concept was in decoding human genetics. By identifying patterns of genetic mutations that converge on protein assemblies, numerous proteins were implicated that had not been previously reported as pediatric cancer drivers (Supplementary Table 10, Fig. 6d).

- The use of references is generally appropriate, as far as I can tell, except perhaps that I noticed the absence of fractionation-based spatial cell mapping (Lopit) in the introduction.

We have now included the Lopit reference in the Introduction, as below:

Biochemical proteomics approaches, such as affinity purification mass spectrometry (AP-MS)(Huttlin et al. 2017; Gordon et al. 2020; Huttlin et al. 2015), cross-linking mass spectrometry(Jiao et al. 2022; Bogdanow et al. 2023; Wheat et al. 2021), size exclusion chromatography mass spectrometry (SEC-MS)(Skinnider et al. 2021; Havugimana et al.

2012; Bludau et al. 2020), proximity labeling(Qin et al. 2021; Kim et al. 2014; Go et al. 2021) and isotope tagging (LOPIT)(Dunkley et al. 2004; Mulvey et al. 2017) have revealed patterns of protein-protein interaction and subcellular localization which inform the makeup of protein complexes and organelles.

Referee #3 (Remarks to the Author):

Review of Nature ms 2024-06-11297 “Multimodal cell maps as a foundation for structural and functional genomics” by Ideker and colleagues

Summary: The authors report the creation of a comprehensive multi-scale cellular map outlining human protein organization. To accomplish this, they combined approximately 20,000 protein immunofluorescence images of more than 10,000 proteins with affinity purification mass spectrometry data obtained for more than 2000 proteins. Integrating these datasets yielded an intersection of 5,147 proteins. By employing a combination of machine learning-based data fusion and their previously described HiDeF algorithm, they predict 275 multiprotein assemblies. These macromolecules were assessed and annotated using a combination of structural modeling, automated LLM/GPT-4 inference, and subjective expertise. The general composition of a substantive fraction of the predicted communities/complexes were independently evaluated using size exclusion co-fractionation.

Perspective: There is potential utility for this cell map model in various biological areas, including structural biology, protein function, and cancer studies. Yet while certainly an interesting read, the paper does not seem as transformational as the title would suggest. I do not see the kind of conceptual, technical or biological advancement expected of a Nature paper, and the novelty overall is borderline. Most of the IF results (over 17,000 images and around 9,000 antibodies) has been previously published (about 3000 images and 1600 antibodies are new to this study).

We are happy the reviewer calls out the potential utility of the U2OS cell map in diverse biological areas, and also that they found the title compelling and the paper to be an interesting read. We would like to clarify that, of the three large-scale proteomics datasets included in our study, two of these datasets (proteome-wide AP-MS and SEC-MS) are indeed novel. The third dataset (IF images of protein spatial localizations) has been largely previously published as pointed out by the reviewer. But please note that immunostaining with 1,600 new validated antibodies is not a small data addition in itself. We have rewritten the opening paragraph of the Results to make these facts more apparent.

As for the methodology, the computational methods do not appear to be innovative, although the overall pipeline (way they put together different steps) is progressive. The Ideker group has for some time now reported using AI to fuse image and molecular data and this is another example of such work. Their approach takes image data that is run through an AI approach as well as the proteomics data to produce an embedding that is subsequently analyzed by HiDef. The use of ChatGPT4 to annotate gene assemblies is an interesting application of this LLM tool. However, in terms of novelty, they are still largely on conventional ground, wherein they analyze a network from their established computational approach under the assumption that the assemblies are static (don't change in different physiological contexts).

We are glad the reviewer found our end-to-end cell map pipeline to be scientifically progressive. To clarify its relation to previous work, our groups have published a single prior study establishing proof-of-principle for integrating proteomics imaging with mass spectrometry (Qin et al., *Nature* 600.7889: 536-542, 2021). This previous paper covered 661 proteins in HEK293 cells using datasets that were already published at that time. **In contrast, the present paper combines data collected for 5,147 proteins in U2OS cells based on mapping experiments that are largely new, and with the addition of a third data stream (SEC-MS).** Several steps of the data integration pipeline are substantial additions or improvements over previous formulations, e.g. the self-supervised multimodal protein embedding and the use of LLMs in map annotation as the reviewer kindly calls out.

We feel the strongest argument for our present advance is the release of the U2OS cell map itself, which achieves the first global catalog of human subcellular components in a model cell type. A substantial number of protein assemblies in this map are previously unreported, an aspect this reviewer acknowledges in their later comments and offers constructive advice as to how to highlight and expand (e.g. see comment #7, which calls out novel protein assemblies as a “highlight of the study”). We achieved coverage across proteins and scales in the data-driven map via 1) generation of proteome-wide datasets and 2) the development of an end-to-end multimodal deep learning workflow. We have worked closely with the European Bioinformatics Institute (EBI) to release the complete U2OS cell map on their Protein Complex Portal as the first example of a user-contributed map.

We fully agree that these cell maps should be dynamic and capture subcellular changes in physiological contexts, and we now mention this point as a key future direction in the new Discussion. However, to achieve this we need new dynamic proteome-wide datasets, and we would thus argue it is beyond the scope of this paper. As a first step towards this goal, we now provide an analysis of the cell-type specificity of protein assemblies in the map across two different cell types (new section “Use Case 3: Studying the cell-type specificity of protein complexes”). This analysis illustrates how a data-driven cell map can elucidate protein assemblies that are specific or shared between cell types, providing a basis to explain different cell phenotypes and identify cell-type specific drug targets.

In the end, they select 3 use cases, wherein they illustrate the utility of their map. The use of AlphaFold-Multimer to predict the structure of assemblies and also figuring out the likely function of unannotated proteins is expected/intuitive. The same holds for the co-analysis with the cancer genomes. What is missing is a big moment, where the authors show something counterintuitive or groundbreaking that their map allows them to define.

We are happy that the use of human cell maps as a resource for 3D structure and protein functional prediction was intuitive for this reviewer. The key point is that each of these applications has led to a substantial number of previously unreported structures and functions emerging from our study. We appreciate that individual biological results can get lost when presenting a genome-wide or proteome-wide analysis. We have largely rewritten the Discussion based on the reviewer’s input to better highlight the advances of this study:

Although the basic sequence of the human genome has been known for over two decades(Lander et al. 2001), knowledge of how its proteins are organized within cells and their associated functions is still very much evolving(Kustatscher et al. 2022). To advance this cause, we have developed a reference human cell map with extensive coverage of subcellular assemblies spanning four orders of magnitude ($\sim 10^{-8}$ to 10^{-5} m). Achieving coverage across proteins and scales relied on at least two advances: interrogating the cell with matched proteome-wide datasets tuned to complementary types of information, and integrating these views systematically via a multimodal deep learning workflow. These advances provide a blueprint for mapping subcellular architecture that can be readily applied across human cell types and disease states. They also pave the way to expanded cell maps incorporating new modalities – such as proximity labeling, subcellular fractionation, or cryo-electron tomography – as well as time-dependent measurements – such as monitoring of subcellular dynamics over a progression of cell-cycle phases.

*With such generality in mind, we surveyed a series of Use Cases representing common areas of investigation in which a global data-driven cell map can powerfully drive biological discovery. First we explored how protein assemblies provide the starting material for 3D structural modeling, leading to 111 high-confidence heterodimeric structures via AlphaFold (**Fig. 4b-g, Supplementary Table 5**) and a large integrative model(Sali 2021; Rout and Sali 2019) of the Rag-Ragulator complex combining computational predictions with experimental 3D coordinates (**Fig. 4h-j**). A second key impact was in the study of individual proteins, in which the cell map suggests unexpected roles for 974 proteins (**Supplementary Table 6**). As proof-of-concept, we further investigated a role for C18orf21 in the RNAse MRP complex (**Fig. 4k-m**) and for DPP9 in the ISGF3 complex (**Extended Data Fig. 6**). Other key applications were in the study of cell-type specificity (**Supplementary Table 7, Extended Data Fig. 7**), molecular condensates (**Supplementary Table 3**), and multi-localizing proteins and protein assemblies (**Supplementary Table 8, Extended Data Fig. 8**). A final, critical proof-of-concept was in decoding human genetics. By identifying patterns of genetic mutations that converge on protein assemblies, numerous proteins were implicated that had not been previously reported as pediatric cancer drivers (**Supplementary Table 10, Fig. 6d**).*

As it currently stands, I believe this is an expensive attempt to produce something of limited general use (.i.e. relevant to researchers studying U2OS cells). A generalizable workflow and associated bioinformatics scripts that capitalize on existing databases would likely have broader adoption and greater impact.

We agree with the reviewer on the high potential impact of releasing a generalizable workflow and associated bioinformatics scripts for constructing cell maps. To address this comment, we have formulated the complete integrative cell mapping workflow used in our paper into a freely available Python toolkit. To emphasize this aspect of our work, we have extended the section “Construction and exploration of human cell maps.” As the reviewer mentions, we anticipate this workflow will lay the foundation for constructing multiscale maps of subcellular components for diverse human cell types and physiological contexts.

... To facilitate continued map improvement, incorporation of new datasets, and construction of new cell maps across subtypes and disease states, we also developed the Cell Mapping Toolkit (<http://cellmaps-pipeline.readthedocs.io/>), which implements the end-to-end pipeline described herein as a series of Python packages complete with full user documentation. This toolkit provides a flexible and generalizable framework for cell map construction, allowing researchers to integrate and construct cell maps via multiple input modalities.

We would like to again emphasize that this is a proteome-wide map of subcellular components produced in a single human cell type. By analogy, the Human Genome Project focused an enormous effort on elucidating the genome of a single human individual before scaling and economizing the pipeline to capture large numbers of individuals and disease contexts – work which occurred over many ensuing years and studies. In this vein, we see the present map of a model human cell type as a critical step towards achieving cell biology maps across diverse contexts and disease states.

In summary, we thank the reviewer for their careful consideration of our manuscript and for taking the time to point out many positive aspects alongside their several concerns.

Other points:

1. The paper identifies many new protein assemblies, but the biological significance and functional implications of these findings are not fully explored. While the study lists novel groupings, it does not always provide a clear delineation of how these discoveries advance current knowledge or potential therapeutic applications. The authors would do well to discuss examples that better showcase the utility of this resource-intensive workflow. Specifically, in all three examples of use cases that the authors have presented, it is unclear how the authors justify the claim that they leverage the combined multi-layer data (=use case). For example, while they effectively leverage AI tools like AlphaFold-Multimer to predict protein structures, the critical link to the microscopy data remains elusive. In other words, how does the addition of the microscopy data to derive PPIs and communities make this workflow superior, in the context of these specific use cases, to any other method of detecting PPIs and protein complexes? In other words, how would the outputs of AF-multimer (i.e., a structural model between protein A and protein B) change if the PPI were derived from another protein complex profiling approach, e.g., SEC-MS?

We agree it is important to show that multimodal analysis leverages information that cannot be obtained with a single dataset only. While we did discuss this aspect in the original manuscript, it was perhaps too cursory. Therefore, for the revision we have expanded on the Results to emphasize the importance of multimodal fusion:

Estimated assembly diameters spanned the relevant scales of cell biology from 10^1 to 10^4 nanometers (Fig. 2c), with assemblies robustly identified at each of these scales (Extended Data Fig. 3a, Methods). In contrast, we found that maps constructed from only the imaging data tended to recover large assemblies but miss small ones, while maps constructed from only the AP-MS data recovered small assemblies but tended to miss large ones (Extended Data Fig. 3b,c). Overall, the integrated map identified the

largest number of assemblies, including 104 that were not resolved by either individual modality (**Extended Data Fig. 3d, Supplementary Table 1**).

We also note the improvement of the integrated co-embedding features over single modality features:

Extended Data Fig. 2 | Self-supervised embedding of multiple data modalities. e) Different protein embedding approaches (colored points, **Methods**) are evaluated by their degree of enrichment (x-axis) across orthogonal functional and physical interaction resources (y-axis, resources from panels b-d above). Enrichment computed using Cliff's Delta (1,000 samplings of 1,000 protein pairs with replacement, **Methods**) yielding values in range $[-1,1]$, with positive values indicating enrichment above random expectation. Error bars denote standard deviations across 1000 jackknife resamplings. * denotes significant difference in comparison with self-supervised multimodal embedding results ($p < 0.05$ for difference in means across jackknife resamplings, **Methods**).

We added the following new paragraphs to the Discussion to further emphasize this point:

Through multimodal analysis, the human cell map presented here unifies and extends multiple ongoing efforts which have thus far progressed independently. These include the separate unimodal studies of protein interaction networks(Huttlin et al. 2017; Go et al. 2021; Luck et al. 2020), protein confocal images(Thul et al. 2017), human genomes(Gröbner et al. 2018), or genome-wide genetic screens(Replogle et al. 2022; Abbott et al. 2015). In this respect, we found that the integration of multiple modes of data substantially broadens the sensitivity and robustness with which subcellular components can be resolved across scales (**Extended Data Fig. 3**). These benefits translate to real impacts in biological discovery as exhibited in the Use Cases.

Approximately half of AlphaFold structures (47/111, **Supplementary Table 5**) and 44% of new protein functional annotations (**Supplementary Table 6**) were driven by assemblies that were robustly identified only by integrating both AP-MS and imaging datasets.

A separate distinct benefit of a multimodal analysis is that, by design, it provides multiple lines of evidence for new biological findings. In a typical omics study, a single modality of data is presented and analyzed with many putative findings, only a few of which can be validated or pursued at any depth. In contrast, each new finding of the U2OS cell map is derived from two complementary experimental platforms by default (AP-MS biochemical pulldowns and spatial proteomics imaging), and the systematic lines of evidence deepen further in the Use Cases via support from SEC-MS, AlphaFold predictions, perturb-seq, and/or transposon mutagenesis. For example, the assembly of multifunctional protein ERH with RNA-binding protein CCDC9B was supported by an AP-MS interaction, image subcellular annotations, SEC-MS elution profiles (**Fig. 4f**), and a high-confidence AlphaFold 3D model (**Fig. 4g**). Such confluence of data, also seen in other recent multi-omic studies (Cho et al. 2022; Cao and Gao 2022; K. D. Yang et al. 2021; Vickovic et al. 2022), not only increases confidence in each result but also provides substantial additional structural, functional, and/or spatial information. This aspect pushes towards a new mode of end-to-end cell biology whereby multiple datasets are generated, integrated, and simultaneously corroborated, informing a unified and foundational representation of the cell (Johnson et al. 2023; Schaff et al. 1997; Karr et al. 2012; Bunne et al. 2024).

2. The authors report an overlap of only 41 assemblies out of 275 identified with a known complex using a Jaccard index of $\geq 50\%$ and an FDR of 20%. Does this level of overlap inspire confidence in the robustness of the analysis pipeline, relative to well-established knowledge? What is the overlap with established knowledge at a more stringent FDR of 1%? This is a critical question, as the model was trained, and often referenced, on this knowledge base. Addressing these concerns is key to evaluate the impact of the study's conclusions.

We reran the analysis with FDR of 1%, and the results are almost identical to those reported in the original manuscript (with the exception of one assembly which is filtered out). We have thus updated the results in the manuscript and Supplementary Tables to reflect an FDR at 1%.

3. While the methodology described in the study captures the latest buzzwords (AI, GPT4, etc.), it raises fundamental questions about why existing resources such as curated databases (UniProt, HPA, etc.) on protein localizations—which contain experimentally validated knowledge—were not sufficient. Did they encounter specific limitations in these databases that necessitated generating their own dataset? These reasons should be clearly articulated in this manuscript.

We are in fact using U2OS images from the Human Protein Atlas resource in building this data-derived cell map. As noted above, the other two major data streams (AP-MS, SEC-MS) are new to this study. Also as mentioned above, integrating these complementary datasets results in a map with coverage of subcellular assemblies that is substantially higher and more robust than

achieved with any one data modality. This is shown in Extended Data Fig. 2d, reproduced below:

Extended Data Fig. 3 | Robustness of cell map assemblies. d) Number of robust assemblies (recovered in >50% jackknife resamplings) versus size of assembly in number of proteins. Gray bars denote total number of assemblies in each size category.

The reviewer comment also asks for a comparison to curated databases. We did indeed annotate the cell map using curated databases (components in the Gene Ontology, CORUM complexes, and Human Protein Atlas localizations). As described in the “Annotation of the U2OS multiscale integrated cell map” section, we found that a substantial portion of the cell map assemblies either contain substantial variation on a known assembly or were previously undocumented. While literature-curated databases of subcellular components, such as the Gene Ontology, are valuable, they are necessarily biased toward well-studied proteins (Kustatscher et al. *Nature Methods* 2022). Furthermore, there is only one Gene Ontology, with no explicit attempt as of yet to create ontologies of cellular components for specific cell types or disease states. We thus feel that large multi-omic data-driven maps of the cell present an important opportunity to elucidate cellular architecture in an unbiased way and in specific cellular contexts.

4. Approximately 20-30% of proteins (if not more) are known to exhibit multifunctional and multicompartmental characteristics, operating within multiple subcellular locations concurrently. Figure 2, however, appears to oversimplify this complex reality, potentially neglecting a critical dimension of cellular organization. It is disconcerting that such a significant aspect of protein behavior and regulation might have been overlooked in this representation.

In fact, the U2OS cell map presented in our paper reveals multicompartmental locations for 30% of proteins, as suspected by the reviewer (1,520 / 5,147). We thus very much agree on the importance of multifunctional, multilocalizing proteins, and we had ourselves previously demonstrated that numerous human proteins have multiple subcellular localizations (Thul et al. 2017). We can see, however, that this aspect was not well highlighted in our manuscript or in Figure 2, which necessarily simplifies the hierarchy of protein assemblies so it can be laid out as a tree-like structure. To address this concern, we now include an explicit analysis of

multilocalizing proteins and protein assemblies, along with a new **Extended Data Fig. 8**. We have added the following text to the **Results**:

Use Case 4: Identification of multi-localizing proteins and complexes

*A substantial fraction of proteins have been postulated to multi-localize, i.e. to play a role in multiple subcellular assemblies or compartments(Thul et al. 2017; Breckels et al. 2024; Bertolini et al. 2024). To this point, we noted that approximately 30% of proteins in the cell map (1,520 / 5,147) are present in more than one distinct assembly (**Extended Data Fig. 8a, Supplementary Table 8**). For example, XAB2, a known factor of the spliceosome and transcription-coupled repair(Donnio et al. 2022), localized not only to nuclear assemblies as expected, but also to the endomembrane (**Extended Data Fig. 8b**). Evidence for such localizations was present in the fluorescent images as well as in the AP-MS interaction network, where XAB2 showed strong interactions with both nuclear spliceosomal and membrane-associated stress factors (**Extended Data Fig. 8c**).*

*Moving beyond single proteins, we also investigated whether there was evidence of multiple localizations for entire protein assemblies, noting 23 that were indeed documented to multi-localize according to the U2OS cell map (**Extended Data Fig. 8d,e**). For example, the Amyloid precursor protein (APP) complex (APP, APBA2, APBA3, APLP2, TJAP1) was clearly resolved in both cytosol and endomembrane compartments (**Extended Data Fig. 8d**), based on evidence from both the protein imaging and biophysical interaction modalities (**Extended Data Fig. 8f,g**). This finding aligns with previous studies showing that APP and its homolog, APLP2, play a role in subcellular trafficking from the endoplasmic membrane (ER) to the cell surface(Kaden et al. 2012) (with vesicular and ER localizations captured in our U2OS imaging data; **Extended Data Fig. 8g**). APBA2 and APBA3 are members of the X11 adaptor protein family, which is known to regulate the translocation of APP(Rogelj et al. 2006). These examples illustrate how a multimodal cell map can reveal both single proteins and whole assemblies that localize to multiple subcellular compartments, suggesting pleiotropic functions.*

Redacted

Extended Data Fig. 8 | Analysis of multi-localized proteins and assemblies. **a)** Number of distinct assemblies per protein, defined as the number of distinct paths to the root of the U2OS cell map hierarchy (shown in Fig. 2). **b)** Tripartite localization of XAB2 in the cell map visualized in circle-packing mode (as per Fig. 7). XAB2 highlighted by red circle and the cell map assemblies in which it participates are highlighted with orange borders. Cytosol and nucleus are filled as blue and pink respectively. **c)** Top: Immunofluorescence images of XAB2 (middle) and representative interacting partners in cytosol (WDR83, left) or nucleus (DHX8, right). These proteins are immunostained (green) with cytoskeleton counterstain (red). Scale bar, 2.5 μm . Bottom: Biophysical interaction partners of XAB2 with cell map localizations in cytosol and membrane (turquoise box) or nucleus (pink box). **d)** Cell map colored to indicate multi-localized assemblies (red nodes). The multiple containing assemblies are indicated in each case (red edges). Dashed box denote the assembly detailed in text and in panel f–g. **e)** Number of assemblies with single (left) versus multiple (right) localizations. **f)** Biophysical interaction data for Amyloid Precursor Protein (APP) complex. **g)** Immunofluorescence images for proteins in APP complex, immunostained (green) with cytoskeleton counterstain (red). Scale bar, 2.5 μm .

5. The use of self-supervised machine learning to fuse IF and AP-MS data relies on the assumption that the features extracted are representative of true biological phenomena. The model's performance might be compromised by systematic biases present in the training data,

such as biases introduced during the selection of proteins for tagging or biases inherent in the imaging and MS techniques. Additionally, the reliance on reconstruction and contrastive loss functions assumes that all relevant biological features can be captured in the learned embedding space, which may not be true if the models lack sufficient capacity or if critical biological features are not well-represented in the input data. Without extensive ground truth datasets to validate these features, the risk of overfitting or capturing spurious correlations remains high.

In development of our approach, we did evaluate the approach for how well it can capture true biological phenomena, including recovery of STRING and CORUM gold standards as well as functionally similar genes (as identified in a recent genome-wide perturb-seq dataset). We compared the self-supervised approach against a panel of alternative methods, including the supervised Random Forests approach used in our previous study by Qin et al. We found that, in practice, the self-supervised approach is not significantly worse than Random Forests for recovery of STRING and CORUM gold standards, and it performs significantly better for recovery of functionally similar genes. We feel these results are particularly impressive given the model was not supervised by these or any other gold-standard pathway or network database.

We now include all of these results in the revised submission. In particular, we have now included new text and a new Extended Data Figure to demonstrate comparison of the self-supervised model with other approaches:

This multimodal embedding exhibited good performance in recovering known subcellular organization (Fig. 2a, Extended Data Fig. 2b-d), performing as well as, or better than, alternative supervised and unsupervised approaches (Methods, Extended Data Fig. 2e).

Extended Data Fig. 2 | Self-supervised embedding of multiple data modalities. e) Different protein embedding approaches (colored points, **Methods**) are evaluated by their degree of enrichment (x-axis) across orthogonal functional and physical interaction resources (y-axis, resources from panels b-d above). Enrichment computed using Cliff's Delta (1,000 samplings of 1,000 protein pairs with replacement, **Methods**) yielding values in range $[-1,1]$, with positive values indicating enrichment above random expectation. Error bars denote standard deviations across 1000 jackknife resamplings. * denotes significant difference in comparison with self-supervised multimodal embedding results ($p < 0.05$ for difference in means across jackknife resamplings, **Methods**).

6. The integration of multiple data types (immunofluorescence, AP-MS, SEC-MS) assumes that these methods are complementary and that discrepancies can be resolved through computational means. However, each method has its own limitations and potential for artifacts, which might not be fully accounted for (or not explained properly in the manuscript) in the integration process. This could lead to erroneous conclusions about the existence or nature of certain protein complexes. For example, how did the authors ensure that the C-terminal FLAG-HA tag did not impact the localization or function of the tagged proteins?

We agree that it is important to explain in the manuscript that each of the three data types has its own limitations and potential for artifacts. A multi-modal integration strategy can mitigate artifacts that differ across modalities, and that is a strength of the approach in comparison to unimodal analysis. We have now added this excellent point to the Discussion:

A separate distinct benefit of a multimodal analysis is that, by design, it provides multiple lines of evidence for new biological findings. In a typical omics study, a single modality of data is presented and analyzed with many putative findings, only a few of which can be validated or pursued at any depth. In contrast, each new finding of the U2OS cell map is

derived from two complementary experimental platforms by default (AP-MS biochemical pulldowns and spatial proteomics imaging), and the systematic lines of evidence deepen further in the Use Cases via support from SEC-MS, AlphaFold predictions, perturb-seq, and/or transposon mutagenesis. For example, the assembly of multifunctional protein ERH with RNA-binding protein CCDC9B was supported by an AP-MS interaction, image subcellular annotations, SEC-MS elution profiles (Fig. 4f), and a high-confidence AlphaFold 3D model (Fig. 4g). Such confluence of data, also seen in other recent multi-omic studies(Cho et al. 2022; Cao and Gao 2022; K. D. Yang et al. 2021; Vickovic et al. 2022), not only increases confidence in each result but also provides substantial additional structural, functional, and/or spatial information.

In regards to the FLAG-HA tag used for AP-MS, the HA tag is very small by design (9 amino acids) and thus does not typically influence the localization of a protein, as demonstrated in previous work (Stadler et al. 2013; Huttlin et al. 2017).

7. While the identification of novel protein assemblies is a highlight of the study, the interpretation of their functional significance is often speculative. The use of large language models like GPT-4 for functional annotation introduces a layer of abstraction that, is interesting and useful, it may not capture nuanced biochemical roles. The reliance on LLMs also raises concerns about interpretability and reproducibility of the annotations, as the models' outputs can be sensitive to phrasing of inputs and the context provided.

We appreciate the reviewer's questions about the reliability of cell map annotation using LLMs. For this reason, while the present manuscript was in review we conducted a separate rigorous study of LLMs as a tool for the functional annotation of gene sets (Hu et al., *Nature Methods*, In Press doi:10.1038/s41592-024-02525-x – an arXiv preprint can be found here: <https://arxiv.org/abs/2309.04019>). This study shows that GPT-4 can accurately synthesize gene set functions for coherent gene sets with high confidence while refusing to name random gene sets. To enhance reproducibility in the present U2OS cell mapping paper, we used a fixed prompt, a specific version of GPT-4 (gpt-4-1106-preview), and conservative settings (temperature set to 0) that ensure consistent outputs.

We further note that GPT-4 annotation was used as just one aspect of a set of complementary tools for suggesting initial names for protein assemblies, alongside traditional enrichment and manual literature analysis by the authors at our annotation jamborees (described in the Results section "Annotation of the U2OS multiscale integrated cell map"). To further support the use of LLMs in the present paper, we now include new Fig. 2d and Extended Data Figs. 3d to confirm that GPT-4 generated names and confidence scores are highly reproducible across replicates. Additionally, GPT-4 provides interpretability by outputting a fully referenced essay (now updated in Extended Data Figs. 3 b, c) of approximately 1000 words providing the rationale and a self-assessment of the confidence for the suggested name. To clarify the use of GPT-4 in providing names for assemblies, we edited and added text to the Results:

In these cases, team members worked collaboratively to consider the current biological literature relevant to the assembly's protein subunits and their potential functions. This process was greatly informed by suggestions from OpenAI's

GPT-4(OpenAI 2023), a generative large language model (LLM) which we had recently shown is capable of providing insightful names and functional interpretations for gene sets identified in 'omics data(Hu et al. 2024 (in press)). As in this previous study, we used an engineered prompt and pipeline (**Extended Data Fig. 4a,b, Methods**) to guide the LLM to generate descriptive names for gene sets indicative of their biological roles, along with a fully referenced analysis essay providing its rationale (**Extended Data Fig. 4c**) and a self-assessment of confidence in the suggested name. When applied to the U2OS cell map, we found that the LLM assigned names to known assemblies with very high confidence (median 0.92 for both high overlap and substantial variation; **Fig. 2d**) and to the previously undocumented assemblies with moderate confidence (median 0.85), contrasting starkly with its confidence for sets of proteins drawn randomly without any correspondence to biological structure (median 0.0). For 104 of the 144 not previously documented assemblies, the literature about the various proteins was sufficiently coherent for GPT-4 to propose a confident assembly name (confidence ≥ 0.85), each of which was subsequently passed to the human curation team for final naming determination (**Supplementary Table 1**).

Fig. 2 | d) GPT-4 self-confidence in generating informative names for assemblies in the cell map, shown for the categories of assemblies denoted in panel b and random assemblies (gray). The distributions of confidence scores are shown as violin plots, with thick black lines representing the median confidence in each category. Significance of differences between distributions calculated by one-sided Mann Whitney U test.

Extended Data Fig. 4 | Annotation of protein assemblies with GPT-4. **a) Left:** Annotation workflow extended from Hu et al. (Hu et al. 2024 (in press)), in which the cell map is used to query GPT-4 for a descriptive name, a confidence score and a supporting rationale. **Right:** Composition of prompt used for GPT-4 query. **b) Schematic of GPT-4 assisted citation module.** GPT-4 is asked to provide gene symbol keywords and functional keywords separately. Multiple gene keywords and functions are combined and used to search PubMed for relevant paper titles and abstracts in the scientific literature. **c) Example assembly with GPT-4 name and supporting analysis paragraphs with citations generated from citation module (see panel d).** **d) Semantic similarity of the original GPT-4 name given an assembly vs. the name assigned in each of five replicate GPT-4 runs.** Results for two example assemblies are shown (yellow and green points), one of which is named identically across replicates (yellow) and one of which shows variation (green). The average performance over all U2OS assemblies ($n=271$, excluding assemblies with more than 1000 proteins) is shown in dark blue.

Point-by-point response to reviewer comments

Referee #1 (Remarks to the Author):

The authors have addressed my comments very thoroughly. In particular, they have showed that the MUSE multimodal data integration can perform as well or better than the previous random forest model and they have clarified what were new experimental datasets, providing the relevant QC metrics for those. I appreciated the addition of the comparison with the HEK293 interactions to emphasize the importance of cell-type specific interactions. Overall, I believe this represents the state-of-the-art for the multi-modal generation of cell-type a specific protein interaction map that can provide a road-map for the development of additional such maps in different cell-types. I have no further concerns.

We sincerely thank the reviewer for their time in reviewing our manuscript. Their feedback was incredibly valuable in improving the study.

Referee #2 (Remarks to the Author):

The authors have successfully addressed all my comments. I was surprised that the self-supervised multimodal embedding works as well as a supervised approach, but on closer inspection of the methods section it appears that the Random Forest was not trained e.g. on a subset of the high-scoring STRING interactions, but on GO annotation similarity. Perhaps that could be mentioned in the figure legend as well. In any case, I think the comparison with the simple input concatenation is more important, and this is clearly outperformed by the self-supervised approach.

I have no more outstanding concerns.

We thank the reviewer for their time reviewing our manuscript and their helpful comments. In response to this comment, we have edited the caption for Extended Data Fig. 2 to clarify the Random Forest was trained with GO.

*e) Different protein embedding approaches (colored points, **Methods**) are evaluated by their degree of enrichment (x-axis) across orthogonal functional and physical interaction resources (y-axis, resources from panels b-d above). Supervised Random Forest trained using the Gene Ontology (**Methods**). Enrichment computed using Cliff's Delta (1,000 samplings of 1,000 protein pairs with replacement, **Methods**) yielding values in range [-1, 1], with positive values indicating enrichment above random expectation. Error bars denote standard deviations across 1000 jackknife resamplings. * denotes significant difference in comparison with self-supervised multimodal embedding results ($p < 0.05$ for difference in means across jackknife resamplings, **Methods**).*

Referee #2 (Remarks on code availability):

I have not reviewed the actual code in detail, because I'm not sufficiently familiar with Python for that. However, I have checked the overall documentation of the code, which is excellent. The code documentation is not limited to a GitHub Readme file, but there is a dedicated website for it. For an earlier and similarly documented version of this software (Qin et al, Nature 2021) a PhD student in my lab had no problems to set up the reported workflow and apply it to a new dataset.

We thank the reviewer for acknowledging the code documentation.

Referee #3 (Remarks to the Author):

The authors have done a commendable job addressing the main concerns, the revised manuscript is improved and now suitable for publication.

We appreciate that the reviewer found our revision commendable. We thank them for their time reviewing the manuscript and thoughtful comments, which certainly improved the manuscript.